# The Forgetting-Retention Dilemma: Certified Unlearning Theory in Continual Learning

Yiting Hu [1 2]  Lingjie Duan [2]  Qian Zhang [1]

## Abstract

Machine unlearning aims to eliminate the influence of specific data from trained models to safeguard privacy. However, this presents a significant challenge in the context of continual learning (CL), where models update sequentially on dynamic datasets. A major limitation is that current certified unlearning algorithms fail to account for the complex, cumulative model evolution inherent to CL framework. In this work, we establish the first theoretical foundation bridging CL and machine unlearning. We formulate the CL's unlearning objective as the minimization of post-unlearning excess risk, which decomposes into CL excess risk and unlearning loss, characterizing the fundamental trade-off between preserving historical knowledge and targeted forgetting. Under mild assumptions, we first establish an upper bound for the CL excess risk in non-convex models. We then adapt two certified unlearning approaches, gradient-based and Hessian-based, to the CL framework. Our analysis reveals that while the gradient-based approach is less effective than the Hessian-based method in minimizing unlearning loss, it offers the distinct advantage of nearly zero storage overhead for enabling unlearning. This insight motivates a hybrid strategy that reduces storage costs while maintaining post-unlearning performance. Experimental results further validate our theoretical findings.

## 1. Introduction

As machine learning applications, ranging from large language models to healthcare systems, increasingly rely on user data for personalization, privacy concerns have grown substantially. To uphold the right to be forgotten and safeguard user privacy, the field of machine unlearning has advanced rapidly, developing methods to eliminate the influence of specific data without the prohibitive cost of retraining models from scratch (Gupta et al., 2021; Warnecke et al., 2023; Sekhari et al., 2021; Suriyakumar & Wilson, 2022; Chien et al., 2022; 2024a; Qiao et al., 2025). Within this landscape, $(\varepsilon, \delta)$-certified unlearning (Guo et al., 2020) has emerged as a promising paradigm. By leveraging insights from differential privacy, these algorithms provide rigorous theoretical guarantees that the output distribution of the unlearning algorithm is indistinguishable from that of retraining on the remaining data.

Certified unlearning methods generally fall into two categories: Hessian-based approaches, which utilize second-order approximations to estimate the retrained model (Sekhari et al., 2021; Suriyakumar & Wilson, 2022; Liu et al., 2023; Qiao et al., 2025; Basaran et al., 2025), and gradient-based approaches, which fine-tune the model on the remaining data via noise-injected updates (Neel et al., 2021; Chien et al., 2024a;b; Koloskova et al., 2025). However, existing certified unlearning algorithms face a critical limitation. Existing methods either necessitate impractical full-data retention (Neel et al., 2021) or, when relaxing this constraint, fail to move beyond isolated, one-time unlearning requests (Sekhari et al., 2021). Consequently, no existing certified unlearning framework simultaneously accommodates the dual constraints of limited data access and the dynamic, sequential nature of unlearning in CL contexts.

Modern machine learning paradigms, such as Large Language Models (LLMs) and dynamic computer vision systems, are increasingly shifting toward CL, in which models are trained on tasks sequentially. Because retaining historical datasets is often infeasible due to storage or privacy constraints, standard CL models struggle with catastrophic forgetting, where acquiring new knowledge severely degrades performance on previously learned tasks (Swartworth et al., 2023; Evron et al., 2022; Deng et al., 2025).

When machine unlearning meets CL, two novel challenges emerge. First, unlearning requests often target data from tasks learned in the distant past. This is problematic because

[1]Singapore University of Technology and Design, Singapore [2]The Hong Kong University of Science and Technology (Guangzhou), Guangzhou, China. Correspondence to: Lingjie Duan <lingjieduan@hkust-gz.edu.cn>.

*Proceedings of the $43^{rd}$ International Conference on Machine Learning*, Seoul, South Korea. PMLR 306, 2026. Copyright 2026 by the author(s).

the model evolves dynamically over non i.i.d. tasks while encodes the information of past data in a complicated way, rendering current certified algorithms ineffective. Second, the goal of effective unlearning inherently conflicts with the objective of preventing forgetting in CL. Hence, a balance must be struck between minimizing the unlearning error and the excess risk of CL, while accounting for storage overhead and the specific influence of unlearning request patterns. Although some experimental works on unlearning in the CL setting have been studied recently, they are vulnerable in terms of privacy protection, as they all fail to satisfy the rigorous certified unlearning requirement. More related works are discussed in Appendix A. We aim to answer the following two questions in this paper:

*(i) How to adapt current certified unlearning method to work in CL framework?*

*(ii) How can the objectives of CL (preventing forgetting) and machine unlearning (effective forgetting) be theoretically reconciled?*

Our main contributions are summarized as follows.

- We present the first theoretical investigation into the problem of certified unlearning within a CL framework, referred as continual learning-unlearning (CLU) (Vahedifar et al., 2025). Our research establishes a novel analytical connection, bridging the excess risk introduced by CL and the unlearning loss resulting from unlearning. To begin, we prove an excess-risk bound for the simple but fundamental $\ell_2$-regularized CL algorithm, extending prior results from linear to nonlinear models.

- We adapt the existing gradient-based certified unlearning that leverages the forgetting effect inherent to CL, directly adding noise to achieve certified unlearning. Our analysis reveals that while this approach may suffer from high unlearning loss, particularly for unlearning recent tasks, its primary advantage of zero storage overhead makes it a crucial benchmark for CLU.

- We adapt Hessian-based unlearning to the CL framework, overcoming the challenge of handling arbitrary unlearning request sequences, achieving a significantly lower unlearning loss than the gradient-based method. To reduce the Hessian-based method's storage costs while maintaining post-unlearning performance, we incorporate the gradient-based unlearning to optimize Hessian-based unlearning. These theoretical findings are validated by experiments on real world dataset.

## 2. Problem formulation for CLU

In this section, we formalize CLU as a sequential two-stage process consisting of learning and unlearning, as illustrated

in Fig. 1. We then define the $(\varepsilon, \delta)$-certified continual unlearning problem building upon this framework. Finally, we introduce the post-unlearning excess risk, which serves as the primary performance measure in the CLU setting and captures the interplay between CL and certified unlearning.

### 2.1. Two-stage CLU framework

CLU refers to a learning paradigm in which a model is trained sequentially on a stream of tasks, while being required to selectively remove the influence of tasks upon request. Specifically, we consider a discrete time horizon $t = 1, \ldots, T$. At each time step $t$, the learner first observes and update the model on a new task with dataset $D_t = \{z_i\}_{i=1}^{n_t} \subseteq \mathcal{Z}^{n_t}$, where $n_t$ denotes the sample size of task $t$. In addition, at time $t$ the learner may receive an unlearning request specified by an index set $S_t \subseteq \{1, 2, \ldots, t\}$, requiring the immediate removal of the influence of datasets $\{D_i\}_{i \in S_t}$ from the current model.[1]

To facilitate the demonstration, we further define the sequence of historical task dataset and historical unlearning index set upon time $t$ as $\mathbf{D}_{1:t} := (D_1, \ldots, D_t)$ and $\mathbf{S}_{1:t} := (S_1, \ldots, S_t)$, respectively. For $t, t' = 1, \ldots, T$, we denote by $w_t^{-\mathbf{S}_{1:t'}}$ the model parameter produced by the CLU process. The subscript $t$ indicates that task datasets up to time $t$ have been learned, while the superscript $-\mathbf{S}_{1:t'}$ denotes that the datasets indexed by $\mathbf{S}_{1:t'}$ have been unlearned.

**Stage I: Learning from $D_t$.** Let $\ell(w, z)$ denote the training loss for a model parameter $w \in \mathcal{W} \subseteq \mathbb{R}^d$ evaluated on a data instance $z \in \mathcal{Z}$. As shown in Fig. 1, at time $t$, we first train on the new task $D_t$ starting from the model obtained at the previous time step $w_{t-1}^{-\mathbf{S}_{1:t-1}}$. We employ the widely adapted $\ell_2$-regularized CL scheme for training, i.e., the model parameter is updated by solving

$$w_t^{-\mathbf{S}_{1:t-1}} \in \arg\min_{w \in \mathcal{W}} \sum_{z_i \in D_t} \frac{\ell(w, z_i)}{n_t} + \frac{\lambda}{2} \|w - w_{t-1}^{-\mathbf{S}_{1:t-1}}\|_2^2.$$
$$(\ell_2\text{-CL})$$

The first term in the objective above minimizes the empirical loss on the current dataset, while the second term mitigates the forgetting of prior tasks by regularizing the model parameters. This method has shown practical effectiveness (Kirkpatrick et al., 2017), and its tractable structure has also made it central to recent theoretical studies of CL (Evron et al., 2022; Lin et al., 2023). We therefore initiate the study of certified unlearning in CL under this method.

**Stage II: Unlearning of $S_t$.**

As illustrated in Fig. 1, after completing the learning stage at time $t$ and obtain model $w_t^{-\mathbf{S}_{1:t-1}}$, the system may re-

---

[1]We focus on task-level unlearning in our CLU framework, while our method can be readily extended to data-level deletion.

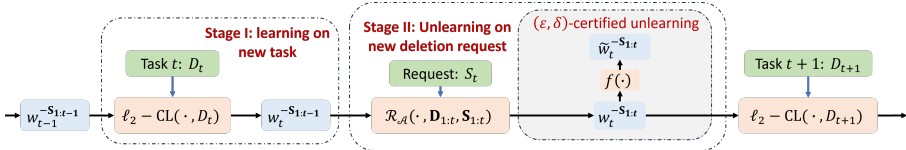

*Figure 1.* Two-stage CL and unlearning at time $t$: starting from the last model $w_{t-1}^{-\mathbf{S}_{1:t-1}}$ at time $t-1$, we first train on task $t$ with dataset $D_t$ to obtain $w_t^{-\mathbf{S}_{1:t-1}}$ in Stage I. Upon receiving a possible deletion request $S_t$, in Stage II, the unlearning scheme $\mathcal{R}_{\mathcal{A}}(\cdot, \mathbf{D}_{1:t}, \mathbf{S}_{1:t})$ in (1) updates the internal model $w_t^{-\mathbf{S}_{1:t}}$, and publishes the final unlearning model $\tilde{w}_t^{-\mathbf{S}_{1:t}}$, by noise adding mapping $f$ in (3) to achieve $(\varepsilon, \delta)$-certified continual unlearning in Definition 2.1. Then, it feeds $w_t^{-\mathbf{S}_{1:t}}$ to train on next task $t+1$.

ceive an unlearning request $S_t \subseteq \{1, \ldots, t\} \setminus S_{\leq t-1}$, where $S_{\leq t-1} = \bigcup_{i=1}^{t-1} S_i$ denotes the set of tasks previously requested for deletion, with $S_0 = \emptyset$. Given the current internal model state $w_t^{-\mathbf{S}_{1:t-1}}$, we apply an unlearning algorithm $\mathcal{R}_{\mathcal{A}}$ to remove the influence of the tasks indexed by $S_t$, resulting in an updated model state $w_t^{-\mathbf{S}_{1:t}}$. If $S_t = \emptyset$, no unlearning is performed and $w_t^{-\mathbf{S}_{1:t}} = w_t^{-\mathbf{S}_{1:t-1}}$. To achieve certified unlearning, the internal model is processed by a noise-adding mechanism $f$ to produce the publish model: $\tilde{w}_t^{-\mathbf{S}_{1:t}} = f(w_t^{-\mathbf{S}_{1:t}})$. The internal model $w_t^{-\mathbf{S}_{1:t}}$ is then carried over to time $t+1$ for further training.

### 2.2. Objectives and guarantees in CLU

Given an unlearning-request history $\mathbf{S}_{1:t}$, the goal of certified unlearning is to produce an updated model that matches the model obtained by perfect retraining on the remaining data. We define an unlearning algorithm at time $t$ by

$$w_t^{-\mathbf{S}_{1:t}} = \mathcal{R}_{\mathcal{A}}\left(w_t^{-\mathbf{S}_{1:t-1}}, \mathbf{D}_{1:t}, \mathbf{S}_{1:t}\right), \quad (1)$$

which aims to approximate the perfect retraining model

$$w_t^{-S_{\leq t}} = \mathcal{A}\left((D_i)_{i \in [t] \setminus S_{\leq t}}\right), \qquad S_{\leq t} := \bigcup_{i=1}^{t} S_i, \quad (2)$$

where $\mathcal{A}$ denotes the CL procedure ($\ell_2$-CL) applied to the sequence of remaining tasks $[t] \setminus S_{\leq t}$. Note that $w_t^{-S_{\leq t}}$ depends only on the retained datasets, whereas $\mathcal{R}_{\mathcal{A}}$ may depend on the sequence of entire request history $\mathbf{S}_{1:t}$. To provide a certified guarantee, we require the publish model

$$\tilde{w}_t^{-\mathbf{S}_{1:t}} = f\left(w_t^{-\mathbf{S}_{1:t}}\right) \quad (3)$$

to be distributionally indistinguishable from the randomized retraining output $\tilde{w}_t^{-S_{\leq t}} = f\left(w_t^{-S_{\leq t}}\right)$. We formalize this requirement by extending the $(\varepsilon, \delta)$-certified unlearning (Neel et al., 2021) to the continual setting.

**Definition 2.1** ($(\varepsilon, \delta)$-**certified continual unlearning**). Let $\varepsilon > 0$ and $\delta \geq 0$. An unlearning algorithm $f \circ \mathcal{R}_{\mathcal{A}}$ satisfies $(\varepsilon, \delta)$-*certified continual unlearning* if, for every time $t$, every dataset sequence, every admissible deletion-request

history, and every measurable subset $\mathcal{O} \subseteq \mathcal{W}$,

$$\Pr\left(\tilde{w}_t^{-\mathbf{S}_{1:t}} \in \mathcal{O}\right) \leq e^{\varepsilon} \Pr\left(\tilde{w}_t^{-S_{\leq t}} \in \mathcal{O}\right) + \delta, \quad (4)$$

$$\Pr\left(\tilde{w}_t^{-S_{\leq t}} \in \mathcal{O}\right) \leq e^{\varepsilon} \Pr\left(\tilde{w}_t^{-\mathbf{S}_{1:t}} \in \mathcal{O}\right) + \delta. \quad (5)$$

Under the $(\varepsilon, \delta)$-certified unlearning constraint, the objective of CLU is to minimize the post-unlearning excess risk over all retained tasks, defined as follows.

**Definition 2.2** (Post-unlearning excess risk). Let $\mathcal{D}_\tau$ denote the data-generating distribution of task $\tau$, from which the dataset $D_\tau$ is drawn, and define the population loss for task $\tau$ as $F_\tau(w) = \mathbb{E}_{z \sim \mathcal{D}_\tau}[\ell(w, z)]$. At time $t$, the post-unlearning excess risk of a CLU algorithm is defined as

$$\mathcal{E}_{\mathrm{LU}} := \mathbb{E}\left[\frac{1}{N_t} \sum_{\substack{\tau=1 \\ \tau \notin S_{\leq t}}}^{t} F_\tau(\tilde{w}_t^{-\mathbf{S}_{1:t}})\right] - \min_{w \in \mathcal{W}} \frac{1}{N_t} \sum_{\substack{\tau=1 \\ \tau \notin S_{\leq t}}}^{t} F_\tau(w), \quad (6)$$

where $N_t := t - |S_{\leq t}|$, and we consider $t$ with $N_t \geq 1$. Here, the expectation is taken over the randomness of the per-task datasets $\{D_\tau\}_{\tau=1}^{t}$ and the publish noise-adding map $f$ used to produce $\tilde{w}_t^{-\mathbf{S}_{1:t}}$.

The post-unlearning excess risk in (6) admits an decomposition as $\mathcal{E}_{\mathrm{LU}} = \mathcal{E}_{\mathrm{U}} + \mathcal{E}_{\mathrm{L}}$, where

$$\mathcal{E}_{\mathrm{U}} := \mathbb{E}\left[\frac{1}{N_t} \sum_{\substack{\tau=1 \\ \tau \notin S_{\leq t}}}^{t} \left(F_\tau(\tilde{w}_t^{-\mathbf{S}_{1:t}}) - F_\tau(w_t^{-S_{\leq t}})\right)\right], \quad (7)$$
$$\underbrace{\qquad\qquad\qquad\qquad\qquad\qquad\qquad\qquad}_{\text{unlearning loss}}$$

$$\mathcal{E}_{\mathrm{L}} := \mathbb{E}\left[\frac{1}{N_t} \sum_{\substack{\tau=1 \\ \tau \notin S_{\leq t}}}^{t} F_\tau(w_t^{-S_{\leq t}})\right] - \min_{w \in \mathcal{W}} \frac{1}{N_t} \sum_{\substack{\tau=1 \\ \tau \notin S_{\leq t}}}^{t} F_\tau(w). \quad (8)$$
$$\underbrace{\qquad\qquad\qquad\qquad\qquad\qquad\qquad\qquad}_{\text{continual learning excess risk}}$$

Here, $\mathcal{E}_{\mathrm{U}}$ captures the unlearning loss induced by the unlearning procedure, while $\mathcal{E}_{\mathrm{L}}$ corresponds to the excess risk incurred by the CL algorithm on the retained tasks. Since the unlearning loss is controlled by the deterministic approximation error $\|w_t^{-\mathbf{S}_{1:t}} - w_t^{-S_{\leq t}}\|$, we also refer to this term as the *approximation error*.

This decomposition highlights a fundamental tension inherent in CLU: strictly mitigating forgetting to minimize

the learning error $\mathcal{E}_\mathrm{L}$ in (8) can inadvertently exacerbate the unlearning error $\mathcal{E}_\mathrm{U}$ in (7). This phenomenon stands in sharp contrast to previous certified unlearning studies in static settings (Sekhari et al., 2021; Suriyakumar & Wilson, 2022; Liu et al., 2023), where minimizing excess risk typically aligns with and facilitates the unlearning process. Consequently, CLU necessitates a novel trade-off: balancing performance retention during CL with the requirement for efficient and certified unlearning.

We introduce the following standing assumption on the loss function, which will be used throughout the paper.

**Assumption 2.3** (Local curvature and smoothness). There exists a radius $r \geq 0$ such that for any $w \in \overline{\mathcal{B}}(w_0, r)$, the loss function $\ell(w, z)$ for any $z \in \mathcal{Z}$ is twice continuously differentiable almost everywhere. On the set of where $\ell(\cdot, z)$ is twice continuously differentiable, its Hessian satisfies

$$\mu I \preceq \nabla^2 \ell(w, z) \preceq MI, \ \forall z \in \mathrm{supp}(\mathcal{D}_t) \qquad (9)$$

for $t = 1 \ldots, T$ and for finite constants $\mu \leq M$. Furthermore, $\ell(w, z)$ is $L$-Lipschitz continuous.

The assumption of almost everywhere (a.e.) twice continuous differentiability is satisfied by many modern architectures, including deep neural networks with piecewise smooth activations. In practice, the Hessian may fail to exist only on nondifferentiable boundaries, such as ReLU kink points or the origin in $L_1$ regularization. These sets have Lebesgue measure zero, so the Hessian is well-defined along almost every point of the localized parameter region. The bounded-curvature requirement is also mild because it is imposed only locally on the bounded region visited by the training and unlearning trajectories, and the constants $\mu$ and $M$ only need to be finite. In particular, allowing $\mu < 0$ accommodates non-convex loss landscapes.

**Assumption 2.4** (Localized empirical minimizers). Set the parameter space as $\mathcal{W} = \mathcal{B}(w_0, r)$. For each task $t$, define the set of empirical risk minimizers as $W_t = \{w \in \mathcal{W}: \sum_{z \in D_t} \ell(w, z)/n_t = \min_{w' \in \mathcal{W}} \sum_{z \in D_t} \ell(w', z)\}$. We assume that there exists an empirical minimizer $\hat{w}_t \in W_t$. Moreover, the deterministic internal states generated by the CLU and retraining procedures exist and remain in $\mathcal{W}$.

Assumption 2.4 is widely applicable to modern CL settings where training is restricted to a bounded or localized parameter region, as is common in analyses of overparameterized models. Theoretical advancements in deep learning, especially within the neural tangent kernel (NTK) framework, demonstrate that sufficiently wide networks can converge to a global minimizer while moving only a small distance from the initial parameters $w_0$ (Jacot et al., 2018; Chizat et al., 2019; Oymak & Soltanolkotabi, 2019; Li & Liang, 2018). Thus, it is natural to focus on a localized parameter space around the initialization, namely $\mathcal{W} = \mathcal{B}(w_0, r)$.

## 3. Preliminary Results on Excess Risk in CL

Before turning to the unlearning analysis, we establish a theoretical upper bound on the CL excess risk $\mathcal{E}_\mathrm{L}$ in (8) for the $\ell_2$-regularized update in ($\ell_2$-CL). This result serves as a fundamental building block for our subsequent analysis of the post-unlearning excess risk.

**Theorem 3.1.** *Let $\{\tau_1, \ldots, \tau_{N_t}\} := [t] \setminus S_{\leq t}$ denote the indices of the retained tasks at time $t$. Suppose Assumptions 2.3 and 2.4 hold, and choose $\lambda > \max\{0, -\mu\}$. Let $B := \sup_{w \in \mathcal{W}, z \in \mathcal{Z}} |\ell(w, z)|$. Then, for the perfect retraining model $w_t^{-S_{\leq t}} = \mathcal{A}((D_i)_{i \in [t] \setminus S_{\leq t}})$, the CL excess risk defined in (8) satisfies $\mathcal{E}_\mathrm{L} \leq \mathcal{E}^{-S_{\leq t}}(\lambda)$, where*

$$\mathcal{E}^{-S_{\leq t}}(\lambda) := L\left(2r\kappa \frac{(N_t - 1)(1 - \rho^{N_t})}{N_t(1 - \rho)} + r\rho^{N_t}\right) + \Gamma\left((n_{\tau_k})_{k=1}^{N_t}\right) \qquad (10)$$

*with $\rho := \frac{\lambda}{\mu + \lambda}, \kappa := \max\left\{\frac{M}{M+\lambda}, \frac{|\mu|}{\mu+\lambda}\right\}$. Here, the generalization term $\Gamma\left((n_{\tau_k})_{k=1}^{N_t}\right)$ is*

$$\Gamma\left((n_{\tau_k})_{k=1}^{N_t}\right)$$
$$:= \sum_{k=1}^{N_t} \left[\frac{4B}{N_t}\sqrt{\frac{2d\ln\left(2Lr\sqrt{n_{\tau_k}}/B + 1\right)}{n_{\tau_k}}} + \frac{2B}{N_t\sqrt{n_{\tau_k}}}\right].$$

The proof of Theorem 3.1 is provided in Appendix C.1. The bound in Theorem 3.1 consists of two interpretable components. The first group of terms reflects the cumulative effect of parameter drift across tasks and are controlled by the contraction factor $\rho = \lambda/(\mu + \lambda)$ introduced by the $\ell_2$ regularization. The second term corresponds to statistical generalization error arising from finite samples in each task.

This result extends existing generalization analyses for ($\ell_2$-CL) of linear models (e.g., Lin et al. (2023)). The regularization weight $\lambda$ that minimizes the upper bound $\mathcal{E}^{-S_{\leq t}}(\lambda)$ depends on the diameter of the parameter space. In addition, one can derive a refined bound whose minimizer also depends on the degree of task dissimilarity. Notably, the upper bound does not vanish even as the per-task sample sizes $n_{\tau_i}$ grow large, reflecting an irreducible error arising from model heterogeneity and the non-i.i.d. nature of the data stream. We analyze the optimal $\lambda$ for $\mathcal{E}_\mathrm{L}$ within a concrete example in Appendix C.2. Generally, the $\lambda$ that minimizes the excess risk $\mathcal{E}_\mathrm{L}$ can differ significantly from the one that minimizes the unlearning error $\mathcal{E}_\mathrm{U}$.

## 4. Gradient-based CLU

In this section, we analyze the natural forgetting phenomenon inherent in the classical CL paradigm as a preliminary baseline that exposes the fundamental trade-off

between forgetting and knowledge preservation in CL. The insights gained here will later be leveraged to guide the design and optimization of more powerful unlearning algorithms in subsequent sections.

In particular, information associated with long-past data is gradually diminished through sequential updates in CL process. This process naturally aligns to the mechanics of gradient-based certified unlearning methods (Neel et al., 2021). However, since natural forgetting can result in the total erasure of historical knowledge, directly contradicting the core objective of CL, this gradient-based unlearning method must be carefully controlled. Hence, we adapt the model state $w_t = \mathcal{A}(\mathbf{D}_{1:t})$ as the internal representation, bypass the explicit unlearning step $\mathcal{R}_{\mathcal{A}}$ and directly calibrate noise to mask the discrepancy between $w_t$ and the target model $w_t^{-S_{\leq t}}$. This prevents indiscriminately removing information from all historical data.

Specifically, in Stage I of Alg. 1, the model is updated according to the CL update rule in ($\ell_2$-CL) as new tasks arrive. In Stage II, if the set $S_{\leq t}$ contains any unlearning requests up to time $t$, we perturb the model parameters with Gaussian noise, using a standard deviation determined by the quantity $\gamma_t(\mathbf{S}_{1:t})$ as specified in Alg. 1 line 8. We next show that $\gamma_t(\mathbf{S}_{1:t})$ provides a rigorous upper bound on the approximation error $\|w_t^{-S_{\leq t}} - w_t\|$. This Gaussian mechanism directly ensures $(\varepsilon, \delta)$-certified unlearning as a consequence of standard results from (Dwork et al., 2014; Qiao et al., 2025).

---

**Algorithm 1** Natural forgetting CLU
___
1: Initialize $S_{\leq 0} = \emptyset$, $U_0 = \emptyset$, and $\mathbf{S}_{1:0} = \emptyset$.
2: **for** $t \leftarrow 1$ to $T$ **do**
    **Stage I. CL on task** $t$
3:    Receive dataset $D_t$, $w_t \leftarrow \ell_2$-CL$(w_{t-1}, D_t)$.
    **Stage II. Model Publishing**
4:    Receive deletion request $S_t \subseteq [t] \setminus S_{\leq t-1}$.
5:    Update $S_{\leq t} \leftarrow S_{\leq t-1} \cup S_t$ and $\mathbf{S}_{1:t} \leftarrow (\mathbf{S}_{1:t-1}, S_t)$.
6:    **if** $S_{\leq t} = \emptyset$ **then**
7:       $U_t \leftarrow U_{t-1}$ and $\epsilon_t \leftarrow \mathbf{0}$.
8:    **else**
9:       $U_t \leftarrow U_{t-1} \cup \{t : S_t \neq \emptyset\}$.
10:       Draw $\epsilon_t \sim \mathcal{N}(\mathbf{0}, \sigma^2 I)$, $\sigma = \gamma_t(\mathbf{S}_{1:t})\sqrt{2\ln(1.25/\delta)}/\varepsilon$, where $\gamma_t(\mathbf{S}_{1:t})$ is given by (11).
11:    **end if**
12:    **Output**: $\tilde{w}_t^{-\mathbf{S}_{1:t}} \leftarrow w_t + \epsilon_t$
13: **end for**
___

Let $U_t = \{t_1, \ldots, t_k\}$ be the set of time indices at which unlearning requests were received. For any $i \in \{1, \ldots, k\}$, let $n_{[a,b]}^i$ denote the number of tasks in $[a, b]$ that have been deleted up to the $i$-th deletion request. The formal performance guarantees for Alg. 1 are established in the following theorem.

**Theorem 4.1.** *Let $\lambda > \max\{0, -\mu\}$, and define*

$$\gamma_t(\mathbf{S}_{1:t}) := \frac{L}{\lambda} \sum_{i=1}^{k} \sum_{s \in S_{t_i}} \rho^{t-s-n_{[s+1,t]}^k}. \quad (11)$$

*We have $\|w_t^{-S_{\leq t}} - w_t\| \leq \gamma_t(\mathbf{S}_{1:t})$, and the CLU process following Alg. 1 satisfies $(\varepsilon, \delta)$-certified continual unlearning as defined in Definition 2.1. Moreover, the output model $\tilde{w}_t^{-\mathbf{S}_{1:t}}$ produced by Alg. 1 achieves a post-unlearning excess risk $\mathcal{E}_{\mathrm{LU}}$ that is upper bounded by*

$$L \cdot \gamma_t(\mathbf{S}_{1:t}) \cdot \left( \sqrt{2d\ln(\frac{1.25}{\delta})}/\varepsilon + 1 \right) + \mathcal{E}^{-S_{\leq t}}(\lambda), \ (12)$$

*where $\mathcal{E}^{-S_{\leq t}}(\lambda)$ denotes the excess risk of the underlying CL algorithm on the remaining tasks, as defined in (10).*

The proof of Theorem 4.1 is provided in Appendix D.1. In (11), each unlearned task $s$ contributes a drift error term proportional to $\rho^{t-s-n_{[s+1,t]}^k} \frac{L}{\lambda}$, which quantifies the residual influence of task $s$ on the current model. In particular, when the loss function is locally convex, i.e., $\mu > 0$, the contribution of each unlearned task decays exponentially with respect to the effective time gap $t - s - n_{[s+1,t]}^k$. This result formalizes the natural forgetting effect of $\ell_2$-regularized CL, where the influence of historical tasks diminishes.

Note that while the publish model (3) produced by Alg. 1 satisfies $(\varepsilon, \delta)$-certified unlearning, the algorithm maintains an internal model state $w_t$ to support subsequent task updates. This internal state may still retain information from deleted datasets. To address this issue, we provide an extension of Alg. 1 in Appendix D.2, which ensures a stronger unlearning guarantee for the internal model state as well.

Alg. 1 is a storage-free baseline that effectively unlearns early tasks, but its dependence on natural forgetting leads to inconsistent performance on recent tasks. To address this, the next section introduces a Hessian-based approach that enables accurate, targeted unlearning for any possible task while maintaining computational and storage efficiency.

## 5. Hessian-based CLU and Gradient Enhancement

In this section, we first adapt Hessian-based unlearning methods to the CL setting. To reduce the Hessian-based method's storage costs while maintaining post-unlearning performance, we further incorporate the gradient-based unlearning to optimize Hessian-based unlearning.

### 5.1. Hessian-based unlearning algorithm for CLU
A pivotal challenge for Hessian-based unlearning in the CL setting arises from the temporal order of unlearning requests. When unlearning requests are *forward-synchronous* as in

Fig. 2, the target tasks are learned strictly after the most recent unlearning event, where previously unlearned data do not affect the current model state, and no additional correction is required for earlier unlearning operations. In contrast, *asynchronous* unlearning requests in Fig. 2 break this clean recursive structure. If an unlearning request targets tasks learned prior to the last unlearning event, their influence has already been propagated into subsequent model updates and implicitly encoded in the Hessian approximations.

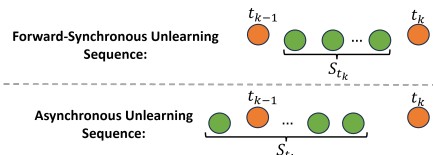

*Figure 2.* Let $t_{k-1}$ and $t_k$ denote two consecutive unlearning time points. An unlearning request sequence is said to be **forward-synchronous** if, at each unlearning time $t_k$, the set of requested tasks $S_{t_k}$ to unlearn contains only CL tasks learned strictly after the previous unlearning event $t_{k-1}$, i.e., $\forall j \in S_{t_k}$, $j > t_{k-1}$. Otherwise, the unlearning sequence is termed **asynchronous**.

We further elucidate this specific challenges of unlearning in CL by outlining the underlying analysis and intuition behind our single-step, Hessian-based update. Specifically, consider a time step $t_k$ at which $S_{t_k} \neq \emptyset$. Suppose two most unlearning requests occur at time $t_{k-1}$ and $t_k$. The objective of the unlearning stage at time $t_k$ is to approximate the retrained model $w_{t_k}^{-S_{\leq t_k}}$, using the current model state $w_{t_k}^{-\mathbf{S}_{1:t_k-1}}$. Let $\hat{F}_i(w) := \frac{1}{|D_i|} \sum_{z_i \in D_i} \ell(w, z_i)$ denote the empirical loss for task $i$. Under the update rule ($\ell_2$-CL) at time $t_k$, the following first-order optimality conditions hold:

$$\nabla \hat{F}_{t_k}(w_{t_k}^{-S_{\leq t_k}}) + \lambda(w_{t_k}^{-S_{\leq t_k}} - w_{t_k-1}^{-S_{\leq t_k}}) = 0,$$
$$\nabla \hat{F}_{t_k}(w_{t_k}^{-\mathbf{S}_{1:t_k-1}}) + \lambda(w_{t_k}^{-\mathbf{S}_{1:t_k-1}} - w_{t_k-1}^{-\mathbf{S}_{1:t_k-1}}) = 0.$$

Taking the first-order Taylor expansion of $\nabla \hat{F}_{t_k}(w_{t_k}^{-S_{\leq t_k}})$ at $w_{t_k}^{-\mathbf{S}_{1:t_k-1}}$ and combining the two equations above yield

$$w_{t_k}^{-S_{\leq t_k}} - w_{t_k}^{-\mathbf{S}_{1:t_k-1}}$$
$$\approx (\nabla^2 \hat{F}_{t_k}(w_{t_k}^{-\mathbf{S}_{1:t_k-1}}) + \lambda I)^{-1} \lambda(w_{t_k-1}^{-S_{\leq t_k}} - w_{t_k-1}^{-\mathbf{S}_{1:t_k-1}}).$$

However, this local approximation propagates in a complicated way under asynchronous unlearning. For a previous time index that has not been deleted by time $t_k$, the corresponding retrained state is unchanged by removing that index from the deletion set. Recursively propagating this relation backward to an earlier time step $\tau$, we obtain a divergence between the model states $w_\tau^{-S_{\leq t_k} \setminus \{\tau, \dots, t_k\}}$ and $w_\tau^{-\mathbf{S}_{1:\tau}}$. In general, the task sets $S_{\leq t_k} \setminus \{\tau, \dots, t_k\}$ and $\mathbf{S}_{1:\tau}$ do not coincide, since unlearning requests may arrive arbitrarily in an asynchronous sequence. This misalignment introduces substantial heterogeneity into subsequent iterations, complicating the update process.

To address these challenges, we propose the following unlearning algorithm upon receiving an unlearning request:

$$w_t^{-\mathbf{S}_{1:t}} = \mathcal{R}_{\mathcal{A}}\left(w_t^{-\mathbf{S}_{1:t-1}}, \mathbf{D}_{1:t}, \mathbf{S}_{1:t}\right) = w_t^{-\mathbf{S}_{1:t-1}} + \bar{\Delta}_t, \quad (13)$$

where $\bar{\Delta}_t$ is a Hessian-based correction term designed to approximate the retrained model in a single update:

$$\bar{\Delta}_t = \sum_{s \in S_t} \prod_{i=s+1, i \notin S_{\leq t}}^{t} \left((\hat{H}_i + \lambda I)^{-1} \lambda I\right) \Delta_s$$
$$+ \left( \sum_{s \in S_{\leq t-1}} \prod_{i=s+1, i \notin S_{\leq t}}^{t} \left((\hat{H}_i + \lambda I)^{-1} \lambda I\right) \Delta_s \right.$$
$$\left. - \sum_{\tau \in U_{t-1}} \prod_{i=\tau+1, i \notin S_{\leq t}}^{t} \left((\hat{H}_i + \lambda I)^{-1} \lambda I\right) \bar{\Delta}_\tau \right). \quad (14)$$

Here $\hat{H}_t$ and $\Delta_t$ denote the stored Hessian information and the model update associated with task $t$, respectively, and are computed as

$$\Delta_t \leftarrow w_{t-1}^{-\mathbf{S}_{1:t-1}} - w_t^{-\mathbf{S}_{1:t-1}},$$
$$\hat{H}_t \leftarrow \sum_{z_i \in D_t} \hat{\nabla}^2 \ell(w_t^{-\mathbf{S}_{1:t-1}}, z_i)/|D_t|, \quad (15)$$

where $\hat{\nabla}^2$ denotes an approximation of the exact Hessian. Intuitively, the first part in $\bar{\Delta}_t$ of (14) isolates and removes the influence of the current unlearning requests $S_t$. The second part acts as a recursive correction that compensates for interactions between earlier unlearning requests $S_{\leq t-1}$ and the historical correction terms $\{\bar{\Delta}_\tau\}_{\tau \in U_{t-1}}$ accumulated along the learning trajectory. Together, these components yield a second-order approximation of the retrained model for arbitrary unlearning orders.

Implementing the unlearning algorithm $\mathcal{R}_{\mathcal{A}}$ requires storing $\hat{H}_t$, $\Delta_t$, and $\mathbf{S}_{1:t}$ during Stage I of the Hessian-based CLU procedure. Subsequently, in Stage II, the model $w_t^{-\mathbf{S}_{1:t}}$ is updated via (13) whenever $S_t \neq \emptyset$ before model publishing. The complete CLU procedure incorporating this unlearning step is summarized in Alg. 2 of Appendix.

**Remark**. The algorithm $\mathcal{R}_{\mathcal{A}}$ in (13) is designed to handle arbitrary unlearning requests. When the unlearning request is forward-synchronous, (14) reduces to its first term, and the complete Hessian-based unlearning in Alg. 2 of the Appendix still works.

While our analysis is grounded in the theoretical properties of Hessian-based methods, we also account for practical considerations by incorporating numerical approximation $\hat{\nabla}^2$ to estimate the Hessian, avoiding the high computational cost of exact calculation in large-scale models. For instance, diagonal Hessian approximation reduces both computation and storage complexity from $O(d^3)$ and $O(d^2)$ to $O(d)$. A

detailed discussion of the choice of approximation methods and its impact is deferred to Appendix E.1. Later we will further reduce the storage dependence on $t$.

## 5.2. Post-unlearning excess risk of Hessian-based CLU

Analogous to Theorem 4.1, the post-unlearning excess risk $\mathcal{E}_{\mathrm{LU}}$ of Alg. 2 is governed by the approximation error $\|w_t^{-S_{\leq t}} - w_t^{-\mathbf{S}_{1:t}}\|$. We begin by analyzing this approximation error induced by the Hessian-based unlearning in (13).

We introduce the following assumption on the accuracy of the Hessian approximation operator $\hat{\nabla}^2$ used in Alg. 2, thereby formally incorporating the approximation into our analysis and bridges the gap between the theoretical guarantees and practical implementations.

**Assumption 5.1.** For any data instance $z \in \mathcal{Z}$ and parameter $w \in \mathcal{W}$, the approximation operator satisfies $\|\hat{\nabla}^2 \ell(w, z) - \nabla^2 \ell(w, z)\|_2 \leq \nu$ where $\|\cdot\|_2$ denotes the spectral norm and $\nu > 0$ is a constant value.

The specific value of $\nu$ depends on the chosen approximation scheme. For example, sub-sampled Hessian methods satisfy this bound with high probability when the batch size is sufficiently large (Tropp, 2015).

Building on Assumption 5.1, Proposition 5.2 establishes a first-order upper bound on the approximation error. Furthermore, Proposition 5.3 derives a tighter second-order upper bound under the additional assumption that the Hessian is Lipschitz continuous with constant $L_3$. The detailed proofs for both propositions are deferred to Appendix E. We focus on nondegenerate cases where $\mu \neq 0$ and $\mu - \nu \neq 0$.

**Proposition 5.2.** Let $U_t = \{t_1, \ldots, t_k\}$ be the set of time indices at which unlearning requests were received up to time $t$. Choose $\lambda > \max\{0, \nu - \mu\}$. Then, the approximation error $\|w_t^{-S_{\leq t}} - w_t^{-\mathbf{S}_{1:t}}\|$ of the unlearning operator $\mathcal{R}_\mathcal{A}$ in (13) is bounded by

$$\gamma_t(\mathbf{S}_{1:t}) := C_2 \rho^{t-t_k} \sum_{i=1}^{k} \sum_{s \in S_{t_i}} \beta_{i,s}^1 +$$
$$C_2 \rho^{t-t_k} \sum_{i=1}^{k} \sum_{s \in S_{t_i}} \hat{\rho}^{t_k - s - n_{[s+1,t_k]}^k} \sum_{a=i+1}^{k} \beta_{i,s,a}^2, \quad (16)$$

where the constant $C_2 := \frac{L(M - \mu + \nu)}{\nu \lambda}$, and

$$\beta_{i,s}^1 := \hat{\rho}^{t_k - s - n_{[s+1,t_k]}^k} \alpha^{t_k - t_i - n_{[t_i+1,t_k]}^k}$$
$$\times \left[ 1 - \alpha^{t_i - s - n_{[s+1,t_i]}^k} \right],$$
$$\beta_{i,s,a}^2 := C_{k-i}^{k-a+1}(t_k, t_i, s) \alpha^{t_k - t_a - n_{[t_a+1,t_k]}^k}$$
$$\times \left[ 1 - \alpha^{t_a - t_{a-1} - n_{[t_{a-1}+1,t_a]}^k} \right].$$

with $\hat{\rho} := \frac{\lambda}{\lambda + \mu - \nu}$ and $\alpha := \rho/\hat{\rho}$.

Here, the coefficient $C_{k-i}^{k-a+1}(t_k, t_i, s)$ quantifies the temporal disorder of the unlearning sequence. More asynchronous unlearning requests lead to larger values of $C(\cdot)$, whereas all $C(\cdot)$ terms vanish in strictly forward-synchronous unlearning sequences. The explicit forms of $C(\cdot)$ are provided in Proposition E.2 of Appendix E, with the special case $\nu = 0$ treated separately in Corollary E.3.

As established in Proposition 5.2, each unlearning request for a task $s$ processed at time $t_i$ introduces an unavoidable error term $\beta_{i,s}^1$, while the second term in (16) captures the additional approximation error induced by temporal disorder. This term depends on deviations from a forward-synchronous unlearning sequence (Fig. 2), and its magnitude is governed by the coefficients $C(\cdot)$. Consequently, a forward-synchronous unlearning can help the unlearning process.

**Proposition 5.3.** Suppose the loss function $\ell(w, z)$ is Hessian-Lipschitz with constant $L_3$. Choose $\lambda > \max\{\nu - \mu, 0\}$ in Alg. 2. Then the approximation error $\|w_t^{-S_{\leq t}} - w_t^{-\mathbf{S}_{1:t}}\|$ of the unlearning algorithm $\mathcal{R}_\mathcal{A}$ with (14) admits the following second-order upper bound $\gamma_t(\mathbf{S}_{1:t})$:

$$\gamma_t(\mathbf{S}_{1:t}) := C_3 \sum_{\substack{\tau = \min S_{\leq t}+1 \\ \tau \notin S_{\leq t}}}^{t_k} \beta_\tau^1 + \sum_{i=1}^{k-1} \beta_{t_i}^2 + \frac{L}{\lambda} \sum_{s \in S_{\leq t}} \beta_s^3, \quad (17)$$

where $C_3 = \frac{L_3}{2(\mu+\lambda)}$, and

$$\beta_\tau^1 = \rho^{t-\tau-n_{[\tau+1,t_k]}^k} \|w_\tau^{-S_{\leq t}\backslash\{\tau+1,\ldots,t\}} - w_\tau^{-\mathbf{S}_{1:\tau-1}}\|^2,$$
$$\beta_{t_i}^2 = \rho^{t-t_k}(\hat{\rho}^{t_k-t_i-n_{[t_i+1,t_k]}^k} - \rho^{t_k-t_i-n_{[t_i+1,t_k]}^k})\|\bar{\Delta}_{t_i}\|,$$
$$\beta_s^3 = \rho^{t-t_k}(\hat{\rho}^{t_k-s-n_{[s+1,t_k]}^k} - \rho^{t_k-s-n_{[s+1,t_k]}^k}).$$

As established by the preceding results, the primary advantage of the Hessian-based approach in Alg. 2 over the natural forgetting mechanism in Alg. 1 lies in its superior second-order approximation accuracy. In particular, Proposition 5.3 shows that, in the locally convex case, the Hessian-based method admits a tighter second-order upper bound of the approximation error on its dominant term $\beta_i^1$. When $\beta_i^1$ is smaller than one, the Hessian-based update reduces the error at a quadratic rate, leading to a substantial improvement in unlearning accuracy compared to first-order methods.

Analogous to Theorem 4.1, we show that Alg. 2 guarantees the certified unlearning performance in Corollary E.1.

While Alg. 2 attains a lower post-unlearning excess risk than Alg. 1, it incurs additional storage overhead of order $O(t\hat{d})$ by time $t$, due to the need to store approximate Hessian information of dimension $\hat{d}$. In contrast, Alg. 1 requires zero storage. To mitigate this cost while retaining a small

post-unlearning excess risk, we next integrate the natural forgetting of Alg. 1 into the Hessian-based unlearning, yielding a more storage-efficient unlearning strategy.

### 5.3. Forgetting enhanced Hessian-based algorithm

Intuitively, by integrating the natural forgetting effect inherent in CL with Hessian-based techniques, we can leverage natural forgetting to handle earlier tasks while reserving Hessian-based methods for more recent unlearning requests. A key component of this hybrid approach is determining a temporal threshold that governs the transition between the two mechanisms. Motivated by Proposition 5.2, which demonstrates that the disorder term $C(\cdot, \cdot, \cdot)$ vanishes when unlearning requests are well ordered, we propose the following modification to the unlearning algorithm $\mathcal{R}_\mathcal{A}$ in (13).

*Define*

$$S_t' = \{s \in S_t : s > \max((U_t \setminus \{t\}) \cup \{0\})\}.$$

*Update* $w_t^{-\mathbf{S}_{1:t}}$:

$$w_t^{-\mathbf{S}_{1:t}} = w_t^{-\mathbf{S}_{1:t-1}} + \sum_{\substack{s \in S_t'}} \prod_{\substack{i=s+1, \\ i \notin S_{\leq t}}}^{t} ((\hat{H}_i + \lambda I)^{-1} \lambda I) \Delta_s. \quad (18)$$

*Then, discard the stored information of $\hat{H}_i$ and $\Delta_i$ on all tasks $i$ before $t$.*

Under this modification, the storage of Alg. 2 reduces to $O\left(\max_{t_i, t_{i-1} \in U_t} (t_i - t_{i-1})\hat{d}\right)$, based on the maximum distance between any two consecutive unlearning times.

**Proposition 5.4.** *Let $U_t = \{t_1, \ldots, t_k\}$ be the set of time indices at which unlearning requests were received. Choose $\lambda > \max\{0, \nu - \mu\}$. Then, the approximation error $\|w_t^{-S_{\leq t}} - w_t^{-\mathbf{S}_{1:t}}\|$ of the unlearning operator $\mathcal{R}_\mathcal{A}$ in (18) is bounded by*

$$\gamma_t(\mathbf{S}_{1:t}) := \sum_{i=1}^{k} \sum_{s \in S_{t_i}'} \left( \rho^{t - t_i - n_{[t_i+1, t]}^k} \beta_{i,s}^3 C_2 \right.$$

$$+ \frac{L(\mu + \lambda)}{\lambda\mu} \kappa \rho^{t - s - n_{[s+1, t]}^k} \left( 1 - \rho^{n_{[s+1, t_i]}^k - n_{[s+1, t_i]}^i} \right) \right)$$

$$+ \sum_{i=1}^{k} \sum_{s \in S_{t_i} \setminus S_{t_i}'} \rho^{t - s - n_{[s+1, t]}^k} \frac{L}{\lambda}. \quad (19)$$

*where $\beta_{i,s}^3 = \hat{\rho}^{t_i - s - n_{[s+1, t_i]}^i} - \rho^{t_i - s - n_{[s+1, t_i]}^i}$, and $C_2$ is given in Proposition 5.2.*

The complete theoretical results and the proofs of the modified Alg. 2 is provided in Appendix E.4. We also provide a stronger version of certified unlearning with the model's internal state via Alg. 3 (Appendix), where the performance guarantee is established in Corollary E.11.

## 6. Experimental Validation

In this section, we simulate the CLU process on MNIST, CIFAR-10, and CIFAR-100. The main text focuses on the more challenging CIFAR-100 setting, while additional results, experimental details, and parameter-estimation procedures are provided in Appendix F. For CIFAR-100, we split the training set into $T = 30$ non-i.i.d. tasks and use a pretrained ResNet-18 (He et al., 2016) backbone with a trainable three-layer classification head. We compare natural forgetting in Alg. 1, Hessian-based unlearning in Alg. 2 using Gauss–Newton and diagonal Hessian approximations, their forgetting-enhanced variants in (18), and full retraining.

**Effect of unlearning order.** We first compare two request sequences that delete the same set of CIFAR-100 tasks but in different orders: a forward-synchronous (Fwd-Sync) sequence and an asynchronous (Async) sequence as illustrated in Fig. 2. The detailed unlearning sequences are listed in Table 4 of Appendix. Fig. 3 shows that under both Gauss-Newton and diagonal Hessian approximations, the approximation error of two unlearning sequences coincide before time 15, when their deletion histories are identical. In the Gauss-Newton Hessian case, from time 17 onward, Fwd-Sync has already deleted more tasks than Async but nevertheless maintains a smaller approximation error. By the final task, the Async error is about twice the Fwd-Sync error. For the diagonal Hessian, the approximation error of Fwd-Sync can be no smaller at intermediate times such as 15 and 25 because it has processed more deletions by then. Once the two schedules have removed the same set of tasks, Fwd-Sync again gives the smaller error. This supports our Proposition 5.2: well-ordered deletion requests remove the temporal-disorder term in the bound, while asynchronous requests accumulate extra error.

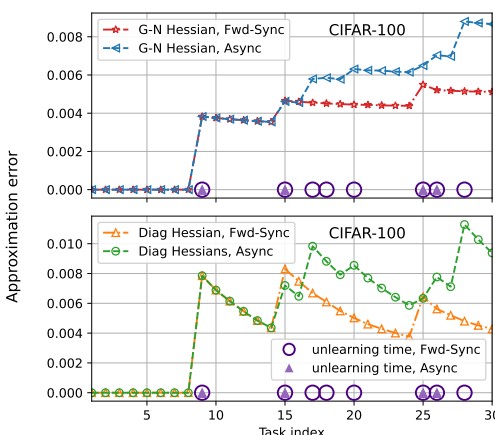

*Figure 3.* Approximation error $\|w_t^{-S_{\leq t}} - w_t^{-\mathbf{S}_{1:t}}\|$ during the CLU process on CIFAR-100 for Fwd-Sync and Async unlearning sequences. The upper and lower panels correspond to the Gauss-Newton Hessian and diagonal Hessian, respectively.

*Table 1.* Runtime, memory, and final empirical approximation error $\|w_t^{-S_{\leq t}} - w_t^{-\mathbf{S}_{1:t}}\|$ on CIFAR-100.

| Metric | Retraining | G–N Hessian | Enhanced G–N | Diag Hessian | Enhanced diag | Natural forgetting |
|---|---|---|---|---|---|---|
| Time (s) ↓ | 950.69 | 269.11 | 264.84 | 283.69 | 277.91 | 260.77 |
| Peak GPU Mem. (MB) ↓ | 932.10 | 5678.90 | 2232.40 | 1129.66 | 1113.52 | 974.30 |
| Empirical approximation error ↓ | 0 | 0.0087 | 0.0463 | 0.0094 | 0.0451 | 0.0835 |

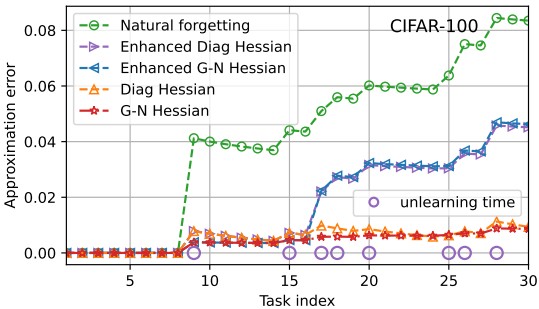

*Figure 4.* Approximation error across the CIFAR-100 CLU process under the Async schedule in Table 4.

**Efficiency and approximation error.** We next compare runtime, peak GPU memory, and final approximation error under the Async CIFAR-100 sequence. Fig. 4 shows that natural forgetting in Alg. 1 has the largest error throughout the process. The pure Hessian-based methods in Alg. 2 are closest to retraining, while the forgetting-enhanced variants in (18) trade a modest increase in error for lower memory usage. Table 1 quantifies the computation-storage trade-off. Gauss-Newton and diagonal Hessian-based unlearning both achieve approximation errors of around 0.009. The Gauss-Newton approximation is faster, but incurs the largest peak memory usage because it stores Jacobian information with complexity $O(Tvd)$, where $v = 100$ and $d = 419684$. In contrast, the diagonal variant stores only $O(Td)$ Hessian information, reducing memory usage by nearly a factor of five, but is slower due to the computation of diagonal second-order information. The forgetting-enhanced variants discard older Hessian information and further reduce memory usage, at the cost of increasing the final error to about 0.045. Full retraining has zero approximation error but is roughly three times slower than the unlearning methods, confirming that our Hessian-based CLU provides a practical middle ground between exact retraining and storage-free natural forgetting.

**Certified noise and final utility.** Finally, we instantiate the theoretical upper bounds $\gamma_T$ from Theorem 4.1 and Proposition 5.4, and calibrate Gaussian noise accordingly to output the publish model $\tilde{w}_T^{-\mathbf{S}_{1:T}}$ that achieves certified unlearning. Table 2 confirms that the computed $\gamma_T$ values upper-bound the empirical approximation errors. Fig. 5 then reports the final test accuracy under $\tilde{w}_T^{-\mathbf{S}_{1:T}}$ with noise added according to $\gamma_T$ and empirical $\|w_t^{-S_{\leq t}} - w_t^{-\mathbf{S}_{1:t}}\|$, where the latter indicates the utility limit if the bound were tight. It shows that Hessian-based methods retain accuracy

*Table 2.* Theoretical upper bound $\gamma_T$ of $\|w_t^{-S_{\leq t}} - w_t^{-\mathbf{S}_{1:t}}\|$ and its empirical value on CIFAR-100.

| Metric | Retrain. | G–N Hess. | Diag Hess. | Nat. Forget. |
|---|---|---|---|---|
| $\gamma_T$ | 0 | 0.0222 | 0.0227 | 0.0899 |
| Emp. error | 0 | 0.0040 | 0.0037 | 0.0465 |

close to retraining under both calibrations, whereas natural forgetting is much more sensitive to bound looseness.

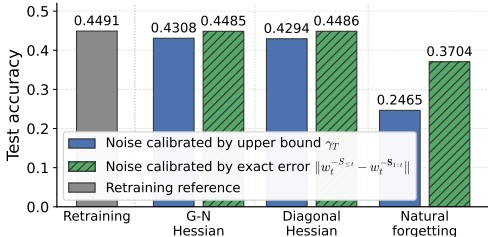

*Figure 5.* Final CIFAR-100 test accuracy of the published model $\tilde{w}_T^{-\mathbf{S}_{1:T}}$. Gaussian noise is calibrated using either the theoretical upper bound $\gamma_T$ from Theorem 4.1 and Proposition 5.4, or the measured exact approximation error.

## 7. Conclusion

We establish a theoretical foundation for certified unlearning in regularization-based continual learning by formulating CLU through a post-unlearning excess risk objective, which decomposes into the CL excess risk and the unlearning loss. This decomposition reveals a fundamental tension between knowledge retention and targeted forgetting: the regularization that improves continual learning performance may not minimize the error required for certified unlearning. Within this framework, we adapt both gradient-based and Hessian-based certified unlearning methods to the continual setting. The former exploits natural forgetting with no additional storage, while the latter enables more accurate targeted unlearning at the cost of curvature storage. We further develop storage-efficient Hessian-based variants that balance computation, storage, and unlearning accuracy.

This work focuses on the fundamental $\ell_2$-regularized CL algorithm, which enables a clean theoretical characterization of certified unlearning in CL. An important direction for future work is to extend the analysis to more sophisticated CL algorithms, including memory-based methods and richer regularization-based approaches such as Elastic Weight Consolidation (EWC).

## Acknowledgments

This work was supported in part by the Guangdong Provincial Key Lab of Integrated Communication, Sensing and Computation for Ubiquitous Internet of Things (No. 2023B1212010007).

## Impact Statement

This paper presents work whose goal is to advance the field of machine learning. There are many potential societal consequences of our work, none of which we feel must be specifically highlighted here.

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

## A. Related works

In continual learning, an increasing number of recent works have begun to investigate the theoretical performance of forgetting and generalization loss induced by CL algorithms ((Evron et al., 2022; Lin et al., 2023; Swartworth et al., 2023; Zhao et al., 2024; Deng et al., 2025)). Due to the inherent difficulty of the CL problem, most prior work has focused on simple regularization-based CL algorithms and linear models for the ease of deriving theoretical results.

Current studies that examine CL and unlearning together are experimental, lacking theoretical guarantees ((Liu et al., 2022; Chatterjee et al., 2024; Cha et al., 2024; Huang et al., 2025)). For example, Chatterjee et al. (2024) uses a controlled knowledge distillation where a CL teacher preserves past knowledge while another unlearning teacher drives targeted forgetting, both guiding a student model to update. Huang et al. (2025) proposes a gradient descent approach that achieves both unlearning and CL by designing a combined loss function, while assuming that the dataset to be unlearned is available again at the time of the unlearning request. These methods merely forget the designated tasks, whereas certified continual unlearning demands a strict approximation of the model that would result from perfect retraining. Furthermore, the efficacy of non-certified unlearning is probabilistic; recent studies reveal that it is still possible to recover supposedly forgotten information even after applying current non-certified techniques (Hu et al., 2025). This highlights the need to develop and rigorously analyze unlearning algorithms for CL.

Basaran et al. (2025) study certified unlearning without access to the original dataset by using a surrogate dataset. However, their work is limited to the static learning case, and their method cannot be adapted to continual learning, where the model evolves with more complex dynamics.

## B. Notation and Conventions

Throughout the paper, we use the following conventions.

For any matrix sequence $P_1, \ldots, P_T$, products are ordered from right to left:

$$\prod_{n=i}^{j} P_n := P_j P_{j-1} \cdots P_i, \qquad i \leq j.$$

When the index set is empty, the product is the identity matrix:

$$\prod_{n=i}^{j} P_n := I, \qquad i > j.$$

Similarly, empty sums are defined as zero:

$$\sum_{n=i}^{j} F_n := 0, \qquad i > j.$$

Although Alg. 2 computes $\bar{\Delta}_t$ only when $S_t \neq \emptyset$, we allow $\bar{\Delta}_t$ to be referenced at any time $t$ by adopting the convention

$$\bar{\Delta}_t = \mathbf{0} \qquad \text{whenever } S_t = \emptyset.$$

Whenever a displayed bound contains factors $1/\mu$ or $1/(\mu - \nu)$, we assume $\mu \neq 0$ and $\mu \neq \nu$, respectively. The degenerate cases can be handled by keeping the corresponding finite geometric sums without rewriting them in closed form.

For the retrained model $w_a^{S_{\leq b}}$, if $a < b$, tasks deleted after time $a$ do not affect the model at time $a$, then

$$w_a^{S_{\leq b}} = w_a^{S_{\leq b} \setminus \{a+1, \ldots, b\}}.$$

## C. Missing details from Section 3

### C.1. Proof of Theorem 3.1

Without loss of generality, we prove the results on the sequence $\{1, \ldots, t\}$, which can be generalized to any retraining sequence $\{\tau_1, \ldots, \tau_{N_t}\} = \{1, \ldots, t\} \setminus S_{\leq t}$. Define $\hat{F}_t(w) = \sum_{z \in D_t} \ell(w, z)/n_t$ as the empirical risk. We prove Theorem

3.1 using Lemmas C.1 and C.2:

$$\mathbb{E}[\frac{1}{t}\sum_{\tau=1}^{t}F_\tau(w_t) - \min_w \frac{1}{t}\sum_{\tau=1}^{t}F_\tau(w)] \leq \mathbb{E}[\frac{1}{t}\sum_{\tau=1}^{t}F_\tau(w_t) - \frac{1}{t}\sum_{\tau=1}^{t}\min_w F_\tau(w)]$$

$$= \mathbb{E}[\frac{1}{t}\sum_{\tau=1}^{t}F_\tau(w_t) - \frac{1}{t}F_\tau(\hat{w}_\tau) + \frac{1}{t}F_\tau(\hat{w}_\tau) - \sum_{\tau=1}^{t}\min_w \frac{1}{t}F_\tau(w)]$$

$$\leq \mathbb{E}\left[\frac{1}{t}\sum_{\tau=1}^{t}L\|w_t - \hat{w}_\tau\| + \frac{4B}{t}\sqrt{\frac{2d\ln(\frac{2Lr\sqrt{n_\tau}}{B}+1)}{n_\tau}} + \frac{2B}{t\sqrt{n_\tau}}\right], \quad (20)$$

where the last line follows from the $L$-Lipschitz property of $F_t$ and Lemma C.1. Substituting Lemma C.2 and rearranging terms, we obtain

$$\sum_{\tau=1}^{t}\|w_t - \hat{w}_\tau\| \leq \sum_{m=1}^{t}\left(\sum_{\substack{i=1\\i\neq m}}^{t}2r\kappa\rho^{t-i} + \rho^t r\right)$$

$$= \sum_{m=1}^{t}\left(\sum_{i=1}^{t}2r\kappa\rho^{t-i}\right) + t\rho^t r - \sum_{m=1}^{t}2r\kappa\rho^{t-m}$$

$$= 2tr\kappa\frac{1-\rho^t}{1-\rho} + t\rho^t r - 2r\kappa\frac{1-\rho^t}{1-\rho}$$

$$= (t-1)2r\kappa\frac{1-\rho^t}{1-\rho} + t\rho^t r.$$

Substituting this bound into (20) completes the proof.

**Lemma C.1.** *Let $w_t^*$ and $\hat{w}_t$ be the population-optimal model and empirical risk minimizer in $\mathcal{W}$ respectively on dataset $D_t$ with size $n_t = |D_t|$. Let $B$ be the upper bound of loss function:*

$$B = \sup_{w\in\mathcal{W},z}|\ell(w,z)|.$$

*Then we have:*

$$\mathbb{E}[F_t(\hat{w}_t)] - \min_w F_t(w) \leq 4B\sqrt{\frac{2d\ln(\frac{2Lr\sqrt{n_t}}{B}+1)}{n_t}} + \frac{2B}{\sqrt{n_t}}.$$

*Proof.* Let $w_t^* \in \arg\min_w F_t(w)$. By the definition of $\hat{w}_t$,

$$F_t(\hat{w}_t) - F_t(w_t^*) = F_t(\hat{w}_t) - \hat{F}_t(\hat{w}_t) + \hat{F}_t(\hat{w}_t) - \hat{F}_t(w_t^*) + \hat{F}_t(w_t^*) - F_t(w_t^*)$$

$$\leq F_t(\hat{w}_t) - \hat{F}_t(\hat{w}_t) + \hat{F}_t(w_t^*) - F_t(w_t^*).$$

Taking expectation over $D_t$, the second term vanishes since $w_t^*$ is fixed. Hence

$$\mathbb{E}[F_t(\hat{w}_t) - F_t(w_t^*)] \leq \mathbb{E}\left[\sup_{w\in\mathcal{W}}\left(F_t(w) - \hat{F}_t(w)\right)\right] \leq 2\mathbb{E}_{D_t\sim\mathcal{D}_t^{n_t}}[R(\ell\circ\mathcal{W}\circ D_t)], \quad (21)$$

where the last inequality follows from Lemma 26.2 in (Shalev-Shwartz & Ben-David, 2014), and $R(l\circ\mathcal{W}\circ D_t)$ is the Rademacher complexity of $\ell\circ\mathcal{W} = \{z \to \ell(w,z) : w\in\mathcal{W}\}$.

We next upper bound $\mathbb{E}_{D_t\sim\mathcal{D}_t^{n_t}}[R(\ell\circ\mathcal{W}\circ D_t)]$. Consider the set of loss vectors $A$ conditioned on the fixed dataset $D_t$:

$$A = \{a \in \mathbb{R}^{n_t} : a = (\ell(w,z_1),\ldots,\ell(w,z_{n_t})), w\in\mathcal{W}\}.$$

Since any loss function is upper bounded by $B$, the $\ell_2$-norm of any element $a \in A$ is bounded by $\sqrt{n_t}B$.

Let $A'$ be a minimal $\zeta$-covering set of $A$ with respect to the $\ell_2$-norm, such that $|A'| = \mathcal{N}(\zeta, A)$. We ensure that every element $a' \in A'$ satisfies the same norm bound, i.e., $\|a'\|_2 \leq \sqrt{n_t}B$. This is guaranteed to exist because projecting any point in a covering set onto the ball $\mathcal{B}(0, \sqrt{n_t}B)$ reduces the distance to the elements of $A$ (which lie within the ball), thereby preserving the $\zeta$-covering property. Furthermore, since $A$ is bounded, the covering number $\mathcal{N}(\zeta, A)$ is finite.

Define the projection mapping $\pi : A \to A'$ such that $\pi(a) \in \arg\min_{a' \in A'} \|a - a'\|_2$. By definition, we have $\|a - \pi(a)\|_2 \leq \zeta$ for all $a \in A$. Then, by the definition of Rademacher complexity:

$$
\begin{aligned}
R(\ell \circ \mathcal{W} \circ D_t) &= \frac{1}{n_t} \mathbb{E}_{\boldsymbol{\sigma}} \left[ \sup_{w \in \mathcal{W}} \sum_{i=1}^{n_t} \sigma_i \ell(w, z_i) \right] \\
&= \frac{1}{n_t} \mathbb{E}_{\boldsymbol{\sigma}} \left[ \sup_{a \in A} \langle a, \boldsymbol{\sigma} \rangle \right] \\
&= \frac{1}{n_t} \mathbb{E}_{\boldsymbol{\sigma}} \left[ \sup_{a \in A} \left( \langle \pi(a), \boldsymbol{\sigma} \rangle + \langle a - \pi(a), \boldsymbol{\sigma} \rangle \right) \right] \\
&\leq \frac{1}{n_t} \mathbb{E}_{\boldsymbol{\sigma}} \left[ \sup_{a' \in A'} \langle a', \boldsymbol{\sigma} \rangle \right] + \frac{1}{n_t} \mathbb{E}_{\boldsymbol{\sigma}} \left[ \sup_{a \in A} \langle a - \pi(a), \boldsymbol{\sigma} \rangle \right] \\
&\leq R(A' \circ D_t) + \frac{\zeta}{\sqrt{n_t}}.
\end{aligned}
$$

By Massart's Lemma (Lemma 26.8 in (Shalev-Shwartz & Ben-David, 2014)), the Rademacher complexity of the finite set $A'$ is bounded by:

$$
R(A' \circ D_t) \leq \frac{\max_{a' \in A'} \|a' - \bar{a}'\|_2 \sqrt{2 \ln |A'|}}{n_t},
$$

where $\bar{a}' = \sum_{a \in A'} \frac{a}{|A'|}$. As established, $\|a'\|_2 \leq \sqrt{n_t}B$ (where $B$ is the maximum scalar loss). Substituting this norm bound:

$$
R(A' \circ D_t) \leq 2B \sqrt{\frac{2 \ln N(\zeta, A)}{n_t}}.
$$

Next, we relate the covering number of the loss vectors to the parameter space. For any $w, w' \in \mathcal{W}$, since $\ell(\cdot, z)$ is $L$-Lipschitz for every $z$,

$$
\|(\ell(w, z_1), \ldots, \ell(w, z_{n_t})) - (\ell(w', z_1), \ldots, \ell(w', z_{n_t}))\|_2 \leq L\sqrt{n_t}\|w - w'\|_2.
$$

Therefore,

$$
N(\zeta, A) \leq N\left( \frac{\zeta}{L\sqrt{n_t}}, \mathcal{W}, \| \cdot \|_2 \right).
$$

Since $\mathcal{W} = \mathcal{B}(w_0, r)$, by Corollary 4.2.11 in (Vershynin, 2026) and translation invariance of covering numbers,

$$
N\left( \frac{\zeta}{L\sqrt{n_t}}, \mathcal{W}, \| \cdot \|_2 \right) \leq \left( 1 + \frac{2Lr\sqrt{n_t}}{\zeta} \right)^d.
$$

Then:

$$
R(\ell \circ \mathcal{W} \circ D_t) \leq 2B \sqrt{\frac{2d \ln(1 + \frac{2Lr\sqrt{n_t}}{\zeta})}{n_t}} + \frac{\zeta}{\sqrt{n_t}}.
$$

Substituting this back into (21) and choosing $\zeta = B$, the expected excess risk is bounded by:

$$
4B \sqrt{\frac{2d \ln(\frac{2Lr\sqrt{n_t}}{B} + 1)}{n_t}} + \frac{2B}{\sqrt{n_t}}. \tag{22}
$$

This completes the proof. $\qquad\square$

**Lemma C.2.** *Let $w_t$ denote the model obtained by running ($\ell_2$-CL) on tasks $\mathbf{D}_{1:t}$ in the CL setting. Let $\hat{w}_t \in W_t \cap \operatorname{int}(\mathcal{W})$ be any empirical risk minimizer in $\mathcal{B}(w_0, r)$. Define*

$$\rho := \frac{\lambda}{\lambda + \mu}, \qquad \kappa := \max\left\{\frac{M}{M + \lambda}, \frac{|\mu|}{\mu + \lambda}\right\}.$$

*Then we have the following for any $m \in [t]$:*

$$\mathbb{E}[\|w_t - \hat{w}_m\|] \leq \sum_{i=1, i\neq m}^{t} 2r\kappa\rho^{t-i} + \rho^t r.$$

*Proof.* Since $\mathcal{W}$ is open, all minimizers and deterministic internal states considered in the analysis are interior points of $\mathcal{W}$, hence the first-order optimality conditions hold. Moreover, since $\mathcal{W} = \mathcal{B}(w_0, r)$ is convex, the line segments used in the Taylor expansions remain in $\mathcal{W}$, where Assumption 2.3 applies.

According to the training algorithm in ($\ell_2$-CL), we have

$$\nabla \hat{F}_t(w_t) + \lambda(w_t - w_{t-1}) = 0.$$

Expanding $\nabla \hat{F}_t(w_t)$ around $\hat{w}_t$ by Taylor's theorem gives

$$\nabla \hat{F}_t(\hat{w}_t) + \int_0^1 \nabla^2 \hat{F}_t(\hat{w}_t + u(w_t - \hat{w}_t))du(w_t - \hat{w}_t) + \lambda(w_t - w_{t-1}) = 0$$

$$\implies w_t = \left(\int_0^1 \nabla^2 \hat{F}_t(\hat{w}_t + u(w_t - \hat{w}_t))du + \lambda I\right)^{-1}\left(\int_0^1 \nabla^2 \hat{F}_t(\hat{w}_t + u(w_t - \hat{w}_t))du\, \hat{w}_t + \lambda w_{t-1}\right),$$

where the last equality uses $\nabla \hat{F}_t(\hat{w}_t) = \mathbf{0}$ since $\hat{w}_t$ is the minimizer of $\hat{F}_t$. Denote

$$\tilde{H}_t = \int_0^1 \nabla^2 \hat{F}_t(\hat{w}_t + u(w_t - \hat{w}_t))du.$$

According to the iteration above, we can write $w_t$ as follows:

$$
\begin{aligned}
w_t =& (\tilde{H}_t + \lambda I)^{-1}(\tilde{H}_t \hat{w}_t + \lambda w_{t-1}) \\
=& (\tilde{H}_t + \lambda I)^{-1}\tilde{H}_t \hat{w}_t + (\tilde{H}_t + \lambda I)^{-1}\lambda w_{t-1} \\
=& (\tilde{H}_t + \lambda I)^{-1}\tilde{H}_t \hat{w}_t + (\tilde{H}_t + \lambda I)^{-1}\lambda I(\tilde{H}_{t-1} + \lambda I)^{-1}(\tilde{H}_{t-1}\hat{w}_{t-1} + \lambda w_{t-2}) \\
=& (\tilde{H}_t + \lambda I)^{-1}\tilde{H}_t \hat{w}_t + (\tilde{H}_t + \lambda I)^{-1}\lambda I(\tilde{H}_{t-1} + \lambda I)^{-1}\tilde{H}_{t-1}\hat{w}_{t-1} + \cdots + \prod_{i=1}^{t}((\tilde{H}_i + \lambda I)^{-1}\lambda I)w_0 \\
=& \sum_{i=1}^{t}\prod_{j=i+1}^{t}((\tilde{H}_j + \lambda I)^{-1}\lambda I)(\tilde{H}_i + \lambda I)^{-1}\tilde{H}_i\hat{w}_i + \prod_{j=1}^{t}((\tilde{H}_j + \lambda I)^{-1}\lambda I)w_0,
\end{aligned}
$$

Then, for any $m$, we have

$$
\begin{aligned}
&\mathbb{E}[\|w_t - \hat{w}_m\|] \\
=& \mathbb{E}\left[\left\|\sum_{i=1, i\neq m}^{t}\left(\prod_{j=i+1}^{t}(\tilde{H}_j + \lambda I)^{-1}\lambda I\right)(\tilde{H}_i + \lambda I)^{-1}\tilde{H}_i(\hat{w}_i - \hat{w}_m) + \left(\prod_{j=1}^{t}(\tilde{H}_j + \lambda I)^{-1}\lambda I\right)(w_0 - \hat{w}_m)\right\|\right] \\
\leq& \mathbb{E}\left[\sum_{i=1, i\neq m}^{t}\left\|\left(\prod_{j=i+1}^{t}(\tilde{H}_j + \lambda I)^{-1}\lambda I\right)(\tilde{H}_i + \lambda I)^{-1}\tilde{H}_i(\hat{w}_i - \hat{w}_m)\right\| + \left\|\left(\prod_{j=1}^{t}(\tilde{H}_j + \lambda I)^{-1}\lambda I\right)(w_0 - \hat{w}_m)\right\|\right] \\
\leq& \sum_{i=1, i\neq m}^{t}\kappa\rho^{t-i}\mathbb{E}[\|\hat{w}_i - \hat{w}_m\|] + \rho^t\mathbb{E}[\|w_0 - \hat{w}_m\|] \\
\leq& \sum_{i=1, i\neq m}^{t}2r\kappa\rho^{t-i} + \rho^t r.
\end{aligned}
$$

Here, the first equality follows from

$$\sum_{i=1}^{t} \left( \prod_{j=i+1}^{t} (\tilde{H}_j + \lambda I)^{-1} \lambda I \right) (\tilde{H}_i + \lambda I)^{-1} \tilde{H}_i + \prod_{j=1}^{t} (\tilde{H}_j + \lambda I)^{-1} \lambda I = I.$$

The second inequality is obtained from Assumption 2.3, which implies

$$\|(\tilde{H}_j + \lambda I)^{-1} \lambda I\| \leq \rho, \qquad \|(\tilde{H}_j + \lambda I)^{-1} \tilde{H}_j\| \leq \kappa.$$

Then we complete the proof. $\qquad \square$

## C.2. Examples of Theorem 3.1

For simplicity, assume that the per-task sample sizes are sufficiently large so that the generalization term is negligible. Consider $N_t = 2$ and $M = \mu > 0$, we have $\kappa = \frac{\mu}{\mu+\lambda}$, $\rho = \frac{\lambda}{\mu+\lambda}$ and $\kappa = 1 - \rho$. Based on the proof of Theorem 3.1, the drift component of the bound can be written as

$$\frac{L}{2} \left[ \kappa \|\hat{w}_{\tau_1} - \hat{w}_{\tau_2}\| + \rho^2 \left( \|w_0 - \hat{w}_{\tau_2}\| + \|w_0 - \hat{w}_{\tau_1}\| \right) + \rho\kappa \left( \|\hat{w}_{\tau_1} - \hat{w}_{\tau_2}\| \right) \right]$$

$$= \frac{L}{2} \left[ (1 - \rho^2) \|\hat{w}_{\tau_1} - \hat{w}_{\tau_2}\| + \rho^2 \left( \|w_0 - \hat{w}_{\tau_1}\| + \|w_0 - \hat{w}_{\tau_2}\| \right) \right].$$

Consequently, the value of $\lambda$ that minimizes the excess-risk bound depends on the relative task dissimilarities $\|w_0 - \hat{w}_{\tau_1}\|$, $\|w_0 - \hat{w}_{\tau_2}\|$, and $\|\hat{w}_{\tau_1} - \hat{w}_{\tau_2}\|$. This illustrates that the optimal regularization strength is determined by the geometry of the task sequence.

# D. Missing details from Section 4

## D.1. Proof of Theorem 4.1

We first prove the upper bound on the approximation error $\|w_t - w_t^{-S_{\leq t}}\|$ given by $\gamma_t(\mathbf{S}_{1:t})$ in (11). Write $S_{\leq t} = \{s_1, \ldots, s_n\}$, where the elements are ordered increasingly. If $t \in S_{\leq t}$, then $t = s_n$, and $w_t^{-S_{\leq t}} = w_{t-1}^{-S_{\leq t} \setminus \{t\}}$, since task $t$ is removed when retraining on the remaining tasks $[t] \setminus S_{\leq t}$. Thus,

$$w_t^{-S_{\leq t}} - w_t = w_{t-1}^{-S_{\leq t}} - w_t = w_{s_n-1}^{-S_{\leq t}} - w_{s_n-1} + \Delta_{s_n}, \tag{23}$$

where $\Delta_{s_n} = w_{s_n-1} - w_{s_n}$.

If $t \notin S_{\leq t}$, then by the CL update rule ($\ell_2$-CL), the perfect retraining model $w_t^{-S_{\leq t}}$ and the model $w_t$ satisfy

$$\nabla \hat{F}_t(w_t^{-S_{\leq t}}) + \lambda(w_t^{-S_{\leq t}} - w_{t-1}^{-S_{\leq t}}) = 0, \ \nabla \hat{F}_t(w_t) + \lambda(w_t - w_{t-1}) = 0.$$

Let

$$\tilde{H}_t = \int_0^1 \nabla^2 \hat{F}_t \left( w_t + u(w_t^{-S_{\leq t}} - w_t) \right) du.$$

Expanding $\nabla \hat{F}_t(w_t^{-S_{\leq t}})$ around $w_t$ gives

$$\nabla \hat{F}_t(w_t) + \tilde{H}_t(w_t^{-S_{\leq t}} - w_t) + \lambda(w_t^{-S_{\leq t}} - w_{t-1}^{-S_{\leq t}}) = 0.$$

Using $\nabla \hat{F}_t(w_t) = -\lambda(w_t - w_{t-1})$, we obtain

$$w_t^{-S_{\leq t}} - w_t = (\tilde{H}_t + \lambda I)^{-1} \lambda(w_{t-1}^{-S_{\leq t}} - w_{t-1}).$$

Iterating this relation backward from $t$ to the latest deleted task $s_n$, we have

$$w_t^{-S_{\leq t}} - w_t = \left( \prod_{i=s_n+1}^{t} (\tilde{H}_i + \lambda I)^{-1} \lambda I \right) (w_{s_n}^{-S_{\leq t}} - w_{s_n}).$$

Since task $s_n$ is removed in the retraining trajectory, $w_{s_n}^{-S_{\leq t}} = w_{s_n-1}^{-S_{\leq t}\backslash\{s_n\}}$. Hence

$$w_t^{-S_{\leq t}} - w_t = \left(\prod_{i=s_n+1}^{t} (\tilde{H}_i + \lambda I)^{-1}\lambda I\right)\left(w_{s_n-1}^{-S_{\leq t}\backslash\{s_n\}} - w_{s_n-1}\right) + \left(\prod_{i=s_n+1}^{t} (\tilde{H}_i + \lambda I)^{-1}\lambda I\right)(w_{s_n-1} - w_{s_n}).$$

The case $t = s_n$ in (23) is also covered by the display above under the empty-product convention. Repeating this argument for all deleted tasks $s_j \in S_{\leq t}$ yields

$$w_t^{-S_{\leq t}} - w_t = \prod_{\substack{i=s_1+1 \\ i\notin S_{\leq t}}}^{t} ((\tilde{H}_i + \lambda I)^{-1}\lambda I)\left(w_{s_1-1}^{-S_{\leq t}\backslash\{s_1,\ldots,s_n\}} - w_{s_1-1}\right) + \sum_{j=1}^{n}\prod_{\substack{i=s_j+1 \\ i\notin S_{\leq t}}}^{t} ((\tilde{H}_i + \lambda I)^{-1}\lambda I)(w_{s_j-1} - w_{s_j})$$

$$= \sum_{j=1}^{n}\prod_{\substack{i=s_j+1 \\ i\notin S_{\leq t}}}^{t} ((\tilde{H}_i + \lambda I)^{-1}\lambda I)(w_{s_j-1} - w_{s_j}),$$

where the last equality follows because no task before $s_1$ is deleted, so $w_{s_1-1}^{-S_{\leq t}\backslash\{s_1,\ldots,s_n\}} = w_{s_1-1}$.

Taking norms gives

$$\|w_t^{-S_{\leq t}} - w_t\| \leq \sum_{j=1}^{n}\prod_{\substack{i=s_j+1 \\ i\notin S_{\leq t}}}^{t} \|(\tilde{H}_i + \lambda I)^{-1}\lambda I\| \cdot \|w_{s_j-1} - w_{s_j}\|. \tag{24}$$

By Assumption 2.3,

$$\|(\tilde{H}_i + \lambda I)^{-1}\lambda I\| \leq \frac{\lambda}{\mu + \lambda} =: \rho.$$

Moreover, by the first-order condition of ($\ell_2$-CL), $\lambda(w_{s_j-1} - w_{s_j}) = \nabla \hat{F}_{s_j}(w_{s_j})$, and the $L$-Lipschitzness of the loss implies $\|w_{s_j-1} - w_{s_j}\| \leq \frac{L}{\lambda}$. Substituting these two bounds into (24) gives

$$\|w_t^{-S_{\leq t}} - w_t\| \leq \frac{L}{\lambda}\sum_{i=1}^{k}\sum_{s\in S_{t_i}} \rho^{t-s-n_{[s+1,t]}^k} = \gamma_t(\mathbf{S}_{1:t}),$$

which proves the desired approximation-error bound.

Given this bound, the Gaussian noise mechanism in Alg. 1 ensures $(\varepsilon, \delta)$-certified continual unlearning by the standard Gaussian-mechanism argument for certified unlearning (Dwork et al., 2014; Qiao et al., 2025).

It remains to bound the post-unlearning excess risk. By the decomposition in (7) and (8), it suffices to upper bound the unlearning loss:

$$\mathbb{E}\left[\frac{1}{t - |S_{\leq t}|}\sum_{\substack{\tau=1 \\ \tau\notin S_{\leq t}}}^{t} \left(F_\tau(\tilde{w}_t^{-\mathbf{S}_{1:t}}) - F_\tau(w_t^{-S_{\leq t}})\right)\right] \leq \mathbb{E}\left[L\|\tilde{w}_t^{-\mathbf{S}_{1:t}} - w_t^{-S_{\leq t}}\|\right]$$

$$\leq \mathbb{E}\left[L\|\tilde{w}_t^{-\mathbf{S}_{1:t}} - w_t^{-\mathbf{S}_{1:t}}\|\right] + \mathbb{E}\left[L\|w_t^{-\mathbf{S}_{1:t}} - w_t^{-S_{\leq t}}\|\right]$$

$$\leq L\sqrt{\mathbb{E}\left[\|\tilde{w}_t^{-\mathbf{S}_{1:t}} - w_t^{-\mathbf{S}_{1:t}}\|^2\right]} + L\gamma_t(\mathbf{S}_{1:t})$$

$$= L\gamma_t(\mathbf{S}_{1:t})\frac{\sqrt{2d\ln(1.25/\delta)}}{\varepsilon} + L\gamma_t(\mathbf{S}_{1:t}).$$

Here, the first inequality follows from the $L$-Lipschitzness of $F_\tau$, the third inequality follows from Jensen's inequality and the approximation-error bound, and the last equality follows from the variance of the Gaussian noise in Alg. 1. In Alg. 1, the natural forgetting baseline uses $w_t^{-\mathbf{S}_{1:t}} = w_t$. Combining the unlearning-loss bound above with $\mathcal{E}^{-S_{\leq t}}(\lambda)$ completes the proof.

## D.2. A stronger version of $(\varepsilon, \delta)$-certified unlearning

To further unlearn the model and ensure certified unlearning even at the system side, we extend Alg. 1 by removing all internal information of deleted tasks between iterations. Specifically, we make two changes to Alg. 1:

- After computing $\tilde{w}_t^{-\mathbf{S}_{1:t}} = f\left(w_t^{-\mathbf{S}_{1:t}}\right)$ in line 12, we discard $w_t$ and use $\tilde{w}_t^{-\mathbf{S}_{1:t}}$ instead for training on next task $t + 1$.

- After training on task $t + 1$ starting from noisy $\tilde{w}_t^{-\mathbf{S}_{1:t}}$, the updated model $w_{t+1}^{-\mathbf{S}_{1:t}}$ still satisfies the certified unlearning guarantee relative to the perfectly retraining model $w_{t+1}^{-S_{\leq t}}$, by the post-processing immunity of differential privacy (Dwork et al., 2014). Therefore, we only need to inject noise when a nonempty deletion request $S_t$ arrives, changing the condition in line 5 from $S_{\leq t} = \emptyset$ to $S_t = \emptyset$.

With these modifications, Theorem 4.1 extends to the corollary below for strong privacy protection.

**Corollary D.1.** *Let $\lambda > \max\{0, -\mu\}$, and define $\gamma_t(\mathbf{S}_{1:t})$ below:*

$$\gamma_t(\mathbf{S}_{1:t}) := \sum_{i=1}^{k} \sum_{s \in S_{t_i}} \rho^{t-s-n_{[s+1,t]}^k} \frac{L}{\lambda}(1+\xi)^{k-i+1}, \tag{25}$$

*where $\xi = \dfrac{d\sqrt{4\ln(1.25/\delta)\ln(2dT/\delta_2)}}{\varepsilon}$. With probability at least $1 - \delta_2$, we have $\|w_t^{-S_{\leq t}} - \tilde{w}_t^{-\mathbf{S}_{1:t}}\| \leq \gamma_t(\mathbf{S}_{1:t})$, and the CLU process following modified Alg. 1 satisfies $(\varepsilon, \delta)$-certified continual unlearning as defined in Definition 2.1 with repect to both the public model $\tilde{w}_t^{-\mathbf{S}_{1:t}}$ and the internal state of the system. Moreover, the output model $\tilde{w}_t^{-\mathbf{S}_{1:t}}$ of modified Alg. 1 achieves a post-unlearning excess risk in (6) that is upper bounded by the following expression with probability at least $1 - \delta_2$:*

$$L\gamma_t(\mathbf{S}_{1:t})\left(\frac{\sqrt{2d\ln(\frac{1.25}{\delta})}}{\varepsilon} + 1\right) + \mathcal{E}^{-S_{\leq t}}(\lambda),$$

*where $\mathcal{E}^{-S_{\leq t}}(\lambda)$ is given in Theorem 3.1, and $\gamma_t(\mathbf{S}_{1:t})$ is in (25).*

$\gamma_t(\mathbf{S}_{1:t})$ in (25) is a high-probability upper bound that carries an extra multiplicative factor $(1 + \xi)^{k-i+1}$ relative to (11). This arises because, in our stronger privacy protection, we feed the noised model $\tilde{w}_t^{-\mathbf{S}_{1:t}}$ into the next task, which, in turn, increases the required noise to mask the unlearning loss at next unlearning time, so each round's injected noise accumulates and increases over the learning–unlearning cycles, adding extra random deviation.

# E. Missing details from Section 5

**Corollary E.1.** *Alg. 2, using the noise coefficient $\gamma_t(\mathbf{S}_{1:t})$ defined in (16) or (17), guarantees $(\varepsilon, \delta)$-certified continual unlearning in Definition 2.1. Moreover, the output model $\tilde{w}_t^{-\mathbf{S}_{1:t}}$ produced by Alg. 2 achieves the post-unlearning excess risk $\mathcal{E}_{\mathrm{LU}}$ in (12), with $\mathcal{E}^{-S_{\leq t}}(\lambda)$ given in Theorem 3.1 and $\gamma_t(\mathbf{S}_{1:t})$ specified in (16) or (17).*

## E.1. Choice of Hessian approximation

In modern deep learning regimes, the parameter dimension $d$ often scales to millions or billions, rendering the $O(d^2)$ storage and $O(d^2)$ computation of the exact Hessian matrix prohibitively expensive. Consequently, practical algorithms relying on second-order information must employ numerical approximations to balance computational tractability with the fidelity of curvature estimation. Next we present and discuss some commonly used Hessian approximation methods.

- *Diagonal Approximation.* The most computationally efficient approach restricts the Hessian to its diagonal elements, assuming independence between parameters. This reduces the storage from $O(d^2)$ to $O(d)$, and inversion complexity from $O(d^3)$ to $O(d)$. This method is widely adapted in CL (e.g., EWC (Kirkpatrick et al., 2017)) and adaptive optimization methods like Adam and Adagrad. While scalable, diagonal approximations discard all covariance information between parameters, potentially leading to poor unlearning performance in scenarios where parameter correlations are significant (e.g., in the shared representations of deep networks).

---

**Algorithm 2** Hessian-based CLU

---

1: Initialize $U_0 = \emptyset$, $S_{\leq 0} = \emptyset$, and $\mathbf{S}_{1:0} = \emptyset$.
2: **for** $t = 1$ to $T$ **do**
    **Stage I. learning and precomputation on task** $t$
3:    Receive $D_t$, update $w_t^{-\mathbf{S}_{1:t-1}} \leftarrow \ell_2\text{-CL}(w_{t-1}^{-\mathbf{S}_{1:t-1}}, D_t)$.
4:    Compute and store $\Delta_t$ and $\hat{H}_t$ defined in (15).
    **Stage II. system unlearning**
5:    Receive deletion request $S_t \subseteq [t] \setminus S_{\leq t-1}$.
6:    **if** $S_t \neq \emptyset$ **then**
7:        Update $S_{\leq t} \leftarrow S_t \cup S_{\leq t-1}$, $\mathbf{S}_{1:t} \leftarrow (\mathbf{S}_{1:t-1}, S_t)$, and $U_t \leftarrow \{t\} \cup U_{t-1}$.
8:        Set $w_t^{-\mathbf{S}_{1:t}} \leftarrow \mathcal{R}_{\mathcal{A}}\left(w_t^{-\mathbf{S}_{1:t-1}}, \mathbf{D}_{1:t}, \mathbf{S}_{1:t}\right) = w_t^{-\mathbf{S}_{1:t-1}} + \bar{\Delta}_t$ with $\bar{\Delta}_t$ in (14).
9:    **else**
10:       Set $S_{\leq t} \leftarrow S_{\leq t-1}$, $\mathbf{S}_{1:t} \leftarrow \mathbf{S}_{1:t-1}$, $U_t \leftarrow U_{t-1}$, and $w_t^{-\mathbf{S}_{1:t}} \leftarrow w_t^{-\mathbf{S}_{1:t-1}}$.
11:    **end if**
    **Stage III. publishing model**
12:    **if** $S_{\leq t} \neq \emptyset$ **then**
13:       Draw $\epsilon_t \sim \mathcal{N}(0, \sigma^2 I)$ with $\sigma = \gamma_t(\mathbf{S}_{1:t}) \frac{\sqrt{2\ln(1.25/\delta)}}{\varepsilon}$ and $\gamma_t(\mathbf{S}_{1:t})$ defined in (16).
14:       **Output** $\tilde{w}_t^{-\mathbf{S}_{1:t}} \leftarrow w_t^{-\mathbf{S}_{1:t}} + \epsilon_t$.
15:    **else**
16:       **Output** $\tilde{w}_t^{-\mathbf{S}_{1:t}} \leftarrow w_t^{-\mathbf{S}_{1:t}}$.
17:    **end if**
18: **end for**

---

- *Block-Diagonal and Kronecker-Factored Approximations (K-FAC)* (Martens & Grosse, 2015). To capture parameter correlations without the full cost of the exact Hessian, block-diagonal approximations treat parameters in different layers as independent while preserving correlations within layers. A prominent instance is the K-FAC. K-FAC approximates the Fisher Information Matrix (often used as a proxy for the Hessian) by decomposing layer-wise blocks into the Kronecker product of smaller matrices. The inversion cost and storage of K-FAC is $O(\sum_{l=1}^{L} d_{in,l}^3 + d_{out,l}^3)$ and $O(\sum_{l=1}^{L} d_{in,l}^2 + d_{out,l}^2)$, respectively, where $d_{in,l}$ and $d_{out,l}$ are the dimensions of input and output of the $l$-layer of the neural network.

- *Newton-Gaussian* (Nocedal, 2006). The Gauss-Newton method typically applies to models where the final loss function is convex with respect to the neural network's output. It approximates the Hessian as: $G \approx J^\top H_{\text{loss}} J$, where $H_{\text{loss}}$ is the Hessian of the loss function with respect to the network output, and $J$ is the Jacobian matrix of the output with respect to all model parameters. Therefore, the storage of Hessian reduces to store the $H_{loss}$ and $J$ with storage overhead $O(v^2 + dv)$, where $v$ is the output dimension (e.g., the number of classes in image classification). Also, by applying the Woodbury matrix identity, the inversion cost of $J^\top H_{\text{loss}} J + \lambda I$ reduces to $O(d^2 v)$. Since $v$ is usually far less than the parameter dimension $d$ ($v \ll d$), this offers both storage and computational advantage. Moreover, a unique benefit of this method is that the approximated curvature matrix is guaranteed to be positive semi-definite.

### E.2. Proof of Proposition 5.2

**Proposition E.2.** *Let $U_t = \{t_1, \ldots, t_k\}$ be the set of time indices at which unlearning requests were received up to time $t$. Choose $\lambda > \max\{0, \nu - \mu\}$. Then, the approximation error $\|w_t^{-S_{\leq t}} - w_t^{-\mathbf{S}_{1:t}}\|$ of the unlearning operator $\mathcal{R}_{\mathcal{A}}$ in (13) is*

*bounded by* $\gamma_t(\mathbf{S}_{1:t})$:

$$\gamma_t(\mathbf{S}_{1:t}) := \frac{L(M-\mu+\nu)}{\nu\lambda}\rho^{t-t_k}\left[\sum_{s\in S_{t_k}}\left(\hat{\rho}^{t_k-s-n^k_{[s+1,t_k]}} - \rho^{t_k-s-n^k_{[s+1,t_k]}}\right)\right.$$

$$+ \sum_{i=1}^{k-1}\sum_{s\in S_{t_i}}\hat{\rho}^{t_k-s-n^k_{[s+1,t_k]}}\left(\left(\frac{\rho}{\hat{\rho}}\right)^{t_k-t_i-n^k_{[t_i+1,t_k]}}\left(1-\left(\frac{\rho}{\hat{\rho}}\right)^{t_i-s-n^k_{[s+1,t_i]}}\right)\right.$$

$$+ \left.\left.\sum_{x=1}^{k-i}\left(\frac{\rho}{\hat{\rho}}\right)^{t_k-t_{k-x+1}-n^k_{[t_{k-x+1}+1,t_k]}}\left(1-\left(\frac{\rho}{\hat{\rho}}\right)^{t_{k-x+1}-t_{k-x}-n^k_{[t_{k-x+1},t_{k-x+1}]}}\right)C^x_{k-i}(t_k,t_i,s)\right)\right]. \quad (26)$$

*Here* $C^x_{k-i}(t_k,t_i,s)$ *is defined recursively by*

$$C^x_{k-i}(t_k,t_i,s) =$$

$$\begin{cases} C^x_{k-i-1}(t_k,t_{i+1},t_i)\dfrac{\mu+\lambda-\nu}{\mu-\nu}\hat{\kappa}\hat{\rho}^{n^k_{[s+1,t_i]}-n^{i+1}_{[s+1,t_i]}}\left(1-\hat{\rho}^{n^{i+1}_{[s+1,t_i]}-n^i_{[s+1,t_i]}}\right)+C^x_{k-i-1}(t_k,t_i,s), & x=1,\ldots,k-i-1, \\[2ex] C^x_{k-i-1}(t_k,t_{i+1},t_i)\dfrac{\mu+\lambda-\nu}{\mu-\nu}\hat{\kappa}\hat{\rho}^{n^k_{[s+1,t_i]}-n^{i+1}_{[s+1,t_i]}}\left(1-\hat{\rho}^{n^{i+1}_{[s+1,t_i]}-n^i_{[s+1,t_i]}}\right)+C^{x-1}_{k-i-1}(t_k,t_i,s), & x=k-i, \end{cases}$$

$$C^1_1(t_k,y,s) = \frac{\mu+\lambda-\nu}{\mu-\nu}\hat{\kappa}\left(1-\hat{\rho}^{n^k_{[s,y]}-n^{k-1}_{[s,y]}}\right), \quad (27)$$

*with the convention* $C^{m+1}_m(\cdot)=1$.

Proposition E.2 shows that, for a task unlearned at time $t_i$, if at a later time $t_{i+1}$ the system receives a new unlearning request for a task between $s$ and $t_i$ (e.g., $n^{i+1}_{[s+1,t_i]} > n^i_{[s+1,t_i]}$), the additional error term is incurred in $C^{k-i}_{k-i}(t_k,t_i,s)$. If further unlearning requests arrive between $s$ and $t_i$, the parameter $C^l_{k-i}(t_k,t_i,s)$ will continue to grow with $l$ according to the recurrence of $C$ defined in (27).

When we use the exact Hessian in Alg. 2, i.e., $\nu=0$, the result above becomes the following bound obtained by taking the limit with $\nu\to 0$.

**Corollary E.3.** *Choose* $\lambda > \max\{0,-\mu\}$ *in* ($\ell_2$-CL) *and Alg. 2. When* $\nu=0$, *the approximation error* $\|w^{-S_{\leq t}}_t - w^{-\mathbf{S}_{1:t}}_t\|$ *of unlearning algorithm* $\mathcal{R}_{\mathcal{A}}$ *in* (14) *is bounded by* $\gamma_t(\mathbf{S}_{1:t})$ *below:*

$$\gamma_t(\mathbf{S}_{1:t}) := \left[\sum_{s\in S_{t_k}}\rho^{t_k-s-n^k_{[s+1,t_k]}}(t_k-s-n^k_{[s+1,t_k]}) + \sum_{i=1}^{k-1}\sum_{s\in S_{t_i}}\rho^{t_k-s-n^k_{[s+1,t_k]}}\left((t_i-s-n^k_{[s+1,t_i]})\right.\right.$$

$$\left.\left.+ \sum_{x=1}^{k-i}(t_{k-x+1}-t_{k-x}-n^k_{[t_{k-x+1},t_{k-x+1}]})C^x_{k-i}(t_k,t_i,s)\right)\right]\frac{L(M-\mu)}{\lambda(\mu+\lambda)}\rho^{t-t_k}, \quad (28)$$

*where* $C^x_{k-i}(t_k,t_i,s)$ *follows the following iteration:*

$$C^x_{k-i}(t_k,t_i,s)$$

$$= \begin{cases} C^x_{k-i-1}(t_k,t_{i+1},t_i)\dfrac{\mu+\lambda}{\mu}\kappa\rho^{n^k_{[s+1,t_i]}-n^{i+1}_{[s+1,t_i]}}(1-\rho^{n^{i+1}_{[s+1,t_i]}-n^i_{[s+1,t_i]}})+C^x_{k-i-1}(t_k,t_i,s), & x=1,\ldots,k-i-1, \\[2ex] C^x_{k-i-1}(t_k,t_{i+1},t_i)\dfrac{\mu+\lambda}{\mu}\kappa\rho^{n^k_{[s+1,t_i]}-n^{i+1}_{[s+1,t_i]}}(1-\rho^{n^{i+1}_{[s+1,t_i]}-n^i_{[s+1,t_i]}})+C^{x-1}_{k-i-1}(t_k,t_i,s), & x=k-i, \end{cases},$$

$$C^1_1(t_k,y,s) = \frac{\mu+\lambda}{\mu}\kappa(1-\rho^{n^k_{[s,y]}-n^{k-1}_{[s,y]}}) \text{ and } C^{m+1}_m(\cdot)=1. \quad (29)$$

*Proof of Proposition E.2.* By Lemma E.6, for any time $t$ after the $k$-th unlearning request and before the next unlearning request, we have $S_{\leq t} = S_{\leq t_k}$ and $\mathbf{S}_{1:t} = \mathbf{S}_{1:t_k}$. Hence,

$$\left\|w^{-S_{\leq t}}_t - w^{-\mathbf{S}_{1:t}}_t\right\| = \left\|\prod_{a=t_k+1}^t \tilde{P}_a\left(w^{-S_{\leq t_k}}_{t_k} - w^{-\mathbf{S}_{1:t_k}}_{t_k}\right)\right\|$$

$$\leq \rho^{t-t_k}\left\|w^{-S_{\leq t_k}}_{t_k} - w^{-\mathbf{S}_{1:t_k}}_{t_k}\right\|.$$

Therefore, it remains to bound $\|w_{t_k}^{-S_{\leq t_k}} - w_{t_k}^{-\mathbf{S}_{1:t_k}}\|$.

By Lemma E.8,

$$w_{t_k}^{-S_{\leq t_k}} - w_{t_k}^{-\mathbf{S}_{1:t_k}} = \sum_{i=0}^{k-1} \sum_{s \in S_{t_{k-i}}} D_i(t_k, s)\Delta_s.$$

Moreover, the first-order condition of ($\ell_2$-CL) gives $\lambda\big(w_{s-1}^{-\mathbf{S}_{1:s-1}} - w_s^{-\mathbf{S}_{1:s-1}}\big) = \nabla \hat{F}_s(w_s^{-\mathbf{S}_{1:s-1}})$, and the $L$-Lipschitz condition implies $\|\Delta_s\| \leq L/\lambda$. Thus,

$$\left\| w_{t_k}^{-S_{\leq t_k}} - w_{t_k}^{-\mathbf{S}_{1:t_k}} \right\| \leq \frac{L}{\lambda} \sum_{i=0}^{k-1} \sum_{s \in S_{t_{k-i}}} \|D_i(t_k, s)\|. \tag{30}$$

Thus, it suffices to prove

$$\|D_0(t_k, s)\| \leq \frac{M - \mu + \nu}{\nu} \left( \hat{\rho}^{t_k - s - n^k_{[s+1, t_k]}} - \rho^{t_k - s - n^k_{[s+1, t_k]}} \right) \tag{31}$$

and

$$\begin{aligned}
\|D_j(t_k, s)\| \leq & \frac{M - \mu + \nu}{\nu} \hat{\rho}^{t_k - s - n^k_{[s+1, t_k]}} \left[ \sum_{x=1}^{j} \left( \frac{\rho}{\hat{\rho}} \right)^{t_k - t_{k-x+1} - n^k_{[t_{k-x+1}+1, t_k]}} \right. \\
& \cdot \left( 1 - \left( \frac{\rho}{\hat{\rho}} \right)^{t_{k-x+1} - t_{k-x} - n^k_{[t_{k-x}+1, t_{k-x+1}]}} \right) C_j^x(t_k, t_{k-j}, s) \\
& \left. + \left( \frac{\rho}{\hat{\rho}} \right)^{t_k - t_{k-j} - n^k_{[t_{k-j}+1, t_k]}} \left( 1 - \left( \frac{\rho}{\hat{\rho}} \right)^{t_{k-j} - s - n^k_{[s+1, t_{k-j}]}} \right) C_j^{j+1}(t_k, t_{k-j}, s) \right]
\end{aligned} \tag{32}$$

for $j = 1, \ldots, k-1$ and any index $s \leq t_{k-j}$ for which the products below are defined.

(31) can be directly obtained by Lemma E.4 and Lemma E.8.

For $j = 1$, Lemma E.9 gives

$$D_1(t_k, s) = D_0'(t_k, s) + D_0(t_k, t_{k-1}) \left( \prod_{\substack{a=s+1 \\ a \notin S_{\leq t_k}}}^{t_{k-1}} \hat{P}_a - \prod_{\substack{a=s+1 \\ a \notin S_{\leq t_{k-1}}}}^{t_{k-1}} \hat{P}_a \right),$$

where

$$D_0'(t_k, s) = D_0(t_k, s) - D_0(t_k, t_{k-1}) \prod_{\substack{a=s+1 \\ a \notin S_{\leq t_k}}}^{t_{k-1}} \hat{P}_a.$$

Using the product decomposition convention,

$$D_0'(t_k, s) = \prod_{\substack{a=t_{k-1}+1 \\ a \notin S_{\leq t_k}}}^{t_k} \tilde{P}_a \left( \prod_{\substack{a=s+1 \\ a \notin S_{\leq t_k}}}^{t_{k-1}} \tilde{P}_a - \prod_{\substack{a=s+1 \\ a \notin S_{\leq t_k}}}^{t_{k-1}} \hat{P}_a \right).$$

Therefore, by Lemma E.4,

$$\|D_0'(t_k, s)\| \leq \frac{M - \mu + \nu}{\nu} \rho^{t_k - t_{k-1} - n^k_{[t_{k-1}+1, t_k]}} \left( \hat{\rho}^{t_{k-1} - s - n^k_{[s+1, t_{k-1}]}} - \rho^{t_{k-1} - s - n^k_{[s+1, t_{k-1}]}} \right).$$

Also, by (31) and Lemma E.5,

$$
\left\| D_0(t_k, t_{k-1}) \left( \prod_{\substack{a=s+1 \\ a\notin S_{\le t_k}}}^{t_{k-1}} \hat{P}_a - \prod_{\substack{a=s+1 \\ a\notin S_{\le t_{k-1}}}}^{t_{k-1}} \hat{P}_a \right) \right\| \le \frac{M-\mu+\nu}{\nu} \left( \hat{\rho}^{t_k-t_{k-1}-n^k_{[t_{k-1}+1,t_k]}} - \rho^{t_k-t_{k-1}-n^k_{[t_{k-1}+1,t_k]}} \right)
$$

$$
\cdot \frac{\mu+\lambda-\nu}{\mu-\nu} \hat{\kappa} \hat{\rho}^{t_{k-1}-s-n^k_{[s+1,t_{k-1}]}} \left( 1 - \hat{\rho}^{n^k_{[s+1,t_{k-1}]} - n^{k-1}_{[s+1,t_{k-1}]}} \right).
$$

Combining the last two displays yields (32) for $j = 1$, with

$$
C_1^2(t_k, t_{k-1}, s) = 1, \qquad C_1^1(t_k, t_{k-1}, s) = \frac{\mu+\lambda-\nu}{\mu-\nu} \hat{\kappa} \left( 1 - \hat{\rho}^{n^k_{[s+1,t_{k-1}]} - n^{k-1}_{[s+1,t_{k-1}]}} \right).
$$

Assume now that (32) holds for some $j = 1, \ldots, k-2$. By Lemma E.9,

$$
D_{j+1}(t_k, s) = D_j'(t_k, s) + D_j(t_k, t_{k-j-1}) \left( \prod_{\substack{a=s+1 \\ a\notin S_{\le t_{k-j}}}}^{t_{k-j-1}} \hat{P}_a - \prod_{\substack{a=s+1 \\ a\notin S_{\le t_{k-j-1}}}}^{t_{k-j-1}} \hat{P}_a \right), \tag{33}
$$

where

$$
D_j'(t_k, s) = \sum_{x=1}^{j} D_{j-x}(t_k, t_{k-j-1}) \left( \prod_{\substack{a=s+1 \\ a\notin S_{\le t_{k-j+x}}}}^{t_{k-j-1}} \hat{P}_a - \prod_{\substack{a=s+1 \\ a\notin S_{\le t_{k-j+x-1}}}}^{t_{k-j-1}} \hat{P}_a \right)
$$

$$
+ D_0(t_k, s) - D_0(t_k, t_{k-j-1}) \prod_{\substack{a=s+1 \\ a\notin S_{\le t_k}}}^{t_{k-j-1}} \hat{P}_a.
$$

This $D_j'(t_k, s)$ has the same algebraic form as $D_j(t_k, s)$ (32), with $t_{k-j}$ replaced by $t_{k-j-1}$. Hence the induction hypothesis gives the following with $t_{k-j}$ replaced by $t_{k-j-1}$.

$$
\|D_j'(t_k, s)\| \le \frac{M-\mu+\nu}{\nu} \hat{\rho}^{t_k-s-n^k_{[s+1,t_k]}} \left[ \sum_{x=1}^{j-1} \left( \frac{\rho}{\hat{\rho}} \right)^{t_k-t_{k-x+1}-n^k_{[t_{k-x+1}+1,t_k]}} \right.
$$

$$
\cdot \left( 1 - \left( \frac{\rho}{\hat{\rho}} \right)^{t_{k-x+1}-t_{k-x}-n^k_{[t_{k-x}+1,t_{k-x+1}]}} \right) C_j^x(t_k, t_{k-j-1}, s)
$$

$$
+ \left( \frac{\rho}{\hat{\rho}} \right)^{t_k-t_{k-j+1}-n^k_{[t_{k-j+1}+1,t_k]}} \left( 1 - \left( \frac{\rho}{\hat{\rho}} \right)^{t_{k-j+1}-t_{k-j-1}-n^k_{[t_{k-j-1}+1,t_{k-j+1}]}} \right) C_j^j(t_k, t_{k-j-1}, s)
$$

$$
+ \left. \left( \frac{\rho}{\hat{\rho}} \right)^{t_k-t_{k-j-1}-n^k_{[t_{k-j-1}+1,t_k]}} \left( 1 - \left( \frac{\rho}{\hat{\rho}} \right)^{t_{k-j-1}-s-n^k_{[s+1,t_{k-j-1}]}} \right) C_j^{j+1}(t_k, t_{k-j-1}, s) \right]. \tag{34}
$$

On the other hand, applying the induction hypothesis to $D_j(t_k, t_{k-j-1})$, and applying Lemma E.5 to the product difference

in (33), gives the second contribution

$$
\left\| D_j(t_k, t_{k-j-1}) \left( \prod_{\substack{a=s+1 \\ a \notin S_{\le t_{k-j}}}}^{t_{k-j-1}} \hat{P}_a - \prod_{\substack{a=s+1 \\ a \notin S_{\le t_{k-j-1}}}}^{t_{k-j-1}} \hat{P}_a \right) \right\|
$$

$$
\le \frac{M-\mu+\nu}{\nu} \hat{\rho}^{t_k-s-n_{[s+1,t_k]}^k} G_j(s) \left[ \sum_{x=1}^{j} \left(\frac{\rho}{\hat{\rho}}\right)^{t_k-t_{k-x+1}-n_{[t_{k-x+1}+1,t_k]}^k} \right.
$$

$$
\cdot \left(1 - \left(\frac{\rho}{\hat{\rho}}\right)^{t_{k-x+1}-t_{k-x}-n_{[t_{k-x}+1,t_{k-x+1}]}^k}\right) C_j^x(t_k, t_{k-j}, t_{k-j-1})
$$

$$
+ \left(\frac{\rho}{\hat{\rho}}\right)^{t_k-t_{k-j}-n_{[t_{k-j}+1,t_k]}^k} \left(1 - \left(\frac{\rho}{\hat{\rho}}\right)^{t_{k-j}-t_{k-j-1}-n_{[t_{k-j-1}+1,t_{k-j}]}^k}\right) C_j^{j+1}(t_k, t_{k-j}, t_{k-j-1}) \right], \qquad (35)
$$

where

$$
G_j(s) := \frac{\mu+\lambda-\nu}{\mu-\nu} \hat{\kappa} \hat{\rho}^{n_{[s+1,t_{k-j-1}]}^k - n_{[s+1,t_{k-j-1}]}^{k-j}} \left(1 - \hat{\rho}^{n_{[s+1,t_{k-j-1}]}^{k-j} - n_{[s+1,t_{k-j-1}]}^{k-j-1}}\right).
$$

Combining (34) and (35), and using

$$
1 - \left(\frac{\rho}{\hat{\rho}}\right)^{t_{k-j+1}-t_{k-j-1}-n_{[t_{k-j-1}+1,t_{k-j+1}]}^k}
$$

$$
= 1 - \left(\frac{\rho}{\hat{\rho}}\right)^{t_{k-j+1}-t_{k-j}-n_{[t_{k-j}+1,t_{k-j+1}]}^k} + \left(\frac{\rho}{\hat{\rho}}\right)^{t_{k-j+1}-t_{k-j}-n_{[t_{k-j}+1,t_{k-j+1}]}^k} \left(1 - \left(\frac{\rho}{\hat{\rho}}\right)^{t_{k-j}-t_{k-j-1}-n_{[t_{k-j-1}+1,t_{k-j}]}^k}\right),
$$

we obtain exactly (32) with $j$ replaced by $j+1$, provided that

$$
C_{j+1}^x(t_k, t_{k-j-1}, s) = C_j^x(t_k, t_{k-j-1}, s) + C_j^x(t_k, t_{k-j}, t_{k-j-1}) G_j(s), \quad x = 1, \ldots, j,
$$

$$
C_{j+1}^{j+1}(t_k, t_{k-j-1}, s) = C_j^j(t_k, t_{k-j-1}, s) + C_j^{j+1}(t_k, t_{k-j}, t_{k-j-1}) G_j(s),
$$

and $C_{j+1}^{j+2} = 1$. This is precisely the recursion in (27). Therefore (32) holds for all $j = 1, \ldots, k-1$.

Finally, take $j = k - i$ in (32). Then for every $i = 1, \ldots, k-1$ and $s \in S_{t_i}$,

$$
\|D_{k-i}(t_k, s)\| \le \frac{M-\mu+\nu}{\nu} \hat{\rho}^{t_k-s-n_{[s+1,t_k]}^k} \left[ \left(\frac{\rho}{\hat{\rho}}\right)^{t_k-t_i-n_{[t_i+1,t_k]}^k} \left(1 - \left(\frac{\rho}{\hat{\rho}}\right)^{t_i-s-n_{[s+1,t_i]}^k}\right) \right.
$$

$$
+ \sum_{x=1}^{k-i} \left(\frac{\rho}{\hat{\rho}}\right)^{t_k-t_{k-x+1}-n_{[t_{k-x+1}+1,t_k]}^k} \left(1 - \left(\frac{\rho}{\hat{\rho}}\right)^{t_{k-x+1}-t_{k-x}-n_{[t_{k-x}+1,t_{k-x+1}]}^k}\right) C_{k-i}^x(t_k, t_i, s) \right],
$$

where we used $C_{k-i}^{k-i+1} = 1$. Combining this bound with (31), then substituting into (30), and finally multiplying by the stability factor $\rho^{t-t_k}$, gives (26). This completes the proof. □

**Lemma E.4.** *Fix time $t$. Define*

$$
\tilde{H}_i = \int_0^1 \nabla^2 \hat{F}_i \left( w_i^{-\mathbf{S}_{1:i-1}} + u \left( w_i^{-S_{\le t} \setminus \{i+1,\ldots,t\}} - w_i^{-\mathbf{S}_{1:i-1}} \right) \right) du, \quad \tilde{P}_i = (\tilde{H}_i + \lambda I)^{-1} \lambda I, \quad \hat{P}_i = (\hat{H}_i + \lambda I)^{-1} \lambda I,
$$

$$
P_i = (\nabla^2 \hat{F}_i(w_i^{-\mathbf{S}_{1:i-1}}) + \lambda I)^{-1} \lambda I, \quad \forall i \in [t].
$$

*Suppose all tasks between $s$ and $t$ have not yet been deleted. Then we have*

$$
\left\| \prod_{i=s}^t \tilde{P}_i - \prod_{i=s}^t \hat{P}_i \right\| \le \frac{(M-\mu+\nu)}{\nu} (\hat{\rho}^{t-s+1} - \rho^{t-s+1}), \quad \left\| \prod_{i=s}^t P_i - \prod_{i=s}^t \hat{P}_i \right\| \le (\hat{\rho}^{t-s+1} - \rho^{t-s+1}).
$$

*When $\nu = 0$, the inequality above becomes*

$$\left\| \prod_{i=s}^{t} \tilde{P}_i - \prod_{i=s}^{t} \hat{P}_i \right\| \le (t - s + 1)\rho^{t-s+1}\frac{M - \mu}{\mu + \lambda}, \quad \left\| \prod_{i=s}^{t} P_i - \prod_{i=s}^{t} \hat{P}_i \right\| = 0.$$

*Proof of Lemma E.4.* By telescoping, we have

$$\left\| \prod_{i=s}^{t} \tilde{P}_i - \prod_{i=s}^{t} \hat{P}_i \right\| = \left\| (\tilde{P}_t - \hat{P}_t)\prod_{i=s}^{t-1} \tilde{P}_i + \hat{P}_t(\tilde{P}_{t-1} - \hat{P}_{t-1})\prod_{i=s}^{t-2} \tilde{P}_i + \cdots + \prod_{i=s+1}^{t} \hat{P}_i(\tilde{P}_s - \hat{P}_s) \right\|. \tag{36}$$

By Assumptions 2.3 and 2.4, $\|\tilde{P}_i\|$ and $\|\hat{P}_i\|$ are upper bounded by $\rho$ and $\hat{\rho}$, respectively. Moreover, for each $i$,

$$\|\tilde{P}_i - \hat{P}_i\| = \lambda \left\| (\tilde{H}_i + \lambda I)^{-1}(\hat{H}_i - \tilde{H}_i)(\hat{H}_i + \lambda I)^{-1} \right\|$$

$$\le \frac{\lambda(M - \mu + \nu)}{(\mu + \lambda)(\mu + \lambda - \nu)}.$$

Applying these bounds in (36), we obtain

$$\left\| \prod_{i=s}^{t} \tilde{P}_i - \prod_{i=s}^{t} \hat{P}_i \right\| \le \rho^{t-s}(1 + \hat{\rho}/\rho + \cdots + (\hat{\rho}/\rho)^{t-s})\frac{\lambda(M - \mu + \nu)}{(\mu + \lambda)(\mu + \lambda - \nu)}$$

$$= \rho^{t-s}\frac{1 - (\hat{\rho}/\rho)^{t-s+1}}{1 - \hat{\rho}/\rho}\frac{\lambda(M - \mu + \nu)}{(\mu + \lambda)(\mu + \lambda - \nu)}$$

$$= \frac{\rho^{-1}}{1 - \hat{\rho}/\rho}\frac{\lambda(M - \mu + \nu)}{(\mu + \lambda)(\mu + \lambda - \nu)}(\rho^{t-s+1} - \hat{\rho}^{t-s+1})$$

$$= \frac{(\lambda + \mu)/\lambda}{1 - (\lambda + \mu)/((\lambda + \mu) - \nu)}\frac{\lambda(M - \mu + \nu)}{(\mu + \lambda)(\mu + \lambda - \nu)}(\rho^{t-s+1} - \hat{\rho}^{t-s+1})$$

$$= \frac{M - \mu + \nu}{\nu}(\hat{\rho}^{t-s+1} - \rho^{t-s+1}).$$

The bound for $\left\| \prod_{i=s}^{t} P_i - \prod_{i=s}^{t} \hat{P}_i \right\|$ follows analogously, with the only difference being

$$\|P_i - \hat{P}_i\| \le \frac{\lambda\nu}{(\mu + \lambda)(\mu + \lambda - \nu)}.$$

When $\nu = 0$, we have $\hat{\rho} = \rho$, and the first line of the preceding bound gives

$$\left\| \prod_{i=s}^{t} \tilde{P}_i - \prod_{i=s}^{t} \hat{P}_i \right\| \le \rho^{t-s}(t - s + 1)\frac{\lambda(M - \mu)}{(\mu + \lambda)^2} = \rho^{t-s+1}(t - s + 1)\frac{M - \mu}{\mu + \lambda}.$$

Moreover, since $\hat{H}_i = \nabla^2 \hat{F}_i(w_i^{-\mathbf{S}_{1:i-1}})$ when $\nu = 0$, we have

$$\left\| \prod_{i=s}^{t} P_i - \prod_{i=s}^{t} \hat{P}_i \right\| = 0.$$

This completes the proof. $\qquad\square$

**Lemma E.5.** *Define $\hat{P}_j = (\hat{H}_j + \lambda I)^{-1}\lambda I$, and $\hat{\kappa} := \max\left\{ \frac{M+\nu}{M+\lambda+\nu}, \frac{|\mu-\nu|}{\mu-\nu+\lambda} \right\}$. Suppose $\lambda > \max\{0, \nu - \mu\}$ and $k > i$. Then we have the following for any $\tau > s$, $\tau, s = 1 \ldots, t$:*

$$\left\| \prod_{\substack{j=s+1 \\ j \notin S_{\le t_k}}}^{\tau} \hat{P}_j - \prod_{\substack{j=s+1 \\ j \notin S_{\le t_i}}}^{\tau} \hat{P}_j \right\| \le \frac{\mu + \lambda - \nu}{\mu - \nu}\hat{\kappa}\hat{\rho}^{\tau-s-n_{[s+1,\tau]}^k}\left( 1 - \hat{\rho}^{n_{[s+1,\tau]}^k - n_{[s+1,\tau]}^i} \right).$$

*When $\nu = 0$, we have*

$$\left\| \prod_{\substack{j=s+1 \\ j \notin S_{\leq t_k}}}^{\tau} \hat{P}_j - \prod_{\substack{j=s+1 \\ j \notin S_{\leq t_i}}}^{\tau} \hat{P}_j \right\| \leq \frac{\mu + \lambda}{\mu} \kappa \rho^{\tau - s - n^k_{[s+1,\tau]}} \left( 1 - \rho^{n^k_{[s+1,\tau]} - n^i_{[s+1,\tau]}} \right).$$

*Moreover, the same bound holds for the exact-curvature products:*

$$\left\| \prod_{\substack{j=s+1 \\ j \notin S_{\leq t_k}}}^{\tau} P_j - \prod_{\substack{j=s+1 \\ j \notin S_{\leq t_i}}}^{\tau} P_j \right\| \leq \frac{\mu + \lambda}{\mu} \kappa \rho^{\tau - s - n^k_{[s+1,\tau]}} \left( 1 - \rho^{n^k_{[s+1,\tau]} - n^i_{[s+1,\tau]}} \right)$$

$$\left\| \prod_{\substack{j=s+1 \\ j \notin S_{\leq t_k}}}^{\tau} \tilde{P}_j - \prod_{\substack{j=s+1 \\ j \notin S_{\leq t_i}}}^{\tau} \tilde{P}_j \right\| \leq \frac{\mu + \lambda}{\mu} \kappa \rho^{\tau - s - n^k_{[s+1,\tau]}} \left( 1 - \rho^{n^k_{[s+1,\tau]} - n^i_{[s+1,\tau]}} \right).$$

*Proof.* Since $S_{\leq t_i} \subseteq S_{\leq t_k}$, compared with the second product, the first product excludes the additional set of indices

$$R := (S_{\leq t_k} \setminus S_{\leq t_i}) \cap [s + 1, \tau].$$

Hence

$$|R| = n^k_{[s+1,\tau]} - n^i_{[s+1,\tau]}.$$

Interpreting "excluding $j$" as replacing the corresponding factor $\hat{P}_j$ by $I$, define

$$Q^{(0)} = \prod_{\substack{j=s+1 \\ j \notin S_{\leq t_i}}}^{\tau} \hat{P}_j, \qquad Q^{(*)} = \prod_{\substack{j=s+1 \\ j \notin S_{\leq t_k}}}^{\tau} \hat{P}_j.$$

If $R = \emptyset$, then $Q^{(*)} = Q^{(0)}$, and the result is immediate. Otherwise, let $R = \{r_1 < \cdots < r_m\}$, where $m = n^k_{[s+1,\tau]} - n^i_{[s+1,\tau]}$. For $\ell = 0, \ldots, m$, define $Q^{(\ell)}$ as the product obtained from $Q^{(0)}$ by replacing $\hat{P}_{r_1}, \ldots, \hat{P}_{r_\ell}$ with $I$. Then $Q^{(m)} = Q^{(*)}$, and

$$Q^{(*)} - Q^{(0)} = \sum_{\ell=1}^{m} \left( Q^{(\ell)} - Q^{(\ell-1)} \right).$$

For each $\ell$, only the factor at position $r_\ell$ changes. Since $\|\hat{P}_j\| \leq \hat{\rho}$, we have

$$\left\| Q^{(\ell)} - Q^{(\ell-1)} \right\| \leq \|I - \hat{P}_{r_\ell}\| \hat{\rho}^{\tau - s - n^i_{[s+1,\tau]} - \ell}.$$

Moreover,

$$I - \hat{P}_j = (\hat{H}_j + \lambda I)^{-1} \hat{H}_j,$$

and therefore

$$\|I - \hat{P}_j\| \leq \hat{\kappa} = \max \left\{ \frac{M + \nu}{M + \lambda + \nu}, \frac{|\mu - \nu|}{\mu - \nu + \lambda} \right\}.$$

Thus,

$$\begin{aligned} \left\| Q^{(*)} - Q^{(0)} \right\| &\leq \hat{\kappa} \sum_{\ell=1}^{m} \hat{\rho}^{\tau - s - n^i_{[s+1,\tau]} - \ell} \\ &= \hat{\kappa} \hat{\rho}^{\tau - s - n^k_{[s+1,\tau]}} \frac{1 - \hat{\rho}^m}{1 - \hat{\rho}} \\ &= \frac{\mu + \lambda - \nu}{\mu - \nu} \hat{\kappa} \hat{\rho}^{\tau - s - n^k_{[s+1,\tau]}} \left( 1 - \hat{\rho}^{n^k_{[s+1,\tau]} - n^i_{[s+1,\tau]}} \right), \end{aligned}$$

where the last equality uses

$$1 - \hat{\rho} = 1 - \frac{\lambda}{\lambda + \mu - \nu} = \frac{\mu - \nu}{\lambda + \mu - \nu}.$$

When $\nu = 0$, the same argument gives

$$\left\| Q^{(\ell)} - Q^{(\ell-1)} \right\| \le \|I - \hat{P}_{r_\ell}\| \rho^{\tau - s - n_{[s+1,\tau]}^i - \ell}, \quad \|I - \hat{P}_j\| \le \kappa.$$

Thus, the stated bound follows by summing the resulting geometric series. Moreover, since $P_j$ and $\tilde{P}_j$ satisfy the same exact-curvature bounds from Assumption 2.3, independently of $\nu$, the same argument applies after replacing $\hat{P}_j$ with $P_j$ or $\tilde{P}_j$. □

**Lemma E.6.** *Fix time $t$. Define*

$$\tilde{H}_i = \int_0^1 \nabla^2 \hat{F}_i \Big( w_i^{-\mathbf{S}_{1:i-1}} + u \big( w_i^{-S_{\le t} \backslash \{i+1,\dots,t\}} - w_i^{-\mathbf{S}_{1:i-1}} \big) \Big) du, \quad \tilde{P}_i = (\tilde{H}_i + \lambda I)^{-1} \lambda I, \quad \forall i \in [t].$$

*Then, consider the unlearning models $w_i^{-\mathbf{S}_{1:k}}$ and $w_j^{-\mathbf{S}_{1:k}}$ updated in Alg. 2, and the corresponding retrained models $w_i^{-S_{\le t}}$ and $w_j^{-S_{\le t}}$, where $t \ge i > j \ge k$. If no unlearning request occurs and no task is deleted during the interval $\{k+1,\dots,t\}$ by time $t$, then the following relation holds:*

$$w_i^{-S_{\le t}} - w_i^{-\mathbf{S}_{1:k}} = \prod_{a=j+1}^{i} \tilde{P}_a (w_j^{-S_{\le t}} - w_j^{-\mathbf{S}_{1:k}}), \tag{37}$$

*and*

$$\|w_i^{-S_{\le t}} - w_i^{-\mathbf{S}_{1:k}}\| \le \rho^{i-j} \|w_j^{-S_{\le t}} - w_j^{-\mathbf{S}_{1:k}}\| \tag{38}$$

*Proof of Lemma E.6.* Under the CL algorithm ($\ell_2$-CL), we have the following first-order conditions for the unlearning model $w_s^{-\mathbf{S}_{1:k}}$ and retraining model $w_s^{-S_{\le t}}$ for $j + 1 \le s \le i$:

$$\nabla \hat{F}_s(w_s^{-S_{\le t}}) + \lambda(w_s^{-S_{\le t}} - w_{s-1}^{-S_{\le t}}) = 0, \ \nabla \hat{F}_s(w_s^{-\mathbf{S}_{1:k}}) + \lambda(w_s^{-\mathbf{S}_{1:k}} - w_{s-1}^{-\mathbf{S}_{1:k}}) = 0.$$

Moreover, since $s > k$, we have $w_s^{-\mathbf{S}_{1:k}} = w_s^{-\mathbf{S}_{1:s-1}}$ for any $i \ge s \ge j+1$. Since the retrained model at task $s$ only depends on tasks up to $s$, for any $j + 1 \le s \le i$ we have $w_s^{-S_{\le t} \backslash \{s+1,\dots,t\}} = w_s^{-S_{\le t}}$.

By Taylor's theorem, expanding $\nabla \hat{F}_s(w_s^{-S_{\le t}})$ around $w_s^{-\mathbf{S}_{1:k}}$ gives

$$\nabla \hat{F}_s(w_s^{-\mathbf{S}_{1:k}}) + \tilde{H}_s(w_s^{-S_{\le t}} - w_s^{-\mathbf{S}_{1:k}}) + \lambda(w_s^{-S_{\le t}} - w_{s-1}^{-S_{\le t}}) = 0$$
$$\implies -\lambda(w_s^{-\mathbf{S}_{1:k}} - w_{s-1}^{-\mathbf{S}_{1:k}}) + \tilde{H}_s(w_s^{-S_{\le t}} - w_s^{-\mathbf{S}_{1:k}}) + \lambda(w_s^{-S_{\le t}} - w_{s-1}^{-S_{\le t}}) = 0$$
$$\implies w_s^{-S_{\le t}} - w_s^{-\mathbf{S}_{1:k}} = (\tilde{H}_s + \lambda I)^{-1} \lambda (w_{s-1}^{-S_{\le t}} - w_{s-1}^{-\mathbf{S}_{1:k}}). \tag{39}$$

Iterating the recursion above over $s = j+1,\dots,i$ yields (37). Taking norms and using $\|\tilde{P}_s\| \le \rho$ for each $s$ gives the desired inequality. □

**Lemma E.7.** *Let $t$ be an unlearning time where $S_t \ne \emptyset$. Define*

$$\tilde{H}_i = \int_0^1 \nabla^2 \hat{F}_i \Big( w_i^{-\mathbf{S}_{1:i-1}} + u \big( w_i^{-S_{\le t} \backslash \{i+1,\dots,t\}} - w_i^{-\mathbf{S}_{1:i-1}} \big) \Big) du,$$

*and*

$$\tilde{P}_i = (\tilde{H}_i + \lambda I)^{-1} \lambda I, \qquad \hat{P}_i = (\hat{H}_i + \lambda I)^{-1} \lambda I.$$

*Then, for the unlearning model $w_t^{-\mathbf{S}_{1:t}}$ updated in Alg. 2 and the retraining model $w_t^{-S_{\le t}}$, the following equation holds:*

$$w_t^{-S_{\le t}} - w_t^{-\mathbf{S}_{1:t}} = \sum_{s \in S_{\le t}} \left( \prod_{\substack{i=s+1 \\ i \notin S_{\le t}}}^{t} \tilde{P}_i - \prod_{\substack{i=s+1 \\ i \notin S_{\le t}}}^{t} \hat{P}_i \right) \Delta_s - \sum_{\tau \in U_{t-1}} \left( \prod_{\substack{i=\tau+1 \\ i \notin S_{\le t}}}^{t} \tilde{P}_i - \prod_{\substack{i=\tau+1 \\ i \notin S_{\le t}}}^{t} \hat{P}_i \right) \bar{\Delta}_\tau. \tag{40}$$

*Proof of Lemma E.7.* We first prove the following identity. For any $\tau = 0, \ldots, t$,

$$A_\tau := w_\tau^{-S_{\leq t} \backslash \{\tau+1,\ldots,t\}} - w_\tau^{-\mathbf{S}_{1:\tau-1}} = \sum_{s \in S_{\leq t} \backslash \{\tau+1,\ldots,t\}} \prod_{\substack{i=s+1 \\ i \notin S_{\leq t}}}^{\tau} \tilde{P}_i \Delta_s - \sum_{j \in U_{\tau-1}} \prod_{\substack{i=j+1 \\ i \notin S_{\leq t}}}^{\tau} \tilde{P}_i \bar{\Delta}_j. \tag{41}$$

Here $U_{-1} = U_0 = \emptyset$, and the identity is trivial when $\tau = 0$.

Assume that (41) holds at time $\tau$. We prove it for $\tau + 1$. If $\tau + 1 \in S_{\leq t}$, then task $\tau + 1$ is skipped in the retraining trajectory. Hence $w_{\tau+1}^{-S_{\leq t} \backslash \{\tau+2,\ldots,t\}} = w_\tau^{-S_{\leq t} \backslash \{\tau+1,\ldots,t\}}$. Using $w_\tau^{-\mathbf{S}_{1:\tau}} = w_\tau^{-\mathbf{S}_{1:\tau-1}} + \bar{\Delta}_\tau$ and $\Delta_{\tau+1} = w_\tau^{-\mathbf{S}_{1:\tau}} - w_{\tau+1}^{-\mathbf{S}_{1:\tau}}$, we obtain

$$\begin{aligned} A_{\tau+1} &= w_{\tau+1}^{-S_{\leq t} \backslash \{\tau+2,\ldots,t\}} - w_{\tau+1}^{-\mathbf{S}_{1:\tau}} \\ &= w_\tau^{-S_{\leq t} \backslash \{\tau+1,\ldots,t\}} - w_\tau^{-\mathbf{S}_{1:\tau}} + w_\tau^{-\mathbf{S}_{1:\tau}} - w_{\tau+1}^{-\mathbf{S}_{1:\tau}} \\ &= w_\tau^{-S_{\leq t} \backslash \{\tau+1,\ldots,t\}} - w_\tau^{-\mathbf{S}_{1:\tau-1}} - \bar{\Delta}_\tau + \Delta_{\tau+1} \\ &= A_\tau - \bar{\Delta}_\tau + \Delta_{\tau+1}. \end{aligned}$$

Substituting the induction hypothesis gives (41) with $\tau$ replaced by $\tau + 1$: the new deleted task contributes $\Delta_{\tau+1}$ with an empty product, and the term $-\bar{\Delta}_\tau$ is included in the second sum if $\tau \in U_\tau$, while it is zero otherwise by the convention $\bar{\Delta}_\tau = \mathbf{0}$ when $S_\tau = \emptyset$.

If $\tau + 1 \notin S_{\leq t}$, then both trajectories train on task $\tau + 1$. By Lemma E.6, we obtain

$$A_{\tau+1} = \tilde{P}_{\tau+1} \left( w_\tau^{-S_{\leq t} \backslash \{\tau+1,\ldots,t\}} - w_\tau^{-\mathbf{S}_{1:\tau}} \right) = \tilde{P}_{\tau+1}(A_\tau - \bar{\Delta}_\tau).$$

Substituting the induction hypothesis again yields (41) at $\tau + 1$, since the retained factor $\tilde{P}_{\tau+1}$ is appended to each product, and the additional term $-\tilde{P}_{\tau+1} \bar{\Delta}_\tau$ is included in the second sum when $\tau \in U_\tau$, and is zero otherwise. This completes the induction.

Taking $\tau = t$ in (41) gives

$$w_t^{-S_{\leq t}} - w_t^{-\mathbf{S}_{1:t-1}} = \sum_{s \in S_{\leq t}} \prod_{\substack{i=s+1 \\ i \notin S_{\leq t}}}^{t} \tilde{P}_i \Delta_s - \sum_{\tau \in U_{t-1}} \prod_{\substack{i=\tau+1 \\ i \notin S_{\leq t}}}^{t} \tilde{P}_i \bar{\Delta}_\tau. \tag{42}$$

Then, since $S_t \neq \emptyset$, combining (42) and the update of $w_t^{-\mathbf{S}_{1:t}}$ in (14), we have

$$\begin{aligned} &w_t^{-S_{\leq t}} - w_t^{-\mathbf{S}_{1:t}} \\ =& w_t^{-S_{\leq t}} - w_t^{-\mathbf{S}_{1:t-1}} - \sum_{s \in S_{\leq t}} \prod_{\substack{i=s+1 \\ i \notin S_{\leq t}}}^{t} \hat{P}_i \Delta_s + \sum_{\tau \in U_{t-1}} \prod_{\substack{i=\tau+1 \\ i \notin S_{\leq t}}}^{t} \hat{P}_i \bar{\Delta}_\tau \\ =& \sum_{s \in S_{\leq t}} \prod_{\substack{i=s+1 \\ i \notin S_{\leq t}}}^{t} \tilde{P}_i \Delta_s - \sum_{\tau \in U_{t-1}} \prod_{\substack{i=\tau+1 \\ i \notin S_{\leq t}}}^{t} \tilde{P}_i \bar{\Delta}_\tau - \sum_{s \in S_{\leq t}} \prod_{\substack{i=s+1 \\ i \notin S_{\leq t}}}^{t} \hat{P}_i \Delta_s + \sum_{\tau \in U_{t-1}} \prod_{\substack{i=\tau+1 \\ i \notin S_{\leq t}}}^{t} \hat{P}_i \bar{\Delta}_\tau \\ =& \sum_{s \in S_{\leq t}} \left( \prod_{\substack{i=s+1 \\ i \notin S_{\leq t}}}^{t} \tilde{P}_i - \prod_{\substack{i=s+1 \\ i \notin S_{\leq t}}}^{t} \hat{P}_i \right) \Delta_s - \sum_{\tau \in U_{t-1}} \left( \prod_{\substack{i=\tau+1 \\ i \notin S_{\leq t}}}^{t} \tilde{P}_i - \prod_{\substack{i=\tau+1 \\ i \notin S_{\leq t}}}^{t} \hat{P}_i \right) \bar{\Delta}_\tau. \end{aligned}$$

This completes the proof.

$\square$

**Lemma E.8.** *Suppose task $s$ is unlearned at the $(k-j)$-th unlearning request $t_{k-j}$. Let $d_j(t_k, s)$ be the coefficient of $\Delta_s$ in $\bar{\Delta}_{t_k}$, so that*

$$\bar{\Delta}_{t_k} = \sum_{i=0}^{k-1} \sum_{s \in S_{t_{k-i}}} d_i(t_k, s)\Delta_s.$$

*Likewise, let $D_j(t_k, s)$ be the coefficient of $\Delta_s$ in $w_{t_k}^{-S_{\leq t_k}} - w_{t_k}^{-\mathbf{S}_{1:t_k}}$, such that*

$$w_{t_k}^{-S_{\leq t_k}} - w_{t_k}^{-\mathbf{S}_{1:t_k}} = \sum_{i=0}^{k-1} \sum_{s \in S_{t_{k-i}}} D_i(t_k, s)\Delta_s.$$

*Then, for $j = 0, \ldots, k-2$ and $s \in S_{t_{k-j-1}}$,*

$$d_0(t_k, s) = \prod_{\substack{i=s+1 \\ i \notin S_{\leq t_k}}}^{t_k} \hat{P}_i, \qquad d_{j+1}(t_k, s) = d_j(t_k, s) - d_j(t_k, t_{k-j-1})d_0(t_{k-j-1}, s), \tag{43}$$

*and*

$$D_0(t_k, s) = \prod_{\substack{i=s+1 \\ i \notin S_{\leq t_k}}}^{t_k} \tilde{P}_i - \prod_{\substack{i=s+1 \\ i \notin S_{\leq t_k}}}^{t_k} \hat{P}_i, \quad D_{j+1}(t_k, s) = D_j(t_k, s) - D_j(t_k, t_{k-j-1}) \prod_{\substack{i=s+1 \\ i \notin S_{\leq t_{k-j-1}}}}^{t_{k-j-1}} \hat{P}_i. \tag{44}$$

*Proof of Lemma E.8.* We use the same notation $d_j(t_k, a)$ and $D_j(t_k, a)$ for any intermediate time index $a$ appearing as the starting point of a propagated correction.

First, it is straightforward to verify that, under the iterative update in (14) and by Lemma E.7, each $\bar{\Delta}_t$ and the difference $w_{t_k}^{-S_{\leq t_k}} - w_{t_k}^{-\mathbf{S}_{1:t_k}}$ can be expressed as a linear combination of $\{\Delta_s : s \in S_{\leq t_k}\}$. Next we prove the equations in (43) by induction.

If task $s$ is unlearned at time $t_k$, then by definition its matrix coefficient in $\bar{\Delta}_{t_k}$ is $d_0(t_k, s)$. Compared to (14), this directly yields the stated expression for $d_0(t_k, s)$ in (43).

Now assume (43) holds for all $i = 0, \ldots, j$. Moreover, by (14) and the definition of $d_l(t_k, s)$, $d_l(t_k, s)$ is the coefficient of $\Delta_s$ in $\bar{\Delta}_{t_k}$ when $s$ is unlearned at time $t_{k-l}$. Note that

$$\bar{\Delta}_{t_k} = \sum_{a=1}^{k} \sum_{s \in S_{t_a}} \prod_{\substack{r=s+1 \\ r \notin S_{\leq t_k}}}^{t_k} \hat{P}_r \Delta_s - \sum_{a=1}^{k-1} \prod_{\substack{r=t_a+1 \\ r \notin S_{\leq t_k}}}^{t_k} \hat{P}_r \bar{\Delta}_{t_a}$$

by (14), and that

$$\prod_{\substack{r=t_a+1 \\ r \notin S_{\leq t_k}}}^{t_k} \hat{P}_r = d_0(t_k, t_a)$$

by our notation.

Then, for any $l = 0, \ldots, k-1$, we have

$$d_l(t_k, s) = d_0(t_k, s) - \sum_{i=0}^{l-1} d_0(t_k, t_{k-l+i})d_i(t_{k-l+i}, s). \tag{45}$$

Hence

$$d_j(t_k, s) - d_{j+1}(t_k, s)$$

$$= d_0(t_k, s) - \sum_{i=0}^{j-1} d_0(t_k, t_{k-j+i}) d_i(t_{k-j+i}, s) - \left( d_0(t_k, s) - \sum_{i=0}^{j} d_0(t_k, t_{k-j+i-1}) d_i(t_{k-j+i-1}, s) \right)$$

$$= -\sum_{i=0}^{j-1} d_0(t_k, t_{k-j+i}) d_i(t_{k-j+i}, s) + \sum_{i=-1}^{j-1} d_0(t_k, t_{k-j+i}) d_{i+1}(t_{k-j+i}, s)$$

$$= -\sum_{i=0}^{j-1} d_0(t_k, t_{k-j+i}) \big( d_i(t_{k-j+i}, s) - d_{i+1}(t_{k-j+i}, s) \big) + d_0(t_k, t_{k-j-1}) d_0(t_{k-j-1}, s)$$

$$= -\sum_{i=0}^{j-1} d_0(t_k, t_{k-j+i}) d_i(t_{k-j+i}, t_{k-j-1}) d_0(t_{k-j-1}, s) + d_0(t_k, t_{k-j-1}) d_0(t_{k-j-1}, s)$$

$$= \left( d_0(t_k, t_{k-j-1}) - \sum_{i=0}^{j-1} d_0(t_k, t_{k-j+i}) d_i(t_{k-j+i}, t_{k-j-1}) \right) d_0(t_{k-j-1}, s)$$

$$= d_j(t_k, t_{k-j-1}) d_0(t_{k-j-1}, s),$$

where the fourth equality follows from the induction condition for $i \le j$, and the last equality uses (45) with $s = t_{k-j-1}$ and $l = j$. This completes the induction for (43).

Similarly, we prove (44) using (43) by induction.

If task $s$ is unlearned at time $t_k$, then by definition its matrix coefficient in $w_{t_k}^{-S_{\le t_k}} - w_{t_k}^{-\mathbf{S}_{1:t_k}}$ is $D_0(t_k, s)$. By Lemma E.7, since $s \in S_{t_k}$ is first unlearned at the most recent time $t_k$ and therefore does not appear in any $\bar{\Delta}_\tau$ with $\tau < t_k$, the stated expression for $D_0(t_k, s)$ in (44) follows.

Assume (44) holds for all $i = 0, \ldots, j$. Additionally, by Lemma E.7, for any $l = 0, \ldots, k-1$,

$$D_l(t_k, s) = D_0(t_k, s) - \sum_{i=0}^{l-1} D_0(t_k, t_{k-l+i}) d_i(t_{k-l+i}, s). \tag{46}$$

Therefore

$$D_j(t_k, s) - D_{j+1}(t_k, s)$$

$$= D_0(t_k, s) - \sum_{i=0}^{j-1} D_0(t_k, t_{k-j+i}) d_i(t_{k-j+i}, s) - \left( D_0(t_k, s) - \sum_{i=0}^{j} D_0(t_k, t_{k-j+i-1}) d_i(t_{k-j+i-1}, s) \right)$$

$$= -\sum_{i=0}^{j-1} D_0(t_k, t_{k-j+i}) d_i(t_{k-j+i}, s) + \sum_{i=-1}^{j-1} D_0(t_k, t_{k-j+i}) d_{i+1}(t_{k-j+i}, s)$$

$$= -\sum_{i=0}^{j-1} D_0(t_k, t_{k-j+i}) \big( d_i(t_{k-j+i}, s) - d_{i+1}(t_{k-j+i}, s) \big) + D_0(t_k, t_{k-j-1}) d_0(t_{k-j-1}, s)$$

$$= -\sum_{i=0}^{j-1} D_0(t_k, t_{k-j+i}) d_i(t_{k-j+i}, t_{k-j-1}) d_0(t_{k-j-1}, s) + D_0(t_k, t_{k-j-1}) d_0(t_{k-j-1}, s)$$

$$= \left( D_0(t_k, t_{k-j-1}) - \sum_{i=0}^{j-1} D_0(t_k, t_{k-j+i}) d_i(t_{k-j+i}, t_{k-j-1}) \right) d_0(t_{k-j-1}, s)$$

$$= D_j(t_k, t_{k-j-1}) d_0(t_{k-j-1}, s),$$

where the fourth equality follows from the induction condition, and the last equality uses (46) with $s = t_{k-j-1}$ and $l = j$. This completes the induction for (44). $\qquad \square$

**Lemma E.9.** *Fix $j = 0, \ldots, k-2$ and $s \in S_{t_{k-j-1}}$. For the $D_j(t_k, s)$ defined in Lemma E.8, we have*

$$D_{j+1}(t_k, s) = \sum_{i=0}^{j} D_{j-i}(t_k, t_{k-j-1}) \left( \prod_{\substack{a=s+1 \\ a \notin S_{\leq t_{k-j+i}}}}^{t_{k-j-1}} \hat{P}_a - \prod_{\substack{a=s+1 \\ a \notin S_{\leq t_{k-j+i-1}}}}^{t_{k-j-1}} \hat{P}_a \right) + D_0(t_k, s) - D_0(t_k, t_{k-j-1}) \prod_{\substack{a=s+1 \\ a \notin S_{\leq t_k}}}^{t_{k-j-1}} \hat{P}_a.$$

(47)

*Proof.* Let $u = t_{k-j-1}$, and define

$$A_i(s) := \prod_{\substack{a=s+1 \\ a \notin S_{\leq t_{k-j+i-1}}}}^{u} \hat{P}_a, \qquad i = 0, \ldots, j+1.$$

By Lemma E.8,

$$D_{j+1}(t_k, s) = D_j(t_k, s) - D_j(t_k, u) A_0(s).$$

We claim that for $b = 0, \ldots, j-1$,

$$D_{j+1}(t_k, s) = \sum_{i=0}^{b} D_{j-i}(t_k, u)\big(A_{i+1}(s) - A_i(s)\big) + D_{j-b-1}(t_k, s) - D_{j-b-1}(t_k, u) A_{b+1}(s).$$

For $b = 0$ and $j \geq 1$, add and subtract $D_j(t_k, u) A_1(s)$, where $u = t_{k-j-1}$. Then

$$D_{j+1}(t_k, s) = D_j(t_k, s) - D_j(t_k, u) A_0(s)$$
$$= D_j(t_k, u)\big(A_1(s) - A_0(s)\big) + \big(D_j(t_k, s) - D_j(t_k, u) A_1(s)\big).$$

It remains to simplify the second term. By Lemma E.8, applied once with starting index $s$ and once with starting index $u$, we have

$$D_j(t_k, s) = D_{j-1}(t_k, s) - D_{j-1}(t_k, t_{k-j}) \prod_{\substack{a=s+1 \\ a \notin S_{\leq t_{k-j}}}}^{t_{k-j}} \hat{P}_a,$$

$$D_j(t_k, u) = D_{j-1}(t_k, u) - D_{j-1}(t_k, t_{k-j}) \prod_{\substack{a=u+1 \\ a \notin S_{\leq t_{k-j}}}}^{t_{k-j}} \hat{P}_a.$$

Moreover, by the product decomposition convention,

$$\prod_{\substack{a=s+1 \\ a \notin S_{\leq t_{k-j}}}}^{t_{k-j}} \hat{P}_a = \left( \prod_{\substack{a=u+1 \\ a \notin S_{\leq t_{k-j}}}}^{t_{k-j}} \hat{P}_a \right) A_1(s).$$

Therefore,

$$D_j(t_k, s) - D_j(t_k, u) A_1(s) = D_{j-1}(t_k, s) - D_{j-1}(t_k, t_{k-j}) \left( \prod_{\substack{a=u+1 \\ a \notin S_{\leq t_{k-j}}}}^{t_{k-j}} \hat{P}_a \right) A_1(s)$$

$$- D_{j-1}(t_k, u) A_1(s) + D_{j-1}(t_k, t_{k-j}) \left( \prod_{\substack{a=u+1 \\ a \notin S_{\leq t_{k-j}}}}^{t_{k-j}} \hat{P}_a \right) A_1(s)$$

$$= D_{j-1}(t_k, s) - D_{j-1}(t_k, u) A_1(s).$$

Thus,
$$D_{j+1}(t_k, s) = D_j(t_k, u)\big(A_1(s) - A_0(s)\big) + D_{j-1}(t_k, s) - D_{j-1}(t_k, u)A_1(s),$$
which proves the base case.

Assume the claim holds for $b = l < j - 1$. Then we have
$$D_{j+1}(t_k, s) = \sum_{i=0}^{l} D_{j-i}(t_k, u)\big(A_{i+1}(s) - A_i(s)\big) + D_{j-l-1}(t_k, s) - D_{j-l-1}(t_k, t_{k-j-1}) \prod_{\substack{a=s+1 \\ a \notin S_{\le t_{k-j+l}}}}^{t_{k-j-1}} \hat{P}_a.$$

The second term can be written as
$$D_{j-l-1}(t_k, s) - D_{j-l-1}(t_k, t_{k-j-1}) \prod_{\substack{a=s+1 \\ a \notin S_{\le t_{k-j+l}}}}^{t_{k-j-1}} \hat{P}_a = D_{j-l-1}(t_k, t_{k-j-1}) \left( \prod_{\substack{a=s+1 \\ a \notin S_{\le t_{k-j+l+1}}}}^{t_{k-j-1}} \hat{P}_a - \prod_{\substack{a=s+1 \\ a \notin S_{\le t_{k-j+l}}}}^{t_{k-j-1}} \hat{P}_a \right)$$
$$+ D_{j-l-1}(t_k, s) - D_{j-l-1}(t_k, t_{k-j-1}) \prod_{\substack{a=s+1 \\ a \notin S_{\le t_{k-j+l+1}}}}^{t_{k-j-1}} \hat{P}_a.$$

It remains to simplify the second line. By Lemma E.8,
$$D_{j-l-1}(t_k, s) = D_{j-l-2}(t_k, s) - D_{j-l-2}(t_k, t_{k-j+l+1}) \prod_{\substack{a=s+1 \\ a \notin S_{\le t_{k-j+l+1}}}}^{t_{k-j+l+1}} \hat{P}_a,$$
$$D_{j-l-1}(t_k, t_{k-j-1}) = D_{j-l-2}(t_k, t_{k-j-1}) - D_{j-l-2}(t_k, t_{k-j+l+1}) \prod_{\substack{a=t_{k-j-1}+1 \\ a \notin S_{\le t_{k-j+l+1}}}}^{t_{k-j+l+1}} \hat{P}_a.$$

Therefore,
$$D_{j-l-1}(t_k, s) - D_{j-l-1}(t_k, t_{k-j-1}) \prod_{\substack{a=s+1 \\ a \notin S_{\le t_{k-j+l+1}}}}^{t_{k-j-1}} \hat{P}_a$$
$$= D_{j-l-2}(t_k, s) - D_{j-l-2}(t_k, t_{k-j+l+1}) \prod_{\substack{a=s+1 \\ a \notin S_{\le t_{k-j+l+1}}}}^{t_{k-j+l+1}} \hat{P}_a - D_{j-l-2}(t_k, t_{k-j-1}) \prod_{\substack{a=s+1 \\ a \notin S_{\le t_{k-j+l+1}}}}^{t_{k-j-1}} \hat{P}_a$$
$$+ D_{j-l-2}(t_k, t_{k-j+l+1}) \prod_{\substack{a=t_{k-j-1}+1 \\ a \notin S_{\le t_{k-j+l+1}}}}^{t_{k-j+l+1}} \hat{P}_a \prod_{\substack{a=s+1 \\ a \notin S_{\le t_{k-j+l+1}}}}^{t_{k-j-1}} \hat{P}_a$$
$$= D_{j-l-2}(t_k, s) - D_{j-l-2}(t_k, t_{k-j-1}) \prod_{\substack{a=s+1 \\ a \notin S_{\le t_{k-j+l+1}}}}^{t_{k-j-1}} \hat{P}_a.$$

This proves the induction step from $b = l$ to $b = l + 1$.

Taking $b = j - 1$, we obtain
$$D_{j+1}(t_k, s) = \sum_{i=0}^{j-1} D_{j-i}(t_k, u)(A_{i+1}(s) - A_i(s)) + D_0(t_k, s) - D_0(t_k, u)A_j(s).$$

Finally, add and subtract $D_0(t_k, u)A_{j+1}(s)$. Since $A_{j+1}(s) = \prod_{\substack{a=s+1 \\ a \notin S_{\le t_k}}}^{u} \hat{P}_a$, this gives exactly (47). $\qquad\square$

### E.3. Proof of Proposition 5.3

By Lemma E.6, for any time $t$ after the $k$-th unlearning request and before the next unlearning request, we have $S_{\leq t} = S_{\leq t_k}$ and $\mathbf{S}_{1:t} = \mathbf{S}_{1:t_k}$. Hence,

$$\left\| w_t^{-S_{\leq t}} - w_t^{-\mathbf{S}_{1:t}} \right\| = \left\| \prod_{i=t_k+1}^{t} \tilde{P}_i \left( w_{t_k}^{-S_{\leq t_k}} - w_{t_k}^{-\mathbf{S}_{1:t_k}} \right) \right\| \leq \rho^{t-t_k} \left\| w_{t_k}^{-S_{\leq t_k}} - w_{t_k}^{-\mathbf{S}_{1:t_k}} \right\|.$$

Therefore, it suffices to upper bound $\left\| w_{t_k}^{-S_{\leq t_k}} - w_{t_k}^{-\mathbf{S}_{1:t_k}} \right\|$.

For any retained task $i \notin S_{\leq t_k}$, the CL update rule ($\ell_2$-CL) gives the following first-order conditions:

$$\nabla \hat{F}_i(w_i^{-S_{\leq t_k}}) + \lambda \left( w_i^{-S_{\leq t_k}} - w_{i-1}^{-S_{\leq t_k}} \right) = 0, \ \nabla \hat{F}_i(w_i^{-\mathbf{S}_{1:i-1}}) + \lambda \left( w_i^{-\mathbf{S}_{1:i-1}} - w_{i-1}^{-\mathbf{S}_{1:i-1}} \right) = 0.$$

By Taylor's theorem with Hessian-Lipschitz remainder,

$$\nabla \hat{F}_i(w_i^{-S_{\leq t_k}}) = \nabla \hat{F}_i(w_i^{-\mathbf{S}_{1:i-1}}) + \nabla^2 \hat{F}_i(w_i^{-\mathbf{S}_{1:i-1}})(w_i^{-S_{\leq t_k}} - w_i^{-\mathbf{S}_{1:i-1}}) + R_i,$$

where

$$\|R_i\| \leq \frac{L_3}{2} \left\| w_i^{-S_{\leq t_k}} - w_i^{-\mathbf{S}_{1:i-1}} \right\|^2.$$

Combining the two first-order conditions gives

$$
\begin{aligned}
w_i^{-S_{\leq t_k}} - w_i^{-\mathbf{S}_{1:i-1}} &= -\left( \nabla^2 \hat{F}_i(w_i^{-\mathbf{S}_{1:i-1}}) + \lambda I \right)^{-1} R_i + \left( \nabla^2 \hat{F}_i(w_i^{-\mathbf{S}_{1:i-1}}) + \lambda I \right)^{-1} \lambda \left( w_{i-1}^{-S_{\leq t_k}} - w_{i-1}^{-\mathbf{S}_{1:i-1}} \right) \\
&= -Q_i + P_i \left( w_{i-1}^{-S_{\leq t_k}} - w_{i-1}^{-\mathbf{S}_{1:i-1}} \right),
\end{aligned}
\tag{48}
$$

where

$$Q_i := \left( \nabla^2 \hat{F}_i(w_i^{-\mathbf{S}_{1:i-1}}) + \lambda I \right)^{-1} R_i, \quad P_i := \left( \nabla^2 \hat{F}_i(w_i^{-\mathbf{S}_{1:i-1}}) + \lambda I \right)^{-1} \lambda I.$$

On the other hand, if $i \in S_{\leq t_k}$, then task $i$ is skipped in the retrained trajectory, and hence

$$w_i^{-S_{\leq t_k}} - w_i^{-\mathbf{S}_{1:i-1}} = w_{i-1}^{-S_{\leq t_k}} - w_i^{-\mathbf{S}_{1:i-1}} = w_{i-1}^{-S_{\leq t_k}} - w_{i-1}^{-\mathbf{S}_{1:i-1}} + \Delta_i. \tag{49}$$

Moreover,

$$w_i^{-S_{\leq t_k}} - w_i^{-\mathbf{S}_{1:i}} = w_i^{-S_{\leq t_k}} - w_i^{-\mathbf{S}_{1:i-1}} - \bar{\Delta}_i, \tag{50}$$

where we set $\bar{\Delta}_i = \mathbf{0}$ if no unlearning request is received at time $i$.

Combining (48)–(50), for each $i$ we obtain

$$
\begin{aligned}
w_i^{-S_{\leq t_k}} - w_i^{-\mathbf{S}_{1:i}} &= \mathbf{1}(i \notin S_{\leq t_k}) \left[ P_i \left( w_{i-1}^{-S_{\leq t_k}} - w_{i-1}^{-\mathbf{S}_{1:i-1}} \right) - Q_i \right] \\
&\quad + \mathbf{1}(i \in S_{\leq t_k}) \left[ w_{i-1}^{-S_{\leq t_k}} - w_{i-1}^{-\mathbf{S}_{1:i-1}} + \Delta_i \right] - \bar{\Delta}_i.
\end{aligned}
\tag{51}
$$

Iterating (51) from $s_0 := \min S_{\leq t_k}$ to $t_k$, and using

$$w_{s_0-1}^{-S_{\leq t_k}} - w_{s_0-1}^{-\mathbf{S}_{1:s_0-1}} = \mathbf{0},$$

yields

$$w_{t_k}^{-S_{\leq t_k}} - w_{t_k}^{-\mathbf{S}_{1:t_k}} = \sum_{i=s_0}^{t_k} \prod_{\substack{j=i+1 \\ j \notin S_{\leq t_k}}}^{t_k} P_j \left( \mathbf{1}(i \in S_{\leq t_k}) \Delta_i - \mathbf{1}(i \notin S_{\leq t_k}) Q_i \right) - \sum_{i=1}^{k} \prod_{\substack{j=t_i+1 \\ j \notin S_{\leq t_k}}}^{t_k} P_j \bar{\Delta}_{t_i}. \tag{52}$$

Using the definition of the Hessian-based correction terms $\bar{\Delta}_{t_k}$ in (14), (52) can be rewritten as

$$w_{t_k}^{-S_{\leq t_k}} - w_{t_k}^{-\mathbf{S}_{1:t_k}}$$

$$= \sum_{\substack{i \in S_{\leq t_k}}} \left( \prod_{\substack{j=i+1 \\ j \notin S_{\leq t_k}}}^{t_k} P_j - \prod_{\substack{j=i+1 \\ j \notin S_{\leq t_k}}}^{t_k} \hat{P}_j \right) \Delta_i - \sum_{i=s_0} \prod_{\substack{j=i+1 \\ j \notin S_{\leq t_k}}}^{t_k} P_j \, \mathbf{1}(i \notin S_{\leq t_k}) Q_i + \sum_{i=1}^{k-1} \left( \prod_{\substack{j=t_i+1 \\ j \notin S_{\leq t_k}}}^{t_k} P_j - \prod_{\substack{j=t_i+1 \\ j \notin S_{\leq t_k}}}^{t_k} \hat{P}_j \right) \bar{\Delta}_{t_i}. \quad (53)$$

Taking norms and applying the triangle inequality gives

$$\left\| w_{t_k}^{-S_{\leq t_k}} - w_{t_k}^{-\mathbf{S}_{1:t_k}} \right\| \leq \sum_{\substack{i \in S_{\leq t_k}}} \left\| \prod_{\substack{j=i+1 \\ j \notin S_{\leq t_k}}}^{t_k} P_j - \prod_{\substack{j=i+1 \\ j \notin S_{\leq t_k}}}^{t_k} \hat{P}_j \right\| \|\Delta_i\| + \sum_{i=s_0} \left\| \prod_{\substack{j=i+1 \\ j \notin S_{\leq t_k}}}^{t_k} P_j \right\| \mathbf{1}(i \notin S_{\leq t_k}) \|Q_i\|$$

$$+ \sum_{i=1}^{k-1} \left\| \prod_{\substack{j=t_i+1 \\ j \notin S_{\leq t_k}}}^{t_k} \hat{P}_j - \prod_{\substack{j=t_i+1 \\ j \notin S_{\leq t_k}}}^{t_k} P_j \right\| \|\bar{\Delta}_{t_i}\|. \quad (54)$$

The telescoping argument in Lemma E.4 applies analogously to products over the retained index set. Since the number of retained factors in $\prod_{\substack{j=i+1 \\ j \notin S_{\leq t_k}}}^{t_k}$ is $t_k - i - n_{[i+1,t_k]}^k$, we have

$$\left\| \prod_{\substack{j=i+1 \\ j \notin S_{\leq t_k}}}^{t_k} P_j - \prod_{\substack{j=i+1 \\ j \notin S_{\leq t_k}}}^{t_k} \hat{P}_j \right\| \leq \hat{\rho}^{t_k - i - n_{[i+1,t_k]}^k} - \rho^{t_k - i - n_{[i+1,t_k]}^k}, \quad (55)$$

and

$$\left\| \prod_{\substack{j=i+1 \\ j \notin S_{\leq t_k}}}^{t_k} P_j \right\| \leq \rho^{t_k - i - n_{[i+1,t_k]}^k}. \quad (56)$$

Similarly, for the product starting from $t_i + 1$, the number of retained factors is $t_k - t_i - n_{[t_i+1,t_k]}^k$, and hence

$$\left\| \prod_{\substack{j=t_i+1 \\ j \notin S_{\leq t_k}}}^{t_k} P_j - \prod_{\substack{j=t_i+1 \\ j \notin S_{\leq t_k}}}^{t_k} \hat{P}_j \right\| \leq \hat{\rho}^{t_k - t_i - n_{[t_i+1,t_k]}^k} - \rho^{t_k - t_i - n_{[t_i+1,t_k]}^k}. \quad (57)$$

Moreover, by the CL first-order condition and the $L$-Lipschitzness of the loss, $\|\Delta_i\| \leq \frac{L}{\lambda}$.

It remains to upper bound $\|Q_i\|$. By the definition of $Q_i$ and the $L_3$ Hessian-Lipschitz condition,

$$\|Q_i\| \leq \left\| \left( \nabla^2 \hat{F}_i(w_i^{-\mathbf{S}_{1:i-1}}) + \lambda I \right)^{-1} \right\| \|R_i\|$$

$$\leq \frac{1}{\mu + \lambda} \cdot \frac{L_3}{2} \left\| w_i^{-S_{\leq t_k}} - w_i^{-\mathbf{S}_{1:i-1}} \right\|^2. \quad (58)$$

Substituting (55), (56), (57), (58), and $\|\Delta_i\| \leq L/\lambda$ into (54) gives the desired upper bound on $\left\| w_{t_k}^{-S_{\leq t_k}} - w_{t_k}^{-\mathbf{S}_{1:t_k}} \right\|$. Combining this with the initial contraction

$$\left\| w_t^{-S_{\leq t}} - w_t^{-\mathbf{S}_{1:t}} \right\| \leq \rho^{t-t_k} \left\| w_{t_k}^{-S_{\leq t_k}} - w_{t_k}^{-\mathbf{S}_{1:t_k}} \right\|$$

completes the proof.

### E.4. Performance guarantees of the forgetting enhanced Hessian-based unlearning algorithm

**Theorem E.10.** *For each $i$-th unlearning request $t_i \in U_t = \{t_1, \ldots, t_k\}$, let $n^i_{[a,b]}$ denote the number of tasks in the time interval $[a,b]$ that have been deleted by time $t_i$. Choose $\lambda > \max\{0, \nu - \mu\}$. Then, the approximation error $\|w_t^{-S_{\le t}} - w_t^{-\mathbf{S}_{1:t}}\|$ of the unlearning operator $\mathcal{R}_{\mathcal{A}}$ in (18) is bounded by*

$$
\gamma_t(\mathbf{S}_{1:t}) := \sum_{i=1}^{k} \sum_{s \in S'_{t_i}} \left( \rho^{t-t_i-n^k_{[t_i+1,t]}} \left( \hat{\rho}^{t_i-s-n^i_{[s+1,t_i]}} - \rho^{t_i-s-n^i_{[s+1,t_i]}} \right) \frac{L(M-\mu+\nu)}{\lambda \nu} \right.
$$
$$
\left. + \frac{L(\mu+\lambda)}{\lambda \mu} \kappa \rho^{t-t_i-n^k_{[t_i+1,t]}} \left( \rho^{t_i-s-n^k_{[s+1,t_i]}} - \rho^{t_i-s-n^i_{[s+1,t_i]}} \right) \right) + \sum_{i=1}^{k} \sum_{s \in S_{t_i} \setminus S'_{t_i}} \rho^{t-s-n^k_{[s+1,t]}} \frac{L}{\lambda}. \tag{59}
$$

*When $\nu = 0$, the $\gamma_t(\mathbf{S}_{1:t})$ becomes:*

$$
\gamma_t(\mathbf{S}_{1:t}) := \sum_{i=1}^{k} \sum_{s \in S'_{t_i}} \left( \frac{L\kappa(\mu+\lambda)}{\lambda \mu} \rho^{t-t_i-n^k_{[t_i+1,t]}} \left( \rho^{t_i-s-n^k_{[s+1,t_i]}} - \rho^{t_i-s-n^i_{[s+1,t_i]}} \right) \right.
$$
$$
\left. + \frac{L(M-\mu)}{\lambda(\mu+\lambda)} \rho^{t-t_i-n^k_{[t_i+1,t]}} (t_i - s - n^i_{[s+1,t_i]}) \rho^{t_i-s-n^i_{[s+1,t_i]}} \right) + \sum_{i=2}^{k} \sum_{s \in S_{t_i} \setminus S'_{t_i}} \rho^{t-s-n^k_{[s+1,t]}} \frac{L}{\lambda}. \tag{60}
$$

*Further, the output model $\tilde{w}_t^{-\mathbf{S}_{1:t}}$ of modified Alg. 2 satisfies $(\varepsilon, \delta)$-certified continual unlearning as defined in Definition 2.1, and achieves the post-unlearning excess risk upper bound as defined in Definition 2.2:*

$$
L\left( \frac{\sqrt{2d \ln(\frac{1.25}{\delta})}}{\varepsilon} + 1 \right) \gamma_t(\mathbf{S}_{1:t}) + \mathcal{E}^{-S_{\le t}}(\lambda),
$$

*where $\mathcal{E}^{-S_{\le t}}(\lambda)$ is given in (10).*

*Proof of Theorem E.10.* By Lemma E.6, for any time $t$ after the $k$-th unlearning request and before the next unlearning request, we have $S_{\le t} = S_{\le t_k}$ and $\mathbf{S}_{1:t} = \mathbf{S}_{1:t_k}$. Hence,

$$
\left\| w_t^{-S_{\le t}} - w_t^{-\mathbf{S}_{1:t}} \right\| = \left\| \prod_{i=t_k+1}^{t} \tilde{P}_i \left( w_{t_k}^{-S_{\le t_k}} - w_{t_k}^{-\mathbf{S}_{1:t_k}} \right) \right\| \le \rho^{t-t_k} \left\| w_{t_k}^{-S_{\le t_k}} - w_{t_k}^{-\mathbf{S}_{1:t_k}} \right\|.
$$

Therefore, it suffices to upper bound $\left\| w_{t_k}^{-S_{\le t_k}} - w_{t_k}^{-\mathbf{S}_{1:t_k}} \right\|$.

We now expand $w_{t_k}^{-S_{\le t_k}} - w_{t_k}^{-\mathbf{S}_{1:t_k}}$. Consider first a retained task $i \notin S_{\le t_k}$. Lemma E.6 gives

$$
w_i^{-S_{\le t_k}} - w_i^{-\mathbf{S}_{1:i-1}} = \tilde{P}_i \left( w_{i-1}^{-S_{\le t_k}} - w_{i-1}^{-\mathbf{S}_{1:i-1}} \right). \tag{61}
$$

On the other hand, if $i \in S_{\le t_k}$, then task $i$ is skipped in the retrained trajectory, and hence

$$
w_i^{-S_{\le t_k}} - w_i^{-\mathbf{S}_{1:i-1}} = w_{i-1}^{-S_{\le t_k}} - w_{i-1}^{-\mathbf{S}_{1:i-1}} + \Delta_i. \tag{62}
$$

At an unlearning time $t_i$, the modified unlearning algorithm (18) gives

$$
w_{t_i}^{-\mathbf{S}_{1:t_i}} = w_{t_i}^{-\mathbf{S}_{1:t_i-1}} + \bar{\Delta}_{t_i},
$$

where $\bar{\Delta}_{t_i} = \sum_{s \in S'_{t_i}} \prod_{\substack{j=s+1 \\ j \notin S_{\le t_i}}}^{t_i} \hat{P}_j \Delta_s$. Thus the approximation error is reduced by $\bar{\Delta}_{t_i}$:

$$
w_{t_i}^{-S_{\le t_k}} - w_{t_i}^{-\mathbf{S}_{1:t_i}} = w_{t_i}^{-S_{\le t_k}} - w_{t_i}^{-\mathbf{S}_{1:t_i-1}} - \bar{\Delta}_{t_i}.
$$

Iterating (61) and (62) from the earliest deleted task up to $t_k$, and subtracting the correction terms at all unlearning times, gives

$$w_{t_k}^{-S_{\leq t_k}} - w_{t_k}^{-\mathbf{S}_{1:t_k}} = \sum_{i=1}^{k} \sum_{s \in S_{t_i}} \prod_{\substack{a=s+1 \\ a \notin S_{\leq t_k}}}^{t_k} \tilde{P}_a \Delta_s - \sum_{i=1}^{k} \prod_{\substack{a=t_i+1 \\ a \notin S_{\leq t_k}}}^{t_k} \tilde{P}_a \bar{\Delta}_{t_i}. \tag{63}$$

Substituting the definition of $\bar{\Delta}_{t_i}$ into (63), we obtain

$$w_{t_k}^{-S_{\leq t_k}} - w_{t_k}^{-\mathbf{S}_{1:t_k}} = \sum_{i=1}^{k} \sum_{s \in S'_{t_i}} \prod_{\substack{a=t_i+1 \\ a \notin S_{\leq t_k}}}^{t_k} \tilde{P}_a \left( \prod_{\substack{j=s+1 \\ j \notin S_{\leq t_k}}}^{t_i} \tilde{P}_j - \prod_{\substack{j=s+1 \\ j \notin S_{\leq t_i}}}^{t_i} \hat{P}_j \right) \Delta_s + \sum_{i=1}^{k} \sum_{s \in S_{t_i} \setminus S'_{t_i}} \prod_{\substack{a=s+1 \\ a \notin S_{\leq t_k}}}^{t_k} \tilde{P}_a \Delta_s. \tag{64}$$

Taking norms in (64) yields

$$\left\| w_{t_k}^{-S_{\leq t_k}} - w_{t_k}^{-\mathbf{S}_{1:t_k}} \right\| \leq \sum_{i=1}^{k} \sum_{s \in S'_{t_i}} \left\| \prod_{\substack{a=t_i+1 \\ a \notin S_{\leq t_k}}}^{t_k} \tilde{P}_a \right\| \left\| \prod_{\substack{j=s+1 \\ j \notin S_{\leq t_k}}}^{t_i} \tilde{P}_j - \prod_{\substack{j=s+1 \\ j \notin S_{\leq t_i}}}^{t_i} \hat{P}_j \right\| \|\Delta_s\|$$

$$+ \sum_{i=1}^{k} \sum_{s \in S_{t_i} \setminus S'_{t_i}} \left\| \prod_{\substack{a=s+1 \\ a \notin S_{\leq t_k}}}^{t_k} \tilde{P}_a \right\| \|\Delta_s\|. \tag{65}$$

For the first product in the first term,

$$\left\| \prod_{\substack{a=t_i+1 \\ a \notin S_{\leq t_k}}}^{t_k} \tilde{P}_a \right\| \leq \rho^{t_k - t_i - n^k_{[t_i+1, t_k]}}.$$

The same telescoping argument as in Lemma E.4 applies to products over the retained index set, with the number of retained factors equal to $t_i - s - n^i_{[s+1, t_i]}$. Thus, for the difference between the exact and approximate Hessian products, Lemma E.4 and Lemma E.5 gives

$$\left\| \prod_{\substack{j=s+1 \\ j \notin S_{\leq t_k}}}^{t_i} \tilde{P}_j - \prod_{\substack{j=s+1 \\ j \notin S_{\leq t_i}}}^{t_i} \hat{P}_j \right\|$$

$$\leq \left\| \prod_{\substack{j=s+1 \\ j \notin S_{\leq t_k}}}^{t_i} \tilde{P}_j - \prod_{\substack{j=s+1 \\ j \notin S_{\leq t_i}}}^{t_i} \tilde{P}_j \right\| + \left\| \prod_{\substack{j=s+1 \\ j \notin S_{\leq t_i}}}^{t_i} \tilde{P}_j - \prod_{\substack{j=s+1 \\ j \notin S_{\leq t_i}}}^{t_i} \hat{P}_j \right\|$$

$$\leq \frac{\mu + \lambda}{\mu} \kappa \left( \rho^{t_i - s - n^k_{[s+1, t_i]}} - \rho^{t_i - s - n^i_{[s+1, t_i]}} \right) + \frac{M - \mu + \nu}{\nu} \left( \hat{\rho}^{t_i - s - n^i_{[s+1, t_i]}} - \rho^{t_i - s - n^i_{[s+1, t_i]}} \right). \tag{66}$$

Moreover, the first-order optimality condition of the CL update and the $L$-Lipschitz condition implies

$$\|\Delta_s\| = \left\| w_{s-1}^{-\mathbf{S}_{1:s-1}} - w_s^{-\mathbf{S}_{1:s-1}} \right\| \leq \frac{L}{\lambda}.$$

Therefore, the first term in (65) is bounded by

$$\sum_{i=1}^{k} \sum_{s \in S'_{t_i}} \rho^{t_k - t_i - n^k_{[t_i+1, t_k]}} \left( \hat{\rho}^{t_i - s - n^i_{[s+1, t_i]}} - \rho^{t_i - s - n^i_{[s+1, t_i]}} \right) \frac{L(M - \mu + \nu)}{\lambda \nu}$$

$$+ \frac{L(\mu + \lambda)}{\lambda \mu} \kappa \rho^{t_k - t_i - n^k_{[t_i+1, t_k]}} \left( \rho^{t_i - s - n^k_{[s+1, t_i]}} - \rho^{t_i - s - n^i_{[s+1, t_i]}} \right). \tag{67}$$

Similarly, for the second term in (65),

$$\sum_{i=1}^{k} \sum_{s \in S_{t_i} \setminus S'_{t_i}} \left\| \prod_{\substack{a=s+1 \\ a \notin S_{\leq t_k}}}^{t_k} \tilde{P}_a \right\| \|\Delta_s\| \leq \sum_{i=1}^{k} \sum_{s \in S_{t_i} \setminus S'_{t_i}} \rho^{t_k - s - n^k_{[s+1, t_k]}} \frac{L}{\lambda}. \tag{68}$$

Combining (67) and (68), we obtain

$$\left\| w_t^{-S_{\leq t}} - w_t^{-\mathbf{S}_{1:t}} \right\| \leq \sum_{i=1}^{k} \sum_{s \in S'_{t_i}} \left( \rho^{t - t_i - n^k_{[t_i+1, t]}} \left( \hat{\rho}^{t_i - s - n^i_{[s+1, t_i]}} - \rho^{t_i - s - n^i_{[s+1, t_i]}} \right) \frac{L(M - \mu + \nu)}{\lambda \nu} \right.$$
$$\left. + \frac{L(\mu + \lambda)}{\lambda \mu} \kappa \rho^{t - t_i - n^k_{[t_i+1, t]}} \left( \rho^{t_i - s - n^k_{[s+1, t_i]}} - \rho^{t_i - s - n^i_{[s+1, t_i]}} \right) \right)$$
$$+ \sum_{i=1}^{k} \sum_{s \in S_{t_i} \setminus S'_{t_i}} \rho^{t - s - n^k_{[s+1, t]}} \frac{L}{\lambda}.$$

This is exactly the bound in (59).

When $\nu = 0$, Lemma E.4 and Lemma E.5 gives

$$\left\| \prod_{\substack{j=s+1 \\ j \notin S_{\leq t_k}}}^{t_i} \tilde{P}_j - \prod_{\substack{j=s+1 \\ j \notin S_{\leq t_i}}}^{t_i} \hat{P}_j \right\| \leq \frac{\mu + \lambda}{\mu} \kappa (\rho^{t_i - s - n^k_{[s+1, t_i]}} - \rho^{t_i - s - n^i_{[s+1, t_i]}}) + \frac{M - \mu}{\mu + \lambda} (t_i - s - n^i_{[s+1, t_i]}) \rho^{t_i - s - n^i_{[s+1, t_i]}}.$$

Substituting this bound into (65) and repeating the same steps gives (60).

Finally, since $\gamma_t(\mathbf{S}_{1:t})$ upper bounds $\|w_t^{-S_{\leq t}} - w_t^{-\mathbf{S}_{1:t}}\|$, the Gaussian mechanism with

$$\sigma = \gamma_t(\mathbf{S}_{1:t}) \frac{\sqrt{2 \ln(1.25/\delta)}}{\varepsilon}$$

guarantees $(\varepsilon, \delta)$-certified continual unlearning. The post-unlearning excess risk bound follows by the same Lipschitz decomposition as in the proof of Theorem 4.1:

$$\mathcal{E}_{\mathrm{LU}} \leq L \left( \frac{\sqrt{2d \ln(1.25/\delta)}}{\varepsilon} + 1 \right) \gamma_t(\mathbf{S}_{1:t}) + \mathcal{E}^{-S_{\leq t}}(\lambda).$$

This completes the proof. $\qquad \square$

### E.5. Stronger certified unlearning

**Corollary E.11.** *For each $i$-th unlearning request $t_i \in U_t = \{t_1, \ldots, t_k\}$, let $n^i_{[a,b]}$ denote the number of tasks in the time interval $[a, b]$ that have been deleted by time $t_i$. Choose $\lambda > \max\{0, \nu - \mu\}$. Then, with probability at least $1 - \delta_2$, the approximation error $\|w_t^{-S_{\leq t}} - \tilde{w}_t^{-\mathbf{S}_{1:t}}\|$ in Alg. 3 is bounded by*

$$\gamma_t(\mathbf{S}_{1:t}) := \sum_{i=1}^{k} \sum_{s \in S'_{t_i}} (1 + \xi)^{k-i+1} \left( \rho^{t - t_i - n^k_{[t_i+1, t]}} \left( \hat{\rho}^{t_i - s - n^i_{[s+1, t_i]}} - \rho^{t_i - s - n^i_{[s+1, t_i]}} \right) \frac{L(M - \mu + \nu)}{\lambda \nu} \right.$$
$$\left. + \frac{L(\mu + \lambda)}{\lambda \mu} \kappa \rho^{t - t_i - n^k_{[t_i+1, t]}} \left( \rho^{t_i - s - n^k_{[s+1, t_i]}} - \rho^{t_i - s - n^i_{[s+1, t_i]}} \right) \right) + \sum_{i=1}^{k} \sum_{s \in S_{t_i} \setminus S'_{t_i}} \rho^{t - s - n^k_{[s+1, t]}} \frac{L}{\lambda} (1 + \xi)^{k-i+1}.$$

$$\tag{69}$$

---

**Algorithm 3** Forgetting-enhanced Hessian-based CLU with stronger certified guarantee

---

1: Initialize $U_0 = \emptyset$, $S_{\leq 0} = \emptyset$, $\mathbf{S}_{1:0} = \emptyset$.
2: **for** $t = 1$ to $T$ **do**
    **Stage I. learning and precomputation on task** $t$
3:    Receive $D_t$, update $w_t^{-\mathbf{S}_{1:t-1}} \leftarrow \ell_2\text{-CL}(w_{t-1}^{-\mathbf{S}_{1:t-1}}, D_t)$.
4:    Compute and store $\Delta_t$ and $\hat{H}_t$ defined in (15).
    **Stage II. system unlearning and model publishing**
5:    Receive deletion request $S_t \subseteq [t] \setminus S_{\leq t-1}$.
6:    **if** $S_t \neq \emptyset$ **then**
7:        Update $S_{\leq t} \leftarrow S_t \cup S_{\leq t-1}$, $\mathbf{S}_{1:t} \leftarrow (\mathbf{S}_{1:t-1}, S_t)$, and $U_t \leftarrow \{t\} \cup U_{t-1}$.
8:        Set $w_t^{-\mathbf{S}_{1:t}} \leftarrow \mathcal{R}_\mathcal{A}\left(w_t^{-\mathbf{S}_{1:t-1}}, \mathbf{D}_{1:t}, \mathbf{S}_{1:t}\right)$ with $\mathcal{R}_\mathcal{A}$ in (18).
9:        Discard all the stored $\hat{H}_\tau$ and $\Delta_\tau$ for $\tau < t$.
10:      Draw $\epsilon_t \sim \mathcal{N}(\mathbf{0}, \sigma^2 I)$ with $\sigma = \gamma_t(\mathbf{S}_{1:t})\frac{\sqrt{2\ln(1.25/\delta)}}{\varepsilon}$ and $\gamma_t(\mathbf{S}_{1:t})$ defined in (69), and $w_t^{-\mathbf{S}_{1:t}} \leftarrow w_t^{-\mathbf{S}_{1:t}} + \epsilon_t$.
11:      **Output** $\tilde{w}_t^{-\mathbf{S}_{1:t}} \leftarrow w_t^{-\mathbf{S}_{1:t}}$.
12:    **else**
13:        Set $S_{\leq t} \leftarrow S_{\leq t-1}$, $\mathbf{S}_{1:t} \leftarrow \mathbf{S}_{1:t-1}$, $U_t \leftarrow U_{t-1}$, and $w_t^{-\mathbf{S}_{1:t}} \leftarrow w_t^{-\mathbf{S}_{1:t-1}}$.
14:      **Output** $\tilde{w}_t^{-\mathbf{S}_{1:t}} \leftarrow w_t^{-\mathbf{S}_{1:t}}$.
15:    **end if**
16: **end for**

---

*Further, with probability at least $1 - \delta_2$, the output model $\tilde{w}_t^{-\mathbf{S}_{1:t}}$ and the internal state of the system in Alg. 3 satisfies $(\varepsilon, \delta)$-certified continual unlearning as defined in Definition 2.1. $\tilde{w}_t^{-\mathbf{S}_{1:t}}$ achieves the post-unlearning excess risk upper bound as defined in Definition 2.2:*

$$L\left(\frac{\sqrt{2d\ln(\frac{1.25}{\delta})}}{\varepsilon} + 1\right)\gamma_t(\mathbf{S}_{1:t}) + \mathcal{E}^{-S_{\leq t}}(\lambda),$$

*where $\mathcal{E}^{-S_{\leq t}}(\lambda)$ is given in Theorem 3.1.*

## F. Experiment details

### F.1. Implementation details

All experiments were conducted on a workstation equipped with an AMD Ryzen 9 9950X 16-Core Processor and an NVIDIA GeForce RTX 5080 GPU with 16GB memory.

**CIFAR-100.** For the main CIFAR-100 experiments, we construct a non-i.i.d. continual learning benchmark with $T = 30$ sequential tasks. Each task contains samples from a randomly selected subset of 80–100 classes. For each class, we first randomly permute all available samples and then evenly allocate them across the tasks in which that class appears, ensuring that each task contains a different class mixture while preserving balanced usage of class samples across the sequence.

We use a ResNet-18 (He et al., 2016) backbone and replace its final classification layer with a three-layer fully connected head. The head has hidden dimensions 512 and 256, followed by a 100-dimensional output layer for CIFAR-100 classification. The pretrained backbone is frozen throughout training and unlearning, so only the classification head is updated. This gives trainable parameter dimension $d = 419{,}684$ and output dimension $v = 100$.

For each sample, the loss $\ell$ is the cross-entropy loss with an additional weight-decay term $10^{-4}\|w\|_2^2/2$. On each task, we train the objective in ($\ell_2$-CL) using Adam with learning rate 0.002, batch size 256, and 200 epochs per task. We use a StepLR scheduler with step size 500 and decay factor 0.5.

**CIFAR-10 and MNIST.** For CIFAR-10 and MNIST, we construct non-i.i.d. continual learning benchmarks with $T = 10$ sequential tasks. For CIFAR-10, each task contains samples from a randomly selected subset of 9-10 classes; for MNIST, each task contains samples from a randomly selected subset of 5–10 digit classes. For each class, we randomly permute all

available training samples and evenly allocate them across the tasks in which that class appears, ensuring different class mixtures across tasks while preserving balanced usage of class samples.

We use a three-block convolutional neural network for both datasets. The convolutional layers have channel sizes 32, 64, and 128, each followed by ReLU and $2 \times 2$ max pooling. The feature map is flattened and passed through a fully connected layer with hidden dimension 256, followed by ReLU, dropout with rate 0.2, and a 10-dimensional output layer. For CIFAR-10, the first convolutional layer takes three input channels and the flattened feature dimension is $128 \times 4 \times 4$. For MNIST, the first convolutional layer takes one input channel and the flattened feature dimension is $128 \times 3 \times 3$. CIFAR-10 images are normalized with mean $(0.4914, 0.4822, 0.4465)$ and standard deviation $(0.2023, 0.1994, 0.2010)$, while MNIST images are normalized with mean $0.1307$ and standard deviation $0.3081$.

For each sample, the loss $\ell$ is the cross-entropy loss plus the term $\frac{\lambda_{\mathrm{sc}}}{2}\|w\|_2^2$, where $\lambda_{\mathrm{sc}} = 10^{-4}$. On each task, CIFAR-10 is trained using Adam with learning rate 0.001, batch size 128, and 200 epochs per task. MNIST is trained using Adam with learning rate $2 \times 10^{-4}$, batch size 128, and 50 epochs per task. For both datasets, the StepLR scheduler has step size 500 and is stepped after every mini-batch update. The decay factor is 0.5 for CIFAR-10 and 1 for MNIST.

For certified unlearning, we adopt the commonly used privacy parameters $\varepsilon = 8$ and $\delta = 10^{-6}$ to calibrate the noise used in Fig. 5.

### F.2. Parameter estimation for theoretical bounds in Table 2 and Fig. 5

For the CIFAR-100 model, we estimate the constants $L$, $M$, $\mu$ and $\nu$ used in Theorem 4.1 and Proposition 5.4. Let $w_0$ denote the initial trainable parameter vector of the CIFAR-100 model. We first construct a local parameter region around initialization,

$$\mathcal{B}(w_0, 25) = \{w : \|w - w_0\|_2 \le 25\},$$

and sample 100 parameter vectors from this ball. Specifically, for each sample, we draw a random Gaussian direction $u$, normalize it to unit norm, draw a radius $r \sim \mathrm{Unif}(0, 25)$, and set

$$w = w_0 + r \frac{u}{\|u\|_2}.$$

At each sampled parameter vector, we temporarily load $w$ into the model and evaluate all quantities below using the same loss and regularization as in training.

For each sampled point $w$, we estimate the Hessian spectrum of the regularized objective

$$\hat{F}_t(w) + \frac{\lambda_{\mathrm{sc}}}{2}\|w\|_2^2,$$

where $\lambda_{\mathrm{sc}} = 10^{-4}$. Since explicitly forming the Hessian is infeasible, we use Hessian-vector products. Given a vector $v$, we compute $H(w)v$ by automatic differentiation. We then run matrix-free power/Lanczos iteration to estimate the largest and smallest algebraic eigenvalues,

$$\lambda_{\max}(H(w)), \qquad \lambda_{\min}(H(w)).$$

After repeating this over all 100 sampled points, we take conservative rounded values

$$M \ge \max_w \lambda_{\max}(H(w)), \qquad \mu \le \min_w \lambda_{\min}(H(w)).$$

To estimate the Hessian approximation error $\nu$, we compare the exact Hessian operator with the approximation operators used by the unlearning algorithms. For the diagonal approximation $D(w)$, we estimate the spectral norm of the operator difference $H(w) - D(w)$ by power iteration, where each iteration only requires products of the form

$$(H(w) - D(w))v = H(w)v - D(w)v.$$

Similarly, for the Gauss–Newton approximation $G(w)$, we estimate

$$\|H(w) - G(w)\|_2$$

using matrix-free products

$$(H(w) - G(w))v = H(w)v - G(w)v.$$

We repeat this at all sampled parameter vectors and set

$$\nu_{\mathrm{diag}} \geq \max_w \|H(w) - D(w)\|_2, \qquad \nu_{\mathrm{G\text{-}N}} \geq \max_w \|H(w) - G(w)\|_2,$$

again rounding upward to obtain conservative constants.

Finally, we estimate the Lipschitz constant $L$ of the objective over the same sampled region. At each sampled parameter point, we evaluate finite-difference ratios between sampled parameter pairs,

$$\frac{|\hat{F}_t(w) - \hat{F}_t(w')|}{\|w - w'\|_2},$$

and take the largest observed value over the sampled region and round it upward before using it in the bound computation.

*Table 3.* Constants used to instantiate the CIFAR-100 theoretical bounds.

| Quantity | Value |
|---|---|
| Hessian spectral upper bound $M$ | 5 |
| Hessian spectral lower bound $\mu$ | -0.8 |
| Diagonal Hessian error $\nu_{\mathrm{diag}}$ | 4.9 |
| Gauss–Newton Hessian error $\nu_{\mathrm{G\text{-}N}}$ | 4.7 |
| Lipschitz constant $L$ | 1.17 |

### F.3. Unlearning request sequences

Table 4 gives the two CIFAR-100 unlearning sequences used to study the temporal-disorder effect in Fig. 3. Both sequences eventually delete the same task set. Fwd-Sync groups requests so that newly deleted tasks follow the previous unlearning time, while Async delays part of the same deletions to later times.

*Table 4.* Forward-synchronous and asynchronous CIFAR-100 unlearning sequences.

| Unlearning time | Fwd-Sync unlearning tasks | Async unlearning tasks |
|---|---|---|
| 9 | 3,4,6,8 | 3,4,6,8 |
| 15 | 10,11,12,13,14 | 11 |
| 17 | ∅ | 12 |
| 18 | ∅ | 13 |
| 20 | ∅ | 14 |
| 25 | 20,23 | 23 |
| 26 | 24,25 | 20,25 |
| 28 | ∅ | 10,24 |

### F.4. Additional MNIST and CIFAR-10 results

Fig. 6 reports the smaller-scale MNIST and CIFAR-10 experiments. Across both datasets, Hessian-based unlearning consistently reduces approximation error compared with natural forgetting. The exact, Gauss–Newton, and diagonal Hessian variants have similar trends, suggesting that diagonal Hessian storage is often sufficient in these settings. The final-error sweep over $\lambda$ further shows that larger regularization reduces unlearning error by limiting parameter drift.

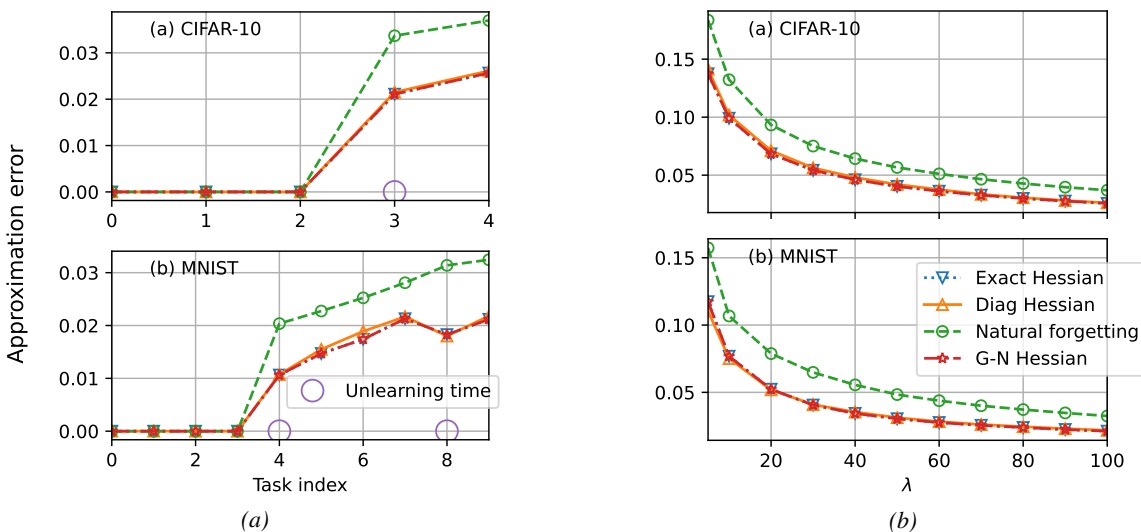

*Figure 6.* Additional approximation-error experiments on MNIST and CIFAR-10. (a) Error across task index. (b) Final error versus $\lambda$.

### F.5. Retention–unlearning trade-off

To illustrate the tension between retaining knowledge and reducing unlearning loss, we run CIFAR-100 experiments with different $\lambda$. Without unlearning, we evaluate the final test accuracy of the $\ell_2$-regularized CL model; with unlearning, we evaluate the final empirical approximation error using the Gauss-Newton Hessian method and natural forgetting. Increasing $\lambda$ decreases the unlearning error because model drift is more strongly controlled, but it also decreases pure-CL test accuracy because the learner becomes less adaptive to new tasks.

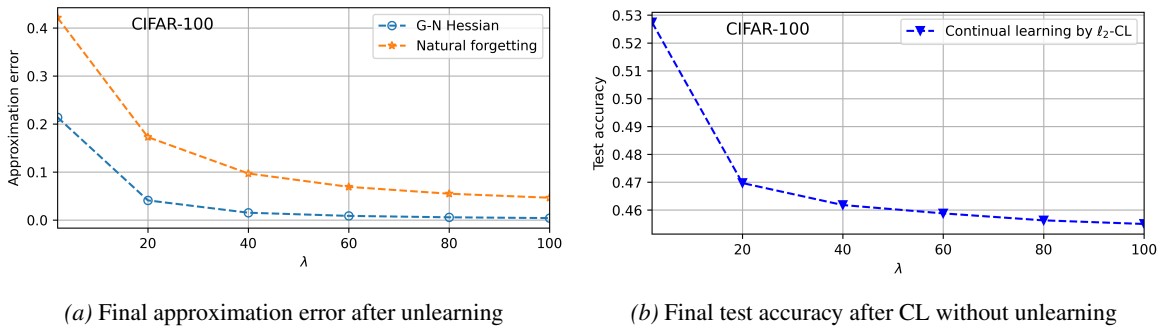

*(a)* Final approximation error after unlearning      *(b)* Final test accuracy after CL without unlearning

*Figure 7.* (a) Final approximation error $\|w_T^{-\mathbf{S}_{1:T}} - w_T^{-S_{\leq T}}\|$ versus the regularization parameter $\lambda$, after unlearning by Alg. 1 and Alg. 2. (b) Final test accuracy versus the regularization parameter $\lambda$, after training by the $\ell_2$-CL algorithm without unlearning.

