# OpenReview forum: "The Forgetting-Retention Dilemma: Certified Unlearning Theory in Continual Learning"
_ICML.cc/2026/Conference — ICML 2026 regular_

### Official Review · Reviewer_jMs5 · 2026-03-12

**Soundness:** 3
**Presentation:** 2
**Significance:** 3
**Originality:** 3
**Overall Recommendation:** 5
**Confidence:** 3

**Summary:**

This paper studies certified unlearning in the continual learning setting. The authors argue that existing certified unlearning methods are mainly designed for static training, whereas continual learning introduces a harder setting in which the model is updated sequentially over non-i.i.d. tasks and deletion requests may arrive after many later updates. The paper formalizes this setting as continual learning-unlearning (CLU) and introduces a post-unlearning objective that captures both continual-learning retention and unlearning quality. On the method side, the paper first analyzes a simple regularized continual learning procedure, then adapts a natural-forgetting-based certified unlearning approach and a Hessian-based certified unlearning approach to the continual setting, and finally proposes a hybrid variant that uses natural forgetting for older tasks and Hessian-based correction for more recent requests in order to reduce storage cost. Experiments on MNIST and CIFAR-10 are used to compare these variants in terms of approximation error and post-unlearning performance.

**Compliance With Llm Reviewing Policy:**

Affirmed.

**Final Justification:**

The paper brings some insights into the area of continual unlearning. Also, the rebuttal addressed my main concerns.

**Key Questions For Authors:**

1. In the experiments, you mention that you use the simulated exact approximation error rather than the theoretical upper bound to calibrate the noise. How much of the reported advantage remains when the full theoretically justified calibration is used? A strong answer here would increase my confidence that the empirical results truly validate the certified framework rather than an idealized approximation.

2. The current experiments are limited to MNIST and CIFAR-10 with small CNNs and only a small number of deletion events. Can you provide additional evidence on larger models, more deletion requests, or more challenging task sequences?

3. The hybrid method is practically appealing, but the paper gives limited guidance on when to switch between natural forgetting and Hessian-based correction. Can you provide either a principled rule or a clearer empirical study of this threshold?

4. The paper compares against a limited set of baselines. Can you clarify how your certified methods compare, conceptually or empirically, to stronger continual-unlearning baselines that may not be certified but do attempt to jointly balance retention and deletion?

**Limitations:**

No. The paper includes only limited discussion of limitations. I would encourage the authors to more explicitly acknowledge the narrow experimental scope, the gap between the theoretical guarantees and the empirical calibration protocol, the storage and computational cost of Hessian-based methods for larger models, and the dependence of the theory on local assumptions that may not hold broadly in realistic nonconvex settings.

**Strengths And Weaknesses:**

Strengths.

- The paper addresses an interesting and timely problem. Bringing certified unlearning into continual learning is an important and nontrivial direction, and the paper clearly motivates why standard static unlearning methods do not directly apply once models are updated sequentially over time.

- The problem formulation is one of the strongest aspects of the paper. I found the framing of the forgetting-retention trade-off intuitive and useful. The paper makes clear that continual learning aims to preserve prior knowledge, while unlearning aims to remove selected past information, and it builds the theory around this tension.

- The paper presents a coherent technical story rather than only a heuristic method. It first analyzes the continual learning backbone, then studies certified unlearning in that setting, and then proposes both a stronger Hessian-based approach and a more storage-efficient hybrid strategy. This gives the paper a well-structured progression from formulation to analysis to algorithms.

- I appreciated the contrast between the storage-free natural forgetting approach and the more accurate but more expensive Hessian-based approach. This trade-off is practically meaningful, and the hybrid method is a reasonable attempt to balance storage cost and post-unlearning performance.

- The paper explicitly considers asynchronous or arbitrarily ordered unlearning requests, which is a genuinely important challenge in continual learning and helps distinguish this setting from simpler static or one-shot unlearning settings.


Weaknesses.

- The experimental evaluation is too narrow relative to the scope of the paper’s claims. The experiments are limited to MNIST and CIFAR-10, small CNN models, and only 1–2 random unlearning requests. This is enough for a proof of concept, but it is not sufficient to fully support broader claims about certified unlearning in continual learning more generally.

- There is a noticeable gap between the formal theory and the empirical validation. In the experiments, the paper states that it uses the simulated exact approximation error to design the noise, rather than the theoretical upper bound, in order to demonstrate empirical performance. This makes the experimental validation less aligned with the exact certified procedure analyzed in the theory.

- The originality is stronger in the formulation and analysis than in the core algorithmic primitives. My impression is that the main novelty lies in adapting and analyzing certified unlearning ideas in the continual learning setting, rather than introducing a fundamentally new unlearning mechanism. This is still a meaningful contribution, but it somewhat limits the paper’s overall originality.

---

> ### Author Rebuttal · Authors · 2026-03-30
>
> We sincerely appreciate your time and thoughtful feedback. Below is our detailed response.
>
> 1.Additional experiments results: We have added extensive new CIFAR-100 experimental results and analysis with more deletion events. Please refer to Response 1, 2, 5 to Reviewer PiXW, and Response 2 to Reviewer HQo4 for more details.
>
> 2.Calibrating noise under $\gamma_t$: We conduct additional experiments to calibrate the noise level using our theoretical results for $\gamma_t$. In the neural network setting, we empirically estimate $\mu, M, \nu, L$ using matrix-free power iteration. Specifically, we sample 100 parameter points from the Euclidean ball of radius 25 centered at the initial parameter vector and estimate these quantities as follows: $M=5,\mu=-0.8, L=0.08$, $\nu=4.9$ for diagonal Hessian and $4.7$ for Gauss-Newton Hessian.
>
> Using these parameter values, we compute the approximation error bound $\gamma_t$ established in Thm. 4.1 and Prop. 5.4. We then calibrate the noise based on this theoretical $\gamma_t$ and evaluate the final test accuracy. The results below compare the theoretical bound $\gamma_t$, the empirical approximation error $\\|w_t^{-S_{\le t}} - w_t^{-\mathbf{S}_{1:t}}\\|$, and the test accuracy under the corresponding $\gamma_t$ noise.
> $$\\begin{array}{lccc}\\hline\\mathrm{Method} & \\mathrm{Test\\ Accuracy} & \\mathrm{\\gamma\\ upper\\ bound} & \\mathrm{Empirical\\ approximation\\ error} \\\\\\hline\\mathrm{Retraining} & 0.4491 & 0 & 0 \\\\
> \\mathrm{Diag\\ Hessian} & 0.4227 & 0.0319 & 0.0037 \\\\\\mathrm{G\\text{-}N\\ Hessian} & 0.4293 & 0.0310 & 0.0040 \\\\\\mathrm{Natural\\ forgetting} & 0.3056 & 0.086 & 0.0465 \\\\\\hline\\end{array}$$The results show that our theoretical bound indeed upper-bounds the empirical error. Under noise calibrated from the theoretical upper bound, both Hessian-based unlearning methods outperform natural forgetting in test accuracy and remain useful and comparable to retraining.
>
> 3.Hybrid method: In our hybrid method, the switch between natural forgetting and the Hessian-based update is determined by the most recent unlearning time $t'$. At unlearning time $t$, for requests $S_t$, we use natural forgetting for tasks in $S_t$ trained before $t'$, and apply the Hessian-based update in Eq. (18) to tasks in $S_t$ trained at or after $t'$. For example, under forward-synchronous unlearning, as shown in Fig. 2 of the main paper, all requests fall into the latter case, so the hybrid method reduces to Hessian-based method. Please further find our new experiments on the hybrid method in Response 2 to Reviewer HQo4.
>
> 4.More baseline methods: Thanks for the valuable comments, and we have considered other baselines for comparison. However, we would like to clarify two points. First, certified unlearning uniquely provides strong guarantees that verify unlearning is genuine. Recent work "Unlearning or Obfuscating? Jogging the Memory of Unlearned LLMs via Benign Relearning (ICLR 2025)", shows that methods without such guarantees can only give the illusion of unlearning. In certified unlearning, we take minimizing the approximation error to the retraining model as objective. Without such an objective, as reported in Section 6 (left column, starting from line 436), the approximation error is shown to far exceed 20, compared to our Fig. 3 result of less than 0.2, making non-certified methods not directly comparable to our approach.
>
> Second, unlearning in continual learning is still a very new problem, and most recent certified unlearning methods are developed for the static setting, where the deletion request is applied to samples from the same dataset on which the model has just been trained, and are not comparable to our approach, e.g., "Siqiao Mu and Diego Klabjan (NeurIPS2025)" and "Qiao, Xinbao, et al. (ICLR 2025)". These methods are not designed for the CL setting, where model evolves across task and past data are no longer accessible. Therefore, casting the unlearning problem studied in our paper into their setting is fundamentally inapplicable.
>
> 5.Originality: Thanks for acknowledging our contribution in analyzing unlearning in CL. We believe our work provides a new mechanism that enables certified unlearning in CL for the first time. Our analysis reveals the distinction between asynchronous and forward-synchronous unlearning as in Fig. 2, a phenomenon unique to the CL setting. Furthermore, our framework introduces a new Hessian-based mechanism in Eq. (14) for forgetting tasks learned several stages earlier, which is not captured by existing certified unlearning works.
>
> 6.Assumptions: We would like to clarify that our local assumptions are mild and are satisfied in most realistic nonconvex settings. Please refer to our Response 3 to Reviewer HQo4 for details.
>
> If our clarifications and additional results have addressed your concerns, we would greatly appreciate your consideration in updating your rating. We would also be happy to answer any further questions.

---

> > ### Author Rebuttal · Reviewer_jMs5 · 2026-04-03
> >
> > Thanks for your detailed rebuttal. I believe this solves all my concerns, and I have raised my score.

---

> > > ### Author Response · Authors · 2026-04-07
> > >
> > > Thank you very much for your careful review and for acknowledging our work.

---

### Official Review · Reviewer_HQo4 · 2026-03-12

**Soundness:** 3
**Presentation:** 2
**Significance:** 3
**Originality:** 3
**Overall Recommendation:** 4
**Confidence:** 3

**Summary:**

This paper studies certified unlearning in the continual learning (CL) setting. The main motivation is that existing certified unlearning methods are largely developed for static training settings and are not well suited to continual learning, where the model evolves sequentially across tasks and unlearning requests may arrive at different time steps. To address this, the authors propose a continual learning-unlearning (CLU) framework, define the objective as minimizing the post-unlearning excess risk, and decompose it into CL excess risk and unlearning loss. This formulation captures the fundamental trade-off between preserving historical knowledge and selectively forgetting targeted data.

**Compliance With Llm Reviewing Policy:**

Affirmed.

**Final Justification:**

This paper is well-structured, rigorously argued, and contains no glaring flaws.

**Key Questions For Authors:**

As noted in the weaknesses, please clarify the following points:

1. the justification for the current experimental scale and setup, especially given the broader motivation described in the paper;

2. whether the assumptions used in the theoretical analysis are intended to be realistic in modern CL/LLM settings, and how the conclusions may be affected when these assumptions are violated in practice.

**Limitations:**

yes

**Strengths And Weaknesses:**

Strengths
1. The paper addresses an important and practically relevant problem.

2. The theoretical analysis is relatively complete and provides a structured discussion of two different unlearning approaches.

Weaknesses
1.The experimental evaluation is limited in scale and remains somewhat disconnected from the broader scenarios described in the paper. The analysis is not sufficiently comprehensive, as it mainly presents overall results without adequately examining the contribution of individual components or providing stronger comparative studies. In addition, the experimental section lacks enough detail to fully support reproducibility.

2.The theoretical analysis relies on fairly strong local assumptions, which appear somewhat idealized for practical applications. The paper does not sufficiently discuss the validity or robustness of these assumptions in realistic settings.

3.The paper is not very easy to follow. The presentation contains a heavy accumulation of theoretical notation and definitions, which reduces overall clarity and readability.

---

> ### Author Rebuttal · Authors · 2026-03-30
>
> We sincerely appreciate your time and thoughtful feedback. Below is our detailed response.
>
> 1.Additional experiments results on CIFAR-100: According to your comments, we have added Fig. A, B, C in the supplementary material
> https://anonymous.4open.science/r/New-experiments-CLU-8956/New_results.pdf and three tables in our responses to provide comprehensive comparative experimental studies. Please refer to Response 1 to Reviewer PiXW for the experimental setup, as well as Responses 2 and 5 to Reviewer PiXW and Response 2 to Reviewer jMs5 for further experiment details.
>
> 2.New experiments with runtime, storage and approximation error: We compare runtime, peak GPU memory, and unlearning approximation error across the following methods to examine individual components and provide stronger comparative studies: our Alg. 2 with diagonal Hessian and Gauss-Newton Hessian, their forgetting-enhanced variants to save memory in Eq. (18), natural forgetting, and full retraining. We omit exact Hessian experiments because computing and storing all exact Hessians for the model are too consuming to implement in practice. The unlearning approximation error $\\|w_t^{-\\mathbf{S}\_{1:t}}-w_t^{-S_{\\leq t}}\\|$ over time $t$ is shown in Fig. A of the supplementary material. The runtime, memory, and final approximation error results are reported below.
> $$\\begin{array}{l|cccccc}\\hline\\mathrm{Metric} & \\mathrm{Retraining} & \\text{G-N}\\mathrm{\\ Hessian} & \\text{Enhanced\\ G-N\\ Hessian} & \\text{Diag\\ Hessian} & \\text{Enhanced\\ diag\\ Hessian} & \\text{Natural\\ forgetting}\\\\\\hline
> \\mathrm{Time\\ (s)} & 950.69 & 269.11 & 264.84 & 283.69 & 277.91 & 260.77\\\\\\mathrm{Peak\\ GPU\\ Mem.\\ (MB)} & 932.10 & 5678.90 & 2232.40 & 1129.66 & 1113.52 & 974.30\\\\\\text{Empirical\\ approximation\\ error} & 0 & 0.0087 & 0.0463 & 0.0094 & 0.0451 & 0.0835\\\\\\hline\\end{array}$$
> Full retraining is roughly three times slower than all unlearning methods. Alg. 2 with the Gauss-Newton and diagonal-Hessian variants achieves the smallest approximation errors relative to retraining, both around 0.009. The Gauss-Newton variant uses the most peak GPU memory because it stores Jacobians for all time steps, with storage complexity $O(Tvd)$, where $v=100$ is the output dimension and $d=419684$ is the model dimension. The diagonal variant uses much less memory, with storage complexity $O(Td)$, since it stores only diagonal Hessian entries, but it runs more slowly than the Gauss-Newton variant because second-order derivatives are more expensive to compute. With forgetting enhancement, both runtime and memory decrease, but the approximation error increases to about 0.04. Natural forgetting benchmark offers little savings in runtime or memory compared with our unlearning algorithms, while its error increases substantially to about 0.08.
>
> 3.Assumptions: We would like to clarify that our main assumptions (Assumptions 2.3 and 2.4) are mild and standard in modern nonconvex deep learning theory. Assumption 2.4 requires only the existence of empirical minimizer within a bounded region. As long as an empirical minimizer is attained at a finite parameter value, one can always choose a sufficiently large radius $r$ so that the assumption holds. Similarly, Assumption 2.3 only requires that the Hessian be bounded within a finite-parameter region. This excludes only pathological scenarios such as regions with unbounded curvature or solutions that lie outside every finite domain. Whenever the Hessian is bounded over a compact region, finite lower and upper spectral bounds can always be specified. This is consistent with modern deep learning regimes, where model updates are often constrained to a practically bounded parameter regime. In practice, our assumptions are easily satisfied in many modern machine learning tasks, including LLM training, LoRA fine-tuning, and ResNet training on CIFAR and ImageNet.
>
> Furthermore, our analysis does not require global convexity, whereas many recent certified unlearning works still rely on it: "Basaran, Umit Yigit, et al. (ICML 2025)", "Chien, Eli, et al. (NeurIPS 2024)". Our method successfully relaxes this restriction.
>
> 4.Presentation: We acknowledge that the paper introduces substantial notation, as it connects both continual learning and unlearning. Similar notation is also used in prior work on continual learning, "Lin, Sen, et al. (ICML 2023)", and unlearning, "Qiao, Xinbao, et al. (ICLR 2025)". We will further improve the presentation for readability and move some proof-related notation and definitions to the appendix.
>
> If our clarifications and additional results have addressed your concerns, we would greatly appreciate your consideration in updating your rating. We would also be happy to answer any further questions.

---

> > ### Author Rebuttal · Reviewer_HQo4 · 2026-04-03
> >
> > Thank you for your detailed response. I’ll give you a higher score.

---

> > > ### Author Response · Authors · 2026-04-07
> > >
> > > Thank you very much for your careful review and for acknowledging our work.

---

### Official Review · Reviewer_PiXW · 2026-03-13

**Soundness:** 3
**Presentation:** 3
**Significance:** 3
**Originality:** 3
**Overall Recommendation:** 4
**Confidence:** 3

**Summary:**

The paper studies certified machine unlearning in a continual learning setting, where tasks arrive sequentially and unlearning requests can target arbitrary past tasks. It formalizes a continual learning-unlearning framework, defining a post-unlearning excess risk that decomposes into a CL excess risk term and an unlearning loss term, thereby exposing a tension between retention and forgetting. Under local smoothness and curvature assumptions, the authors derive an upper bound on the CL excess risk for non-convex models trained with a regularized CL update. They then adapt two families of certified unlearning methods to CL: a gradient-based "natural forgetting" algorithm that uses Gaussian noise calibrated to a bound on model drift, and a more accurate but storage-heavy Hessian-based algorithm that handles arbitrary (including asynchronous) unlearning requests, plus a hybrid variant that trades off storage and approximation error.

**Compliance With Llm Reviewing Policy:**

Affirmed.

**Key Questions For Authors:**

1. How sensitive are your methods to the accuracy of the Hessian approximation $\hat\nabla^2$?
Can you provide empirical evidence (e.g., experiments comparing diagonal, Gauss-Newton, and perhaps K-FAC approximations) that relate the bound constant $\nu$ in Assumption 5.1 to observed approximation error and runtime? This would help interpret Propositions 5.2 and 5.3 in practice.

2. Can you show explicit experiments contrasting forward-synchronous vs asynchronous unlearning?
For a fixed dataset and model, construct two request sequences with identical sets of tasks to unlearn but different temporal orders as in Figure 2, then compare approximation error (Figure 3(a)) and final accuracy (Figure 4). How large is the gap in practice?

3. What happens when you calibrate noise using the theoretical bound $\gamma_t(\mathbf S_{1:t})$ instead of the exact approximation error?
The current experiments rely on the simulated error, which is not realistic in practice. Please quantify how much larger the required noise becomes when using the bound, and what impact this has on post-unlearning accuracy.

4. Can you experimentally validate the resource advantage of natural forgetting and the hybrid scheme?
For your experimental setups, please report peak memory and runtime for natural forgetting (Alg. 1), full Hessian-based CLU (Alg. 2 with exact/diagonal/GN Hessian), and the forgetting-enhanced algorithm (Eq. (18)), so that the storage-accuracy claims are concretely substantiated.

**Limitations:**

Not yet.

**Strengths And Weaknesses:**

# Strengths
1. Clear and principled CLU formulation.
The two-stage CLU process in Section 2 and Figure 1 gives a clean abstraction of how learning and unlearning interleave over time, including internal vs. published models and request histories $\mathbf{S}_{1:t}$. The post-unlearning excess risk in Definition 2.2, together with its decomposition into $\mathcal{E}_L$ and $\mathcal{E}_U$, is conceptually neat and seems like a good lens for future work.

2. Nontrivial theoretical analysis for non-convex CL.
Theorem 3.1 provides an explicit upper bound on CL excess risk for $\ell_2$-regularized CL in non-convex settings under local curvature assumptions, extending previous linear analyses. Lemma B.2's recursive decomposition of the parameter path, and the dependence on $\rho = \lambda/(\mu+\lambda)$, give interpretable insight into how regularization affects drift across tasks.

3. Adapting certified unlearning to continual dynamics.
The paper systematically adapts gradient-based (Alg. 1) and Hessian-based (Alg. 2, Eq. (13)-(15)) certified unlearning to CL, including the nontrivial handling of arbitrary unlearning request sequences. The explicit modeling of asynchronous vs. forward-synchronous sequences in Figure 2, and the way this affects approximation error via the $\hat C(\cdot)$ terms, addresses an important and previously under-theorized challenge.

4. Storage-accuracy trade-off is explicitly reasoned about.
The analysis makes clear that the gradient-based method has almost zero storage cost but relatively loose unlearning guarantees (Theorem 4.1), whereas the Hessian-based method yields significantly tighter approximation error bounds (Propositions 5.2, 5.3) at $O(t\tilde d)$ storage. The hybrid "forgetting enhanced" scheme in Eq. (18) and Proposition 5.4 is a thoughtful attempt to reduce storage while leveraging natural forgetting.

5. Empirical support aligns reasonably with theory.
Figure 3(a) shows approximation error trajectories over tasks where Hessian variants consistently outperform natural forgetting, and on MNIST the Hessian-based error even decreases after a second unlearning event, reflecting the benefits of more accurate corrections. Figure 3(b) illustrates how increasing $\lambda$ decreases the final approximation error, echoing its theoretical role. Figure 4 shows post-unlearning accuracy close to retraining, especially for the Hessian-based methods.

# Weaknesses
1. Experimental scope is too limited relative to the theoretical ambition.
The paperfs theoretical development is extensive, but the experiments are confined to small CNNs on MNIST and CIFAR-10, with $T\le10$ and only 1-2 unlearning events. There is no evaluation on more challenging CL benchmarks (e.g., Split CIFAR-100, TinyImageNet, or non-vision domains), nor on longer task sequences. This mismatch undercuts claims about practical relevance and generality.

2. Lack of empirical exploration of asynchronous vs forward-synchronous unlearning.
A major conceptual point is that asynchronous unlearning sequences inflate approximation error via the $\hat C(\cdot)$ terms (Figure 2, Proposition 5.2). However, the experiments do not control or report the unlearning sequence type. Without comparing performance under explicitly forward-synchronous vs highly asynchronous schedules, this key theoretical insight remains untested.

3. No quantitative storage or runtime analysis.
While the text qualitatively discusses storage overhead $O(t\tilde d)$ and the possibility of diagonal approximations, there is no actual reporting of memory usage or runtime. For a paper that emphasizes storage as a central differentiator between gradient-based and Hessian-based unlearning, this omission is significant.

4. Limited exploration of the CL-unlearning trade-off parameter (\lambda).
Figure 3(b) shows how approximation error decreases with increasing $\lambda$, but the effect on CL performance (i.e., $\mathcal{E}_L)$ is not shown. Theorem 3.1 highlights that $\lambda$ controls both terms, but there is no experiment sweeping $\lambda$ and plotting both CL accuracy and unlearning error to demonstrate the "forgetting-retention dilemma" empirically.

5. Use of "simulated exact approximation error" to calibrate noise.
In Section 6, the authors state that they design the noise based on the "simulated exact approximation error" rather than the theoretical upper bound to "demonstrate empirical performance." This undercuts the certified-unlearning claim in practice: in any real deployment, one would not have access to the exact retraining model, only the bound. The paper does not show what happens when noise is calibrated strictly according to the theoretical $\gamma_t$, which could be significantly larger.

---

> ### Author Rebuttal · Authors · 2026-03-30
>
> We sincerely appreciate your time and thoughtful feedback. Below is our detailed response.
>
> 1. Additional experiments on CIFAR-100 dataset: We further evaluate our method on CIFAR-100 with $T=30$ non-i.i.d sequential tasks, under more complex unlearning sequences. We use a pretrained ResNet-18 backbone, replace its final layer with a three-layer fully connected head (hidden dimensions $512\times256$ with ReLU activations), freeze the backbone, and train only the head. This setup is also used in "Certified Unlearning for Neural Networks (ICML 2025)" whose model is smaller than ours with $32 \times 32$ dimensions. All experiments are conducted on a machine with an AMD 9950X CPU and an NVIDIA RTX 5080 GPU. Under the new setup, we have added Fig. A, B, C in the supplementary material https://anonymous.4open.science/r/New-experiments-CLU-8956/New_results.pdf and three tables in our responses to provide more comprehensive experimental comparison. We next present these new results in more detail.
>
> 2. Experiments on asynchronous vs forward-synchronous unlearning: We simulate the unlearning process using Alg. 2 with Gauss-Newton and diagonal Hessian under both a forward-synchronous unlearning (Fwd-Sync) sequence and an asynchronous (Async) unlearning sequence for comparison. The two sequences unlearn the same set of tasks, but at different time points:
> $$\\begin{array}{l|cc}\\hline\\mathrm{Unlearning\\ time} & \\mathrm{Fwd\\text{-}Sync\\ unlearning } & \\mathrm{Async\\ unlearning }\\\\\\hline
> 9  & 3,4,6,8 & 3,4,6,8\\\\
> 15 & 10,11,12,13,14 & 11\\\\
> 17 & \\emptyset & 12\\\\
> 18 & \\emptyset & 13\\\\
> 20 & \\emptyset & 14\\\\
> 25 & 20,23 & 23\\\\
> 26 & 24,25 & 20,25\\\\
> 28 & \\emptyset & 10,24\\\\\\end{array}$$The approximation error results are shown in Fig. B of the supplementary material. In Fig. B(a), before time 15, the two sequences have the same unlearning history, so their approximation errors are identical. From time 17 onward, although the Fwd-Sync sequence has unlearned more tasks than the Async sequence, its approximation error becomes consistently lower. By the end, the approximation error under the asynchronous sequence is about twice that under the forward-synchronous sequence, showing that the gap can be substantial. The results for diagonal Hessian in Fig. B(b) hold similarly. According to Proposition 5.2, this is because $\hat{C}(\cdot)$ inside $\beta_{i,s,a}^2$ in the second line of Eq. (16), which captures the effect of asynchrony, reduces to 0 under the forward-synchronous sequence.
>
> 3. Experiments with runtime, storage and approximation error: According to your comments, we compare runtime, peak GPU memory, and unlearning error across our algorithms and benchmarks. Please refer to Response 2 to Reviewer HQo4 for details.
>
> 4. Sensitivity to the accuracy of Hessian approximation: Our method is not sensitive to $\nu$, because it is small relative to $\lambda$ in determining the value of $\hat{\rho}$ in our theoretical bound. Empirically, we estimate the Hessian approximation error for both the Gauss-Newton Hessian and the diagonal Hessian in practice, and observe that the approximation error of the diagonal Hessian is consistently slightly larger than that of the Gauss-Newton Hessian (See Response 2 to Reviewer JMs5 for more details.) However, in our experimental results, the resulting unlearning approximation error under the diagonal Hessian is often not much worse than that under the Gauss-Newton Hessian, as shown in Fig. A of the supplementary material, and even outperforms the Gauss-Newton Hessian. Another explanation is that the diagonal Hessian is more naturally positive semidefinite, which can make the Hessian-based method more stable in practice.
>
> 5. Experiments with trade-off between CL and UL: We simulate the CL process without unlearning by $\ell_2$-CL algorithm to evaluate the final CL risk $\mathcal{E}_L$, and the CL-UL process by Alg. 1 and 2 to evaluate the final approximation error, under different values of $\lambda$, shown in Fig. C of supplementary material. Fig. C(a) shows that the approximation error decreases as $\lambda$, because a larger regularization term controls the model updates, thereby reducing the error. Fig. C(b) shows that test accuracy under pure CL decreases as $\lambda$ increases, since a larger $\lambda$ may hinder the model from learning new knowledge. This demonstrates that achieving a low $\mathcal{E}_L$ does not necessarily imply a low $\mathcal{E}_U$, illustrating the Forgetting-Retention Dilemma.
>
> 6. Calibrating noise under $\gamma_t$: We now calibrate the noise according to the approximation error upper bound $\gamma_t$ for certified unlearning. Please refer to our Response 2 to Reviewer JMs5 for details.
>
> If our clarifications and additional results have addressed your concerns, we would greatly appreciate your consideration in updating your rating. We would also be happy to answer any further questions.

---

> > ### Author Rebuttal · Reviewer_PiXW · 2026-04-03
> >
> > Satisfied.

---

> > > ### Author Response · Authors · 2026-04-07
> > >
> > > Thank you very much for your careful review and for acknowledging our work.

---

### Decision · Program_Chairs · 2026-04-30

**Decision:**

Accept (regular)

**Comment:**

This paper tackles a timely and nontrivial problem: extending certified machine unlearning to the continual learning setting, where the model evolves sequentially across tasks and deletion requests may arrive asynchronously. The formulation — a CLU framework decomposing post-unlearning excess risk into a CL retention term and an unlearning loss — is clean and well-motivated. The distinction between asynchronous and forward-synchronous unlearning sequences in Figure 2 is a meaningful and previously under-theorized contribution, and the hybrid method combining natural forgetting for older tasks with Hessian-based correction for recent requests is a sensible design choice.

Scores:
PiXW: weak accept, confidence 3
HQo4: weak accept, confidence 3
jMs5: accept, confidence 3

Reviewer PiXW fully resolved concerns after the rebuttal. Reviewers jMs5 and HQo4 retained concerns primarily around the experimental scope (MNIST and CIFAR-10 only, small CNNs, very few deletion events) and the gap between the theoretical procedure and the empirical validation (which uses simulated rather than theoretically calibrated noise). The authors responded with results under the theoretical noise calibration, though the scope remains limited. Presentation also needs work.

The theoretical contribution is the paper’s core selling point: this appears to be the first formal framework bridging certified unlearning and continual learning. The empirical evaluation is thin by ICML standards and limits the paper’s impact. I’d have liked to see larger-scale experiments, but given the theoretical nature of the paper, I lean toward accepting. The authors should commit to expanding the experimental scope in their revision.